# Modelling evaporation with local, regional and global BROOK90 frameworks: importance of parameterization and forcing.

Ivan Vorobevskii, Thi Thanh Luong, Rico Kronenberg, Thomas Grünwald, Christian Bernhofer

Faculty of Environmental Sciences, Department of Hydrosciences, Institute of Hydrology and Meteorology, Chair of Meteorology, Technische Universität Dresden, Tharandt, 01737, Germany

*Correspondence to*: Ivan Vorobevskii (ivan.vorobevskii@tu-dresden.de)

**Abstract.**

Evaporation plays an important role in the water balance on different spatial scales. However, its direct and indirect measurements are globally scarce and accurate estimations are a challenging task. For the correct process approximation in modelling of the terrestrial evaporation is still difficult. A physically-based 1D lumped soil-plant-atmosphere model (BROOK90) is applied to study the role of parameter selection and meteorological forcing for the simulation of evaporation at the point scale. By the integration of the model into global, regional and local frameworks, cross-combinations were elaborated out of their parameterization and forcing schemes to analyze and show their roles in the estimation of evaporation.

Five sites with different land uses (grassland, cropland, deciduous broadleaf forest, two evergreen needleleaf forests) located in Saxony, Germany were selected for the study. All tested combinations showed a good agreement with FLUXNET measurements (KGE values 0.35-0.80 for a daily scale). For most of the sites, the best results were found for the calibrated model with in-situ meteorological input data, while the worst were observed for the global setup. The setups' performance in the vegetation period was much higher than for the winter period. Among the tested setups, the model parameterization leads to a higher spread in the model performance than it was observed due to the meteorological forcings. The analysis of the evaporation components revealed that transpiration dominates (up to 65-75 %) in the vegetation period, while interception (in forests) and soil/snow evaporation (in fields) prevails in the winter months. Furthermore, it was found that different parameter sets impact the model performance and redistribution of evaporation components throughout the whole year, while the influence of meteorological forcing was evident only in summer months. Finally, the results suggest that ERA5 data might serve as reasonable meteorological forcing for evaporation simulations even at a local, respectively point scale.

## 1 Introduction

Evaporation as a water balance component plays an important role in the hydrological process at multiple spatial scales: from a single leaf to an entire catchment. As a result of mass and energy exchange between the soil-plant and atmosphere system, the global annual terrestrial evaporation amount yields approximately ⅔ of the total precipitation (McDonald, 1961), showing

however large range even on a macroscale (Haddeland et al., 2011; Harding et al., 2011; Miralles et al., 2016). However, with

the need of higher spatial and temporal resolution, the high variability of evaporation should be taken into account and properly

addressed (Anderson et al., 2007; Baldocchi et al., 2001; Jung et al., 2011; Pan et al., 2020; Zhang et al., 2010). Thus, accurate

estimates of evaporation on different scales as well as advanced understanding of the process itself, are beneficial for planning,

developing and monitoring of hydrologic, agriculture and ecological systems, e.g., irrigation scheduling, water distribution

systems, crop modelling, quantification of energy and moisture exchange between the land surface and the atmosphere (Fisher

et al., 2017; McNally et al., 2019; Schulz et al., 2021). Apart from the total evaporation itself, it is sometimes necessary to

assess and quantify its components (Chang et al., 2018; Lawrence et al., 2007; Leuning et al., 2008; Schulz et al., 2021),

namely components, like transpiration, evaporation from the ground or snow surface, and evaporation of intercepted rain and

snow from the canopy. However the partition of the evaporation is a subject of a large variability and depends not only on the

location, but on scale as well (Wei et al., 2017; Zhang et al., 2017).

Various direct (i.e. porometer, eddy-covariance and lysimeter) and indirect (catchment water balance, energy balance,

theoretical models based on meteorological data) methods have been developed and used to measure evaporation at different

spatio-temporal scales. Each method has its strengths and weaknesses, but what they have in common is that the results have

limited representativeness. Namely, they are valid only within a certain space of scale and time(so-called "footprint"), which

is usually quite small, thus only a local scale could be represented by it (Baldocchi, 1997; Wilson et al., 2001). Recently, these

methods were extended to include remote sensing techniques for the regional and global scale (Anderson et al., 2008; Leuning

et al., 2008; Miralles et al., 2011, 2016), but the quality of the output products possess still a potential for improvement (Pan

et al., 2020; Zeng et al., 2012). Among the datasets of the in-situ evaporation measurements, the FLUXNET network

(http://www.fluxnet.ornl.gov) provides eddy-covariance data from about 500 stations worldwide within FLUXNET2015

dataset (Pastorello et al., 2020) and still acts as the main driver in advancing evaporation research (Baldocchi et al., 2001; Jung

et al., 2011; Mauder et al., 2018). Evaporation measurements are still scarcely available due to high costs and the problem of

large-scale representability (in comparison to e.g. discharge measurements).

Hence, mathematical modelling in favour of its feasibility is a practical substitute. Besides empirical formulas (Cerro et al.,

2021; Feng et al., 2016; Zeng et al., 2012), evaporation is often estimated by physically-based models (Beven et al., 2021;

Boulet et al., 2015; Liu et al., 2012; Mallick et al., 2018), in which Penman-Monteith (and Shuttleworth and Wallace extension)

formula is one of the most frequently used. This approach reduces potential evaporation to an actual one accounting for the

available water in the soil-plant system. Thus, it is incorporated into many land surface models and frameworks regardless of

scale: local, regional or even global (Leuning et al., 2008; Mallick et al., 2018; Zink et al., 2017). Despite many efforts to

improve evaporation models on different scales, large uncertainties still remain (Allen et al., 1998; Miralles et al., 2011;

Mueller et al., 2011). In general, the sources of evaporation modelling (or more in general – hydrological modelling)

uncertainties can be classified as following: model structure and process representation, choice of an appropriate parameter

set, meteorological input data, spatio-temporal miss-scaling and uncertainties of measurements for the model validation themselves (Mallick et al., 2018; Mauder et al., 2018; Mueller et al., 2011; Zhang et al., 2010). Studying these sources of uncertainties from different approaches and frameworks gained more attention in recent years, however most of these studies are limited by the focus on one single spatio-temporal scale (Chang et al., 2018; Jung et al., 2011; Liu et al., 2012). Only a few researchers focused on investigations of the uncertainties in multiple frameworks with multiple input datasets and simultaneously accounting for point, regional and global scales (Pan et al., 2020; Su et al., 2005; Winter and Eltahir, 2010).

Here we aim to extend the knowledge on evaporation estimations based on the soil-plant-atmosphere physically-based lumped BROOK90 model, which we integrated into three frameworks. These frameworks use different "state-of-the-art" sources of data for the model parameterisation and forcing which represent various spatial scales. Namely these scales are global, regional and local. By mixing these different datasets and validating the simulated evaporation with eddy-covariance measurements, we want to show dependencies of the spatial scale of BROOK90 model parameterization and forcing data on the accuracy of evaporation estimates. Our main hypothesis is that the goodness of fit of the setups increases from global to local scale with respect to the parameterization as well as to the forcing. However, it was unclear how the scale combinations will perform, i.e. local meteorological data with global parameterization and vice versa. Therefore, this study presents the first qualitative analysis of the model input scale uncertainty exemplarily, based on the best- globally and locally available data sets. Therefore, this study possesses a practical outcome. Namely in the presence of limited resources and data, first conclusions about the reliability of evaporation estimates for a point (hydrological response unit) scale can be drawn from the global or regional BROOK90 frameworks. Moreover, the study points to a direction where the BROOK90 user should put more attention – accurate parameterization or meteorological input. Thus, the outcome of this study provides a better understanding of the BROOK90 model as well as shows the directions to improve effectively evaporation simulations.

## 2 Material and methods

### 2.1 Study sites and eddy-covariance measurements

The evaluation of simulated evaporation was carried for five sites with various land covers and long-term eddy-covariance measurements (Fig. 1, Table 1). All selected towers are located in Saxony, Germany. The study area is characterized by temperate suboceanic/subcontinental climate (Cfb, Kottek et al., 2006). The average mean daily temperature varies between +15 $^0$C and +20 $^0$C in summer months and between -5 $^0$C and +5 $^0$C in winter months. The average annual precipitation varies between 750 mm and 960 mm. The measurements of atmospheric fluxes with standardized methods are operated by Technische Universatät Dresden within ICOS and FLUXNET projects. In this study, we used daily evaporation values calculated from measured latent heat fluxes corrected for the observed site-specific energy budget closure gap. In general, from 10 (Hetzdorf) up to 23 (Tharandt) years of continuous time-series are available.

The Grillenburg site (DE-Gri, the sensor height is 3 m above the ground) is a permanent and extensively managed (one to three

cuts per year) flat-terrain grassland (mesophytic hay meadow). Regular mowing usually takes place in June and September. In

the case of three cuts per year, the second one is usually done in July. Typical plant species include couch grass (*Elymus*

*repens*), meadow foxtail (*Alopecurus pratensis*), common yarrow (*Achillea millefolium*), common sorrel (*Rumex acetosa*) and

white clover (*Trifolium repens*). The area is generally used for forage and rarely for pasture. Vegetation height is measured

once per week, with the lowest values (5-10 cm) measured at the beginning of growing season or after cutting and highest

values (typically 30-40 cm, maximum 90 cm) in the summer before cutting. Although the LAI was only occasionally measured,

the significant correlation between vegetation height and LAI made it possible to interpolate the annual range. Therefore, the

range of LAI was estimated between 0.25 $m^2$ $m^{-2}$ and 5 $m^2$ $m^{-2}$ in the yearly course. The topography around the site promotes

cold air deposition, thus daily minima of air temperature are often much lower than at the other sites. The site is mainly

characterized by gleysol soil that contains silty loam, loam, and loamy silt as soil textures.

The Klingenberg site (DE-Kli, the sensor height is 3.5 m above the ground) is an intensively farmed arable land located 4 km

south from the Tharandt forest (Fig. 1). This site is characterized by annual and inter-annual crop rotation of rapeseed (*Brassica*

*napus*), winter wheat (*Triticum aestivum*), forage maize (*Zea mays*), spring barley (*Hordeum vulgare*) and winter barley

(*Hordeum vulgare*) with occasional intercropping. As a result, plant cover, vegetation height, LAI and rooting depth varied

greatly across time periods, i.e. measured annual maximum canopy height values vary between 0.7 m and 2.2 m and LAI could

reach up to 6 $m^2$ $m^{-2}$. Soil properties and runoff behaviour are strongly influenced by tillage and fertilizer application.

According to the (Ad-hoc-AG Boden, 2005), the soil was classified as gleysol and has a clay or loam texture.

The Hetzdorf site (DE-Hzd, the sensor height is 5 m (2010-2017), 11.5 m (2017-2021) and 17.5 m (since 2021) above the

ground) is a young oak (*Quercus robur*) forest planted after the Kyrill storm in 2007, which caused severe windthrow (40 ha)

in an old Norway spruce (*Picea abies*) forest. This site has a moderate slope to the North and a main wind direction to the

South due to a gap in the surrounding old spruce forest. The young oak stand is approximately 8-10 m high (2021) and enclosed

by spruce forest (up to 30 m height). Due to the high amount of deadwood and the young oak plantation until 2017 this

ecosystem was a net $CO_2$ source, but since 2018 it already acts as a moderate $CO_2$ sink (Drought 2018 Team and COS

Ecosystem Thematic Centre, 2020; Warm Winter 2020 Team and COS Ecosystem Thematic Centre, 2022). As a young

growing site, LAI varies dynamically from year to year and was only measured sporadically. The site is dominated by

pseudogley soil with a silt and silty loam texture.

The Tharandt site (DE-Tha, the sensor height is 42 m above the ground) is a 120-year-old mixed conifer forest with a mean

canopy height of 30 m, consisting mainly of Norway Spruce (*Picea abies*, 80 %), European larch (*Larix decidua*, 18%), and

various other evergreen and deciduous tree species (2 %) such as Scots pine (*Pinus sylvestris*), silver birch (*Betula pendula*)

and mountain ash (*Sorbus aucuparia*). Root depth amounted between 30 cm and 40 cm, relative to the predominant Spruce

tree. The forest was thinned five times (1983, 1988, 2002, 2011 and 2016) and European beech (*Fagus sylvatica*) and Silver

fir were planted in the understorey in 1995 and 2017, respectively. The site has silty podzol soils with relatively high stone

content (10-20 %). These soils were developed from a periglacial sediment consisting of debris from rhyolite and loess and

are very heterogeneous.

The Oberbaerenburg site (DE-Obe, the sensor height is 30 m above the ground) is an 80-year-old dense evergreen forest 15-

17 m height with predominantly Norway spruce trees (*Picea abies*). In contrast to the other sites, this site is located much

higher (734 m a.s.l.) with a prevailing NW wind direction and mean temperature and precipitation of $6.9^0$C and 960 mm,

respectively. Spruce density has been thinned over the years (e.g., 1057 trees ha$^{-1}$ in 1994, 987 trees/ha in 2000, 884 trees ha$^{-1}$

in 2005, and 846 trees ha$^{-1}$ in 2011). However, this has had little effect on the site characteristics. The soil is characterized as

podzol and has a sandy texture with high stone content (20-40 %).

According to on-site measurements, the groundwater tables for all sites are at least 3 m deep, thus it is assumed, that there is

no significant influence of groundwater on the water demand for the evaporation.

Due to the principles of eddy-covariance measurements, the observed fluxes refer to a certain footprint that varies depending

on wind speed, wind direction and atmospheric stability. Moreover, it is also affected by the height of measurement and the

surface roughness. According to long-term micro-meteorological measurements around the study sites, it was found that in

relation to predominant weather conditions the area of the highest flux density of the eddy-covariance signal (90 %) was within

a radius of 120-380 m. The values differ significantly among sites, but not greatly between wind directions (< 10 %). Thus,

equidistance footprints for each station (red circles on Fig. 1, shape files can be found in Supplementary) were assigned as

mean values from all wind directions. These values are further used in the simulations in model frameworks.

Selected daily evaporation data and other climatological variables can be found in the Supplementary.

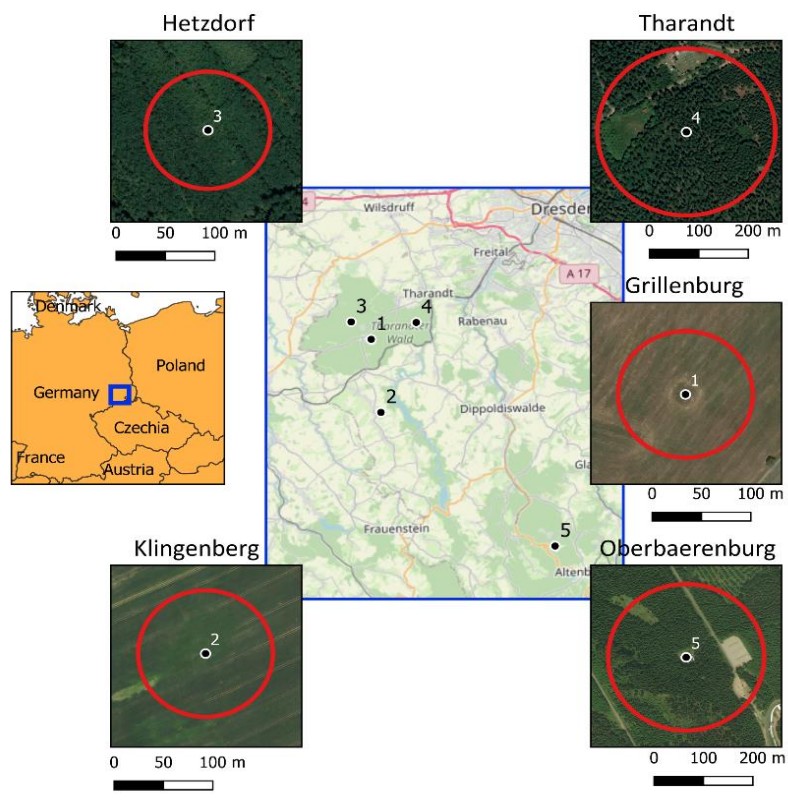

**Figure 1. Location of chosen FLUXNET sites. Red circles represent footprints for each tower. OpenSteet Maps (Planet dump retrieved from https://planet.osm.org) and Bing Satellite images (BingTM Maps tiles, 2020) are used as a background.**

**Table 1. Short summary on the chosen FLUXNET sites.**

| ID | Site name | Latitude | Longitude | Available data | Footprint, m | Dominant soil type | Land cover type |
|----|-----------|----------|-----------|----------------|--------------|-------------------|-----------------|
| 1 | Grillenburg | 50.950 | 13.513 | 2003-2020 | 135 | gleysol | Permanent grassland |
| 2 | Klingenberg | 50.893 | 13.522 | 2005-2020 | 135 | gleysol | Agriculture (with crop rotation) |
| 3 | Hetzdorf | 50.9641 | 13.490 | 2010-2020 | 125 | pseudogley | Young oak forest (after storm) |
| 4 | Tharandt | 50.963 | 13.565 | 1997-2020 | 360 | podzol | Old spruce forest |
| 5 | Oberbaerenburg | 50.787 | 13.721 | 2008-2020 | 350 | podzol | Spruce forest |

**2.2 BROOK90 model**

BROOK90 (Federer et al., 2003) is a 1D process-oriented model for simulation of vertical water fluxes in soil-plant-atmosphere systems. Precipitation input (snow or rain) first goes through the canopy, where it could be intercepted and then evaporated. The portion, which reaches ground level, could be infiltrated, frozen, evaporated, converted to surface flow, percolated or stored as soil moisture. Infiltrated water follows a top-down approach as a macropore bypass and matrix flow. The soil column

has groundwater, seepage and downslope outflow. Finally, soil water storage is used for evaporation and transpiration. The

model has more than 100 physically-based input parameters, but typically most are straightforward and can be set easily (as

location or slope). As the study mainly reflects evaporation, this part of the model is described in more detail.

The model uses a two-layer version of Penman-Monteith (PM) equation by Shuttleworth-Wallace (SW) (Shuttleworth and

Wallace, 1985) to estimate the potential evaporation (PE) separately for canopy and soil surface accounting for the surface

energy budget and the gradient for the sensible heat flux respectively. Canopy-dependent PE consists of evaporation of

intercepted snow and rain and plant transpiration. It is defined as the maximum evaporation that would occur from a given

land surface under given weather conditions if all plant and soil surfaces were externally wetted. Surface-dependent PE

includes evaporation from soil and snow surfaces. It is defined as the maximum evaporation that would occur from a given

land surface under given weather conditions if plant surfaces were externally dry and soil water was at field capacity. The SW

method considers multiple resistances like the above canopy, within canopy from canopy and ground, canopy surface, vapour

movement in soil. They are applied in the standard PM equation, thus giving separate estimates of all five components of PE.

It should be noticed, as BROOK90 distinguishes between soil and plant evaporation, only one canopy process and one ground

process can occur at a given timestep. Subsequently, actual evaporation (E) is based on the water availability in the system

(within the canopy, on the soil and within the soil matrix). Daily evaporation rates are calculated as a weighted sum of the

daytime and night-time values (based on the sunshine duration); however, interception could be estimated at a higher frequency

(hourly).

Originally, the model was written in FORTRAN programming language, here we used an R 'line-by-line' direct translated

version (Kronenberg and Oehlschlägel, 2019).

**2.3 Model frameworks and parameterization schemes**

In the study, four different scale-dependent setups for the model are used to simulate evaporation and its components: Global

BROOK90, EXTRUSO, BROOK90 with manual parameterization and calibrated BROOK90. To parameterize the model for

global, regional and local scale different topography, soil and land cover datasets were utilized. Most of the model's physical

parameters are either default and thus fixed by the model developer or valid for the whole model region (i.e. average duration

of rain precipitation per month). Variable site-specific parameters (around 40 depending on the setup) and their values for all

tested frameworks are listed in Appendix C (Table C1).

2.3.1. Global BROOK90 (GBR90)

The Global BROOK90 (GBR90) framework incorporates open-source global datasets for parameterization and forcing of the

model using an R-package (Vorobevskii et al., 2020). The main feature of the package is wrapping of the modelling process

in a fully automatic mode based only on the location and time-interval input. The input area of interest is divided in a regular

50x50 m grid, and then hydro response units (HRU) are identified based on the unique combinations of land cover, soil

characteristics, and topography (aspect and slope). GBR90 provides fixed parameter sets for 20 land cover types based of

Copernicus Global land Cover 100 m (Buchhorn et al., 2020): closed and opened forest (evergreen/deciduous, needle/broad

leaf or mixed, and unknown), shrubs, herbaceous vegetation, moss and lichen, bare/sparse vegetation, cultivated and managed

vegetation, urban territories and snow/ice. Additionally, Leaf Area Index (LAI) and tall canopy height parameters were

assigned using MODIS 8-day composite dataset with 500 meter resolution (Myneni et al., 2015) and Global Forest Canopy

Height with 30 m resolution (Potapov et al., 2021) respectively. The SoilGrids250 dataset (Hengl et al., 2017) provides global

information on standard soil properties with 250 m resolution. Number of soil layers, stone fracture and profile depth

parameters are directly derived from this dataset, while soil hydraulic parameters are assigned from the standard model

developer's sets based on the derived USDS soil texture class. Amazon Web Service Terrain Tiles (Mapzen Data Products,

2020) are used as provider for the global digital elevation model data (SRTM30 in case of Saxony). The model is applied

separately to each HRU and an area-weighted mean is calculated. A more detailed description of the framework is presented

in (Vorobevskii et al., 2020).

2.3.2. EXTRUSO (EXTR)

EXTRUSO (EXTR) is a semi-automatic framework for spatial water balance simulations on a regional scale limited to the

domain of Saxony, Germany and is distributed via R-package (Luong et al., 2020). The HRU subset is also based on the

overlay of soil and land cover types derived from the regional datasets. Due to specifics of these datasets (polygons rather than

regular grid rasters) HRUs do not have regular dimensions. The framework has fixed parameterization for 5 land cover types

(agriculture/cultivated land, deciduous forest, evergreen forest, grassland/meadows, urban/other territories). They are assigned

according to the European land cover map CORINE 2012 (European Environment Agency, 2020) with 100 m resolution (some

vegetation types from the map are generalized). Soil parameters are assigned similarly to GBR90, but using Saxon soil map

BodenKarte50 (Sächsisches Landesamt für Umwelt, Landwirtschaft und Geologie, 2020) with 50 m resolution. The 10 m

digital elevation model (Staatsbetrieb Geobasisinformation und Vermessung Sachsen, 2020) is used for slope and aspect

estimates. As in GBR90, BROOK90 is run for each HRU and an area-weighted mean is stored. A full description of the

framework is available in (Luong et al., 2020).

2.3.3. BROOK90 (BR90) with "expert-knowledge" parameterization

Finally, we made a setup using the original BROOK90 model (BR90) with manual parameterization based on field

measurements. These include long-term observations of the different canopy parameters conducted on the chosen FLUXNET

sites (height, LAI, conductivity, albedo), soil profile data (soil texture, depth, stone fracture) and expert knowledge (i.e.

interception parameters).

## 2.3.4. Calibrated BROOK90 (CBR90) as a benchmark

The calibrated BROOK90 (CBR90) serves as a benchmark for all other runs. For the calibration of BROOK90, we choose a multi-objective optimizer recently developed for the calibration of hydrological models. The algorithm is a hybrid of the MEAS algorithm (Efstratiadis and Koutsoyiannis, 2005), which uses the method of directional search based on the simplexes of the objective space and the epsilon-NSGA-II algorithm with the method of classification of the parameter vectors archiving management by epsilon-dominance (Reed and Devireddy, 2004). A pareto-optimal solution was used to address two issues. First, as most of total annual evaporation occurs in the vegetation period, it is reasonable to separate this period as the contribution of the winter months should have lesser 'weight' during model fitting. Second we tried to account for possible systematic errors of eddy-covariance measurements themselves, which could vary significantly depending on the season (Hollinger and Richardson, 2005; Twine et al., 2000; Widmoser and Michel, 2021). Therefore, the pareto front could help to choose an optimal parameter set, namely enhancing winter month performance with insignificant loss of performance in vegetation period).

Here, we performed calibration and validation with a 70 % – 30 % data split focusing on maximising daily KGE values for total evaporation for the growing season (March-October) and the winter period (November-February). The initial parameter sets were set by "expert-knowledge". For the calibration we initially took the 'location' parameters within a physically meaningful range, which are recommended by the developer and other researchers as the most sensible (Groh et al., 2013; Habel et al., 2021; Schwärzel et al., 2009; Vilhar, 2016). After the manual sensitivity analysis conducted using the given site-specific data, 21 parameters were chosen. In general, these include albedo, vegetation and flow characteristics. Meteorological forcing was derived from in-situ measurements. The total number of trials was limited to 1000 model runs, which was sufficient to achieve stable performances for all three optimization functions.

Results of the calibration and validation are presented in Table 2. A complete list of chosen parameters with given ranges and a graphical overview of the resulting Pareto fronts for each site are provided in Appendix C (Tables C1 and C2). The raw outputs of calibration results for all trials with optimized parameters can be found in the Supplementary. It can be stated that calibration and validation showed satisfactory results for the vegetation period even on a daily scale, while the results for the winter time were poor at most sites (more in detail in Sect. 5.2 and 5.3).

**Table 2. Daily Kling-Gupta-Efficiency for BROOK90 calibration and validation.**

| ID | Site name | KGE (Vegetation period) | | KGE (Winter period) | |
|---|---|---|---|---|---|
| | | Calibration | Validation | Calibration | Validation |
| 1 | Grillenburg | 0.89 | 0.81 | 0.49 | 0.44 |
| 2 | Klingenberg | 0.72 | 0.67 | 0.19 | -0.03 |
| 3 | Hetzdorf | 0.82 | 0.75 | 0.30 | 0.17 |

| 4 | Tharandt | 0.72 | 0.69 | 0.26 | 0.14 |
| 5 | Oberbaerenburg | 0.72 | 0.61 | 0.02 | -0.94 |

## 2.4 Meteorological forcings

We have chosen ERA5 (Copernicus Climate Change Service (C3S): ERA5: Fifth generation of ECMWF atmospheric reanalyses of the global climate. ERA5 hourly data on single levels from 1979 to present., 2020), RaKliDa (Kronenberg and Bernhofer, 2015) and in-situ station measurements to represent the global, regional, and local scales, respectively, as meteorological forcing for the model. The list of standard climatological variables required to run BROOK90 consists of minimum and maximum 2 m air temperature, mean 10 m wind speed, solar radiation on the horizontal surface, vapour pressure, and precipitation. Typically, daily data is required; however, if available, sub-daily precipitation data is more favourable.

The ERA5 is a global climate reanalysis dataset from Copernicus and European Centre for Medium-Range Weather Forecasts, available from 1950 to near real time at hourly resolution. It was derived using data assimilation principles by combining a global physical model of the atmosphere and observations from around the world. The original model resolution is 0.28125°, which corresponds to about 31x20 km rectangle in the area of interest. For the present study, data from the nearest to each site ERA5 grid was downloaded and processed by aggregating hourly to daily values.

RaKliDa is an open-source daily climatological dataset covering the south-eastern part of Germany (namely Saxony, Saxony-Anhalt and Thuringia) with a time span of 1961-2020. The original station data from the German Meteorological Service and the Czech Hydrological Meteorological Institute are first corrected for wind errors (Richter, 1995) and then interpolated on a 1x1 km grid using the Kriging indicator (Wackernagel, 2003). This approach is intended to reflect the orographic influence of downwind and upwind effects and to account for convective and small-scale precipitation events. As with ERA5, the nearest grid to each tower grid was used.

Daily meteorological data was taken from standard climate stations located in close proximity to the eddy-covariance towers. Exception is the wind speed, which is measured on the same height with eddy-covariance. In addition, the available net radiation was assimilated above the canopy. Prior data analysis revealed up to 15 % of missing values (depending on location and variables). Since these values are generally not drastic, the majority of the missing parts fall within the model "warm-up" period, and the variance of the most problematic variable (wind speed) within a site is not very high; it was decided to fill the gaps with simple monthly averages.

All of the inputs required by BROOK90 are directly available in all three data sets, except for the vapour pressure, which was calculated using dew temperature data (Murray, 1967) for ERA5 and mean daily temperature with relative humidity for two others (Magnus formula).

The meteorological data prepared for BROOK90 can be found in Supplementary. A graphical overview of the differences
between three data sets is presented in Appendix A.

Of the six input meteorological variables, net solar radiation and precipitation have the biggest influence on evaporation.
Global radiation in the gridded datasets showed minor but systematic overestimation compared to measurements on the mean
daily scale (around 1 MJ*m$^{-2}$*day$^{-1}$ in winter and 2-3 MJ*m$^{-2}$*day$^{-1}$ in summer months). However, summer variations (peaks
and minimums) are underestimated probably due to cloud coverage problems in ERA5 and RaKliDa. Precipitation showed a
much larger and non-systematic difference between the three datasets. In general, higher mean daily precipitation was
measured from September to March in Grillenburg, Hetzdorf and Tharandt (0.5-2 mm*day$^{-1}$). However, when looking at the
BIAS values (Table 3), a negative BIAS is typical for both datasets (except Klingenberg for both and Tharandt for RaKliDa).
The behaviour of the vegetation and winter periods separately follows the annual BIAS. Temperature and available vapour
pressure appear to be consistent, with 1-3 degree and 0.01-0.03 kPa respectively variation from measurements in the summer
276 months. The exception is Oberbaerenburg, where the maximum temperature and available vapour pressure from ERA5 and
277 RaKliDa have higher deviations, probably due to neglecting higher altitude in the datasets. Finally, wind speed possesses a
278 systematic positive BIAS (1-2 m*s$^{-1}$) for all months, except for ERA5 in forests and Klingenberg.

Table 3. Precipitation BIAS (to in-situ measurements).

| Site name | Meteo Dataset | Year | Vegetation period | Winter period |
|---|---|---|---|---|
| Grillenburg | | 0.91 | 0.95 | 0.83 |
| Klingenberg | | 1.05 | 1.05 | 1.05 |
| Hetzdorf | ERA5 | 0.92 | 0.96 | 0.85 |
| Tharandt | | 0.96 | 1.01 | 0.85 |
| Oberbaerenburg | | 0.76 | 0.85 | 0.59 |
| Grillenburg | | 0.88 | 0.92 | 0.8 |
| Klingenberg | | 1.04 | 1.02 | 1.08 |
| Hetzdorf | RaKliDa | 0.88 | 0.93 | 0.77 |
| Tharandt | | 1.15 | 1.16 | 1.12 |
| Oberbaerenburg | | 0.71 | 0.78 | 0.57 |

**2.5 Evaluation of parameterization and forcings combinations**

To assess the sensitivity of the BROOK90 to different parameter and meteorological inputs with regard to the evaporation
simulations, we propose to create different combinations of the framework's parameterizations from global, regional and, local
schemes and meteorological inputs from global, regional and local datasets (Fig. 2). Additionally, we tested the sensitivity of
the setups to the temporal resolution of the forcing data (hourly and daily for ERA5).

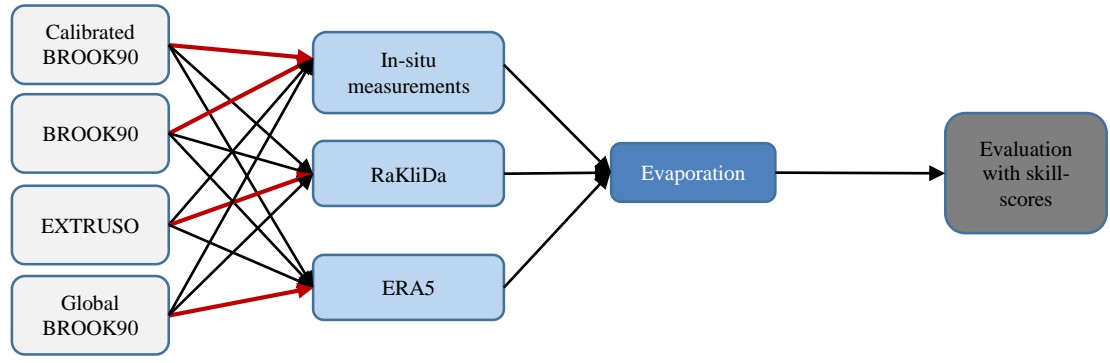

**Figure. 2. Principal scheme of the framework's mixture. Red arrows represent the original "parameter set – meteorological forcing" combination.**

From the model runs, we extracted total evaporation and its five components: transpiration, evaporation of intercepted snow

and rain, evaporation from soil, and snow evaporation. These results were evaluated on daily and monthly scales for the whole

290 year and separately for the winter and vegetation periods using the following performance metrics: Mean Absolute Error,

Nash-Sutcliffe Efficiency (NSE) (Nash and Sutcliffe, 1970) and Kling-Gupta Efficiency (KGE) (Gupta et al., 2009). The last

one can be decomposed into three main components important to assess process dynamics: correlation, BIAS, and variability

errors. The formulas and optimal ranges for each performance metrics are listed in Appendix B.

Additionally, to test the uncertainty of the obtained performance, a small data resampling experiment was designed (here only

for the daily KGE values). It helps to show the possible performance spread due to general time-series shortage and occurrence

of some extreme years (e.g. like wet 2003 and 2012 or dry 2018 and 2019). Thus, for each station we calculated multiple KGE

values with reduced time-series length by randomly (1000 samples with replacement) throwing away 3 years of data (same for

all cross-combinations). Obtained values serve to assess the possible KGE spread for each framework and meteorological

dataset.

**2.6 FAO grass-reference evaporation**

The FAO approach was chosen for the comparison with the BROOK90 model. Both of them are based on the Penman-Monteith

equation. The FAO approach is considered state-of-the-art for grass-reference evapotranspiration estimates (Paredes et al.,

2020; Sentelhas et al., 2010). Potential daily evaporation values are obtained on the basis of a simplified Penman-Monteith

approach with radiation (shortwave and longwave), air temperature, wind speed and humidity as input data (Allen et al., 1998).

The approach simplifications are concerning the aerodynamic and surface resistances calculations.

## 3 Results

### 3.1 Daily and monthly total evaporation

At first, a visual analysis of the modelled evaporation was performed. Therefore, daily (for 2020) and monthly (for the whole period with available measurements) time-series (Appendix D), monthly quantile-quantile (Fig. 3) and mean monthly (Fig. 4) plots were analysed.

Daily evaporation of 0-0.5 mm in winter and up to 6-7 mm in summer months (with a maximum of about 10 mm) was found for the Grillenburg's grassland. All model setups showed similarly low values in November-February. The growing period (March-May) was represented with a delay of 3-4 weeks for GBR90 and EXTR and 2-3 weeks for BR90. Calibration helped to eliminate this time shift on a monthly scale, however at the same time enhancing the unreasonably high variability on a daily scale. During the summer months (June-August), the frameworks suffered from the systematic overestimation of variance ratio and underestimation of the mean values, which is especially noticeable within the higher evaporation values range. Moreover, monthly maximum values vary from year to year due to differences in the timing of grass cuts. Evaporation in autumn is well captured but advanced by 2-3 weeks in EXTR and BR90. Finally, the difference between meteorological datasets is only noticeable in the summer months.

In Klingenberg's crop field, evaporation of 0-1 mm in winter and 4-6 mm in summer months (with maximum around 9 mm) is usually observed. In most of the years, all model setups showed a similar small overestimation in November-January. It was relatively difficult to achieve a good model fit regarding the timing of the growing and harvesting periods even on a monthly scale. Since both periods of the various crops differ by up to two months and the annual rotation with clear cuts are irregular. The growing period (February-May) had in general a delay of 2-6 weeks. Here CBR90 shows higher daily evaporation values, thus fitting low BIAS, while the variance ratio stays underestimated. In contrast with the grassland site, summer months (June-August) did not depict a high BIAS, the main problem appears in a considerable scattering due to poor correlation, which is higher in the middle part of QQ-plot. Furthermore, the different setups showed different peak values in the summer months, BR90 matched observations in June, while GBR90 and EXTR showed the maximum in July. Finally, in autumn, none of the setups provided satisfactory results, namely both over- and underestimations, especially in September and October. Again, based on the meteorological datasets, the variability of the model performance is visible only in the summer months.

For the Hetzdorf deciduous broadleaf forest, typical values of winter and summer evaporation are 0-1 mm and 3-5 mm (with maximum around 8.5 mm), respectively. All model setups showed small amounts of evaporation in winter with a low BIAS, but also low correlation. The main leaf development period (March-May) was represented well by GBR90, with a 2-3 weeks' time lag in April for EXTR and BR90. In the summer months (mostly in June and July) GBR90 and EXTR underestimated evaporation by 10 %, while 'expert knowledge' BR90 gave positive BIAS. It can be noticed on the monthly plots that as the forest keeps developing and growing intensively within the last 10 years, higher evaporation rates were observed from year to

year. At the same time due to model parameter stationarity, BR90 shows closer to the observed evaporation values only in the last two years. The annual mean monthly peak (July) and leaf fall were well captured by all models. Here the variance ratio reaches the closest to the optimum values in comparison to all the other sites. Only for the summer months, a rather small difference of about 10 mm per month between the meteorological forces could be captured.

In the evergreen coniferous forest of Tharandt, daily evaporation usually yields 0-0.3 mm in winter and 2-3 mm in summer (with maximum around 7 mm). All setups except CBR90 demonstrated a high BIAS for the seasons (15-20 mm per month), which is larger in winter, where daily peaks are sometimes as high as summer maximums. Moreover, the inter-annual variability appears to be highly overestimated as well. Like for the grassland, the model calibration reduced the mean error to optimum values, but the problem of daily peaks in winter remained unsolved. In contrast to the other sites, a noticeable difference between forcings can be observed (up to 10 % in the summer months) with the in-situ measurements delivering the highest evaporation amount.

The evergreen coniferous forest of Oberbaerenburg normally has evaporation rates of 0-0.3 mm in winter and 2-3 mm in summer (with maximum around 8 mm). Evaporation here is 5-10% higher in the growing season than at the Tharandt site. Still, most of the setups (except in spring and CBR90) showed a positive BIAS, which is higher in winter and July. Similar to Tharandt, winter daily peaks sometimes exceeded summer extremes. Here, even the calibrated model did not demonstrate a good agreement in general and did not remove winter overestimations. Oberbaerenburg was the only site where the well-known European drought of 2018 is clearly visible on a monthly scale. The data shows around 30 % less evaporation in summer months due to depletion of the soil water and overall precipitation deficit. However, most of the model setups did not depict this effect properly. Finally, the spread between meteorological datasets here is not as broad as for the Tharandt site.

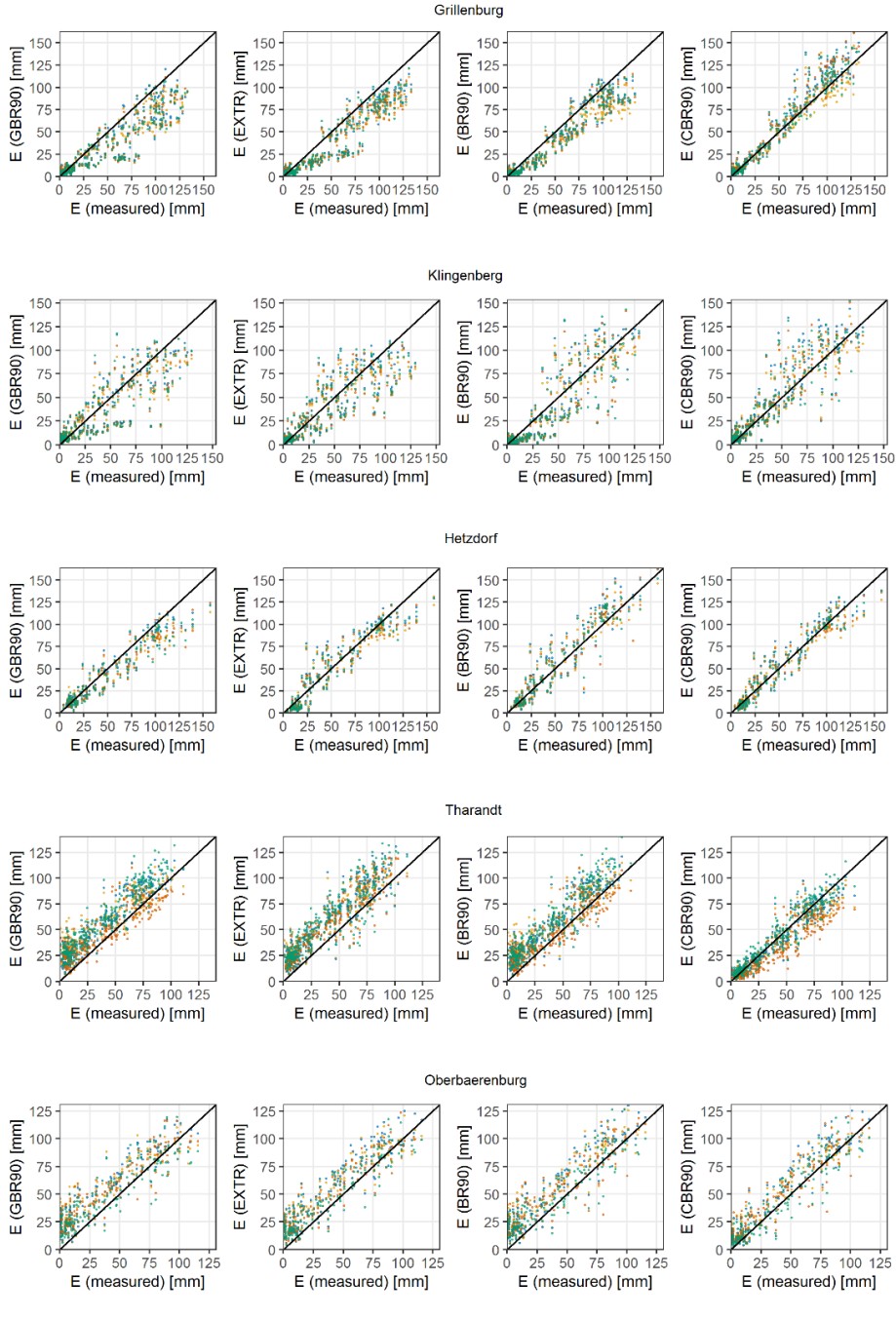

**Figure 3. Observed and modelled monthly evaporation values for all setups.**

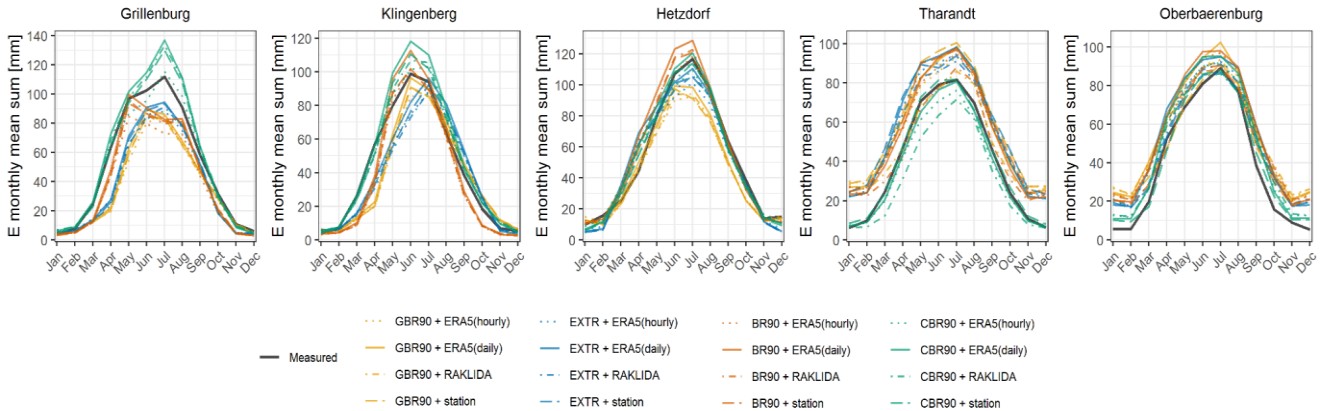

**Figure 4. Observed and modelled monthly mean evaporation values for all setups.**

In Fig. 5, the daily KGE values are shown, while the monthly results and other criteria (NSE, MAE) are presented in Appendix E. Based on KGE values, a good agreement was found between all model setups and observations for all the sites (Fig. 5). The best agreement showed the combination "CBR90 + station data" (from 0.72 in Oberbaerenburg to 0.91 in Grillenburg) and the worst "GBR90 + hourly ERA5" (from 0.36 in Grillenburg to 0.71 in Hetzdorf). On the monthly scale, all setups demonstrated higher performance, which is approximately 5 % better than on the daily scale. The goodness-of-fit in the vegetation period was better and very similar to the whole year, while the performance in winter time for all setups was lower, resulting sometimes in negative KGE values (down to -0.6). Here BR90 and EXTR showed poor agreement with the observations in the fields (Grillenburg and Klingenberg) and in the deciduous forest (Hetzdorf) respectively.

With a few exceptions, the best performance among the meteorological datasets was achieved for the station data and daily ERA5. On average for all the five sites, in terms of KGE values, the spreads in the meteorological forcings yielded 0.09 (maximum of 0.17 showed BR90 for Grillenburg), while scattering in the parameterization schemes was much higher and yielded 0.25 (with the maximum of 0.54 for Grillenburg and in-situ meteo data).

Finally, KGE spreads calculated for each combination from a resampled time-series are generally small. On the annual scale and for the vegetation period, higher uncertainties of obtained KGE values were found in Grillenburg, Klingenberg and Hetzdorf (10-15 % on average); while in Tharandt and Oberbaerenburg KGE deviations were low (around 5 %). For the winter months, the spread possessed the same behaviour, but resulted in much higher values (up to 100%). Among all the frameworks, GBR90 was associated with the largest uncertainty on the annual scale in almost all the cases, while it had the smallest spread in the winter, where uncertainty of EXTR and BR90 dominated.

NSE values are in general similar to KGE, but slightly smaller, which range from -0.05 for GBR90 in Grillenburg and Oberbaerenburg to 0.88 for CBR90 with station data. Mean average errors vary from 0.39 up to 0.98 mm*day$^{-1}$ with the highest values in evergreen forests for GBR90 and the lowest in Grillenburg for CBR90.

The hourly-resolved ERA5 data did not produce better results, showing the worst performance on the annual scale in most
cases.

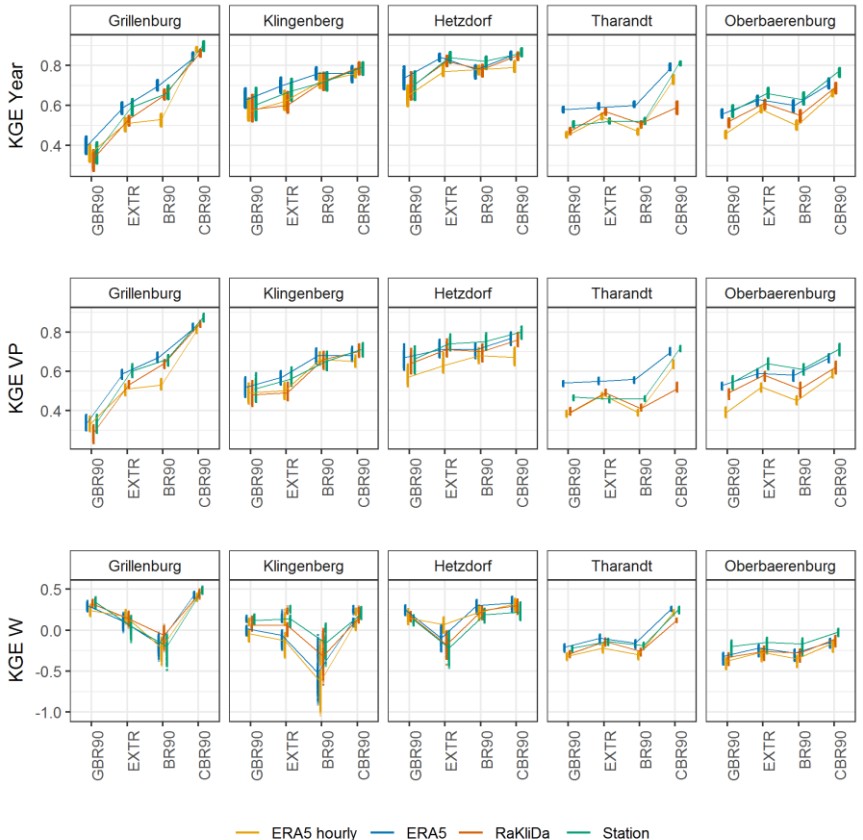

**Figure 5. KGE values for daily evaporation: whole year, vegetation and winter periods. Vertical lines for each cross-combination**
**refer to bootstrapped KGEs.**

The major advantage of the KGE criteria is the possibility to obtain a deeper understanding of model performance through its
decomposition. A closer look at the KGE components (Fig. 6) reveals that correlation coefficients for the fields (Grillenburg
and Klingenberg) and deciduous forest (Hetzdorf) are relatively high for all model setups (0.75-0.95), and the main problems
occur in underestimation of the mean (0.7-0.8) and variability ratios (0.55-0.7) (except for BR90 in Hetzdorf). In general, there
are only small fluctuations between model forcings for these three sites. In evergreen forests, on the other hand, the correlation
showed much higher spread among both parameterizations and meteorological datasets (0.4-0.75). Furthermore, BIAS and
variability ratios possess, on the other side, significant positive deviations from the optimal values (except variability in
Oberbaerenburg), especially in Tharandt (up to 1.6). Overall, ERA5 and station data perform better than others in most of the
cases. The hourly ERA5 forcing did not show a noticeable difference in evaporation BIAS or variability, but reduced

correlation in the forests (by 5-15 %). Finally, it could be noticed that in comparison to the other setups, CBR90 bring BIAS

and variance ratio almost to one, but did not improve correlation for all the sites (i.e. Hetzdorf).

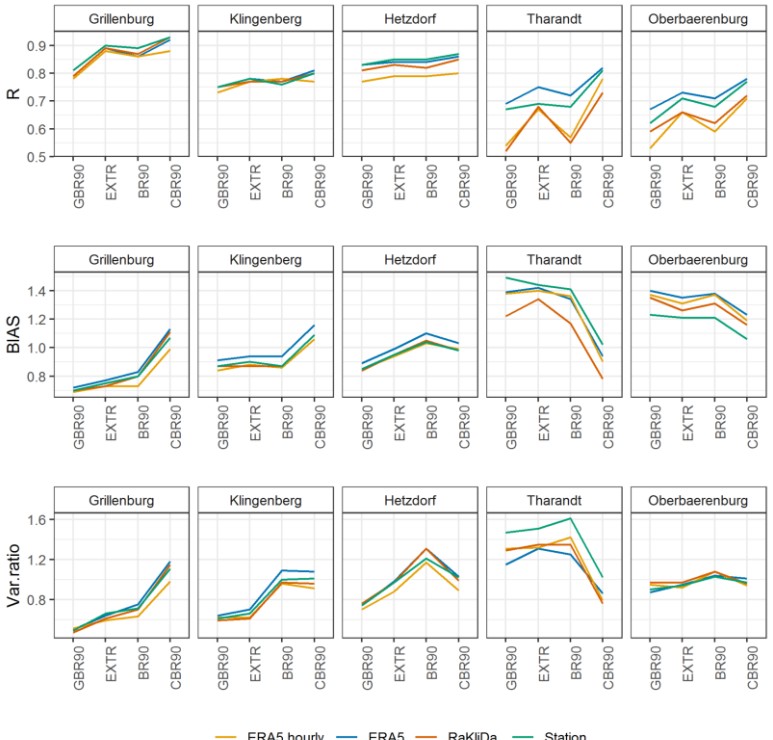

**Figure 6. Decomposition of KGE for daily evaporation for the whole year: correlation, BIAS and variance ratio**

**3.2 Evaporation components**

The 40-60 % partitioning between total flow and evaporation components in global terrestrial water balance (Müller Schmied

et al., 2016) also applies to the BROOK90 point simulations. With a variation in mean annual precipitation from 877 mm

(Klingenberg) to 1141 mm (Oberbaerenburg), measured mean annual evaporation varies from 476 mm (Tharandt) up to 625

(Hetzdorf) mm. This leads to measured E-P ratios of 0.41 to 0.65, with the lowest values observed in old spruce forest and the

highest in grassland and growing deciduous forest. Here, both the global and regional frameworks showed an overestimation

of the ratio for the evergreen forests (Tharandt and Oberbaerenburg) and an underestimation for the fields (Grillenburg and

Klingenberg) (could be found in Supplementary).

Summarized annual evaporation components (averaged from all tested model setups) are presented on Fig. 7. According to

this figure, transpiration in fields and deciduous forest yields 68-73 %, and evergreen forest transpires about 58-59 %. In

Tharandt and Oberbaerenburg 31-35 % of precipitation goes to interception (mainly rain, interception of snow is less than 2

%). In Grillenburg, Klingenberg and Hetzdorf evaporation of the intercepted precipitation is lower and yields 14-23 %. Soil evaporation on the other side, is higher in the fields (11-15 %) and lower in forests (4-8 %). Evaporation from snow is less than 2 % at all sites. The vegetation period spans 8 months in total and accounts for most of the annual evaporation (85-95 %). Thus, the distribution of components is generally consistent with a slightly higher contribution from transpiration. In winter, evaporation consists mainly of interception in forests and soil or snow evaporation of the fields.

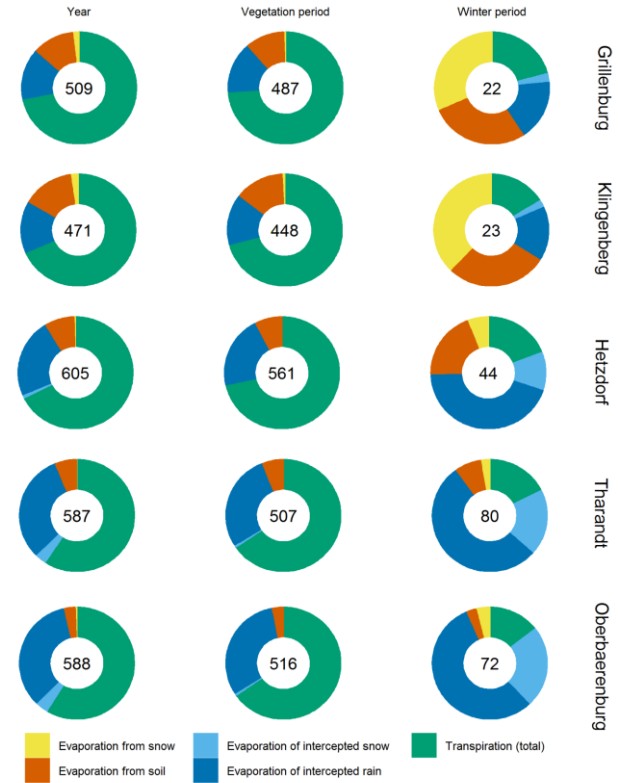

**Figure 7. Mean annual and seasonal evaporation components averaged over all model setups. The numbers inside pie charts refer to the mean evaporation sums per year or season.**

To get more insights on the possible setups' differences regarding the evaporation partitioning, we show "natural" model parameterization and forcing combinations (Fig. 8). Only minor differences were observed in evergreen coniferous forests. This mainly concerns intercepted rain. GBR90 with hourly ERA5 shows the largest amount (40-68 %) and CBR90 with station data reduces interception up to 15-30 %, which is especially noticeable in Oberbaerenburg. At the other three sites, seasonality plays a bigger role in the redistribution of evaporation components. Indeed, in the fields, almost no interception was modelled in EXTR using RaKliDa and BR90 with station data in winter and early spring, and all evaporation in these months consists of snow and soil evaporation. Furthermore, the transpiration is dominant in summer and autumn times with sharper edges due to crop and grass cutting. In general, EXTR delivers more soil evaporation than other model setups, while GBR90 produces

more rain interception. Slightly smoothened but similar results could be observed in the deciduous forest of Hetzdorf. Since
the actual distribution of the components is unknown, we can only assume that CBR with in-situ meteorological data indicates
conditions that are the closest to reality. Considering this, we can rank the goodness of the framework in the evaporation
representation in the following order (best to worst by similarity to CBR90): BR90, EXTR, GBR90, which seems indeed
logical.

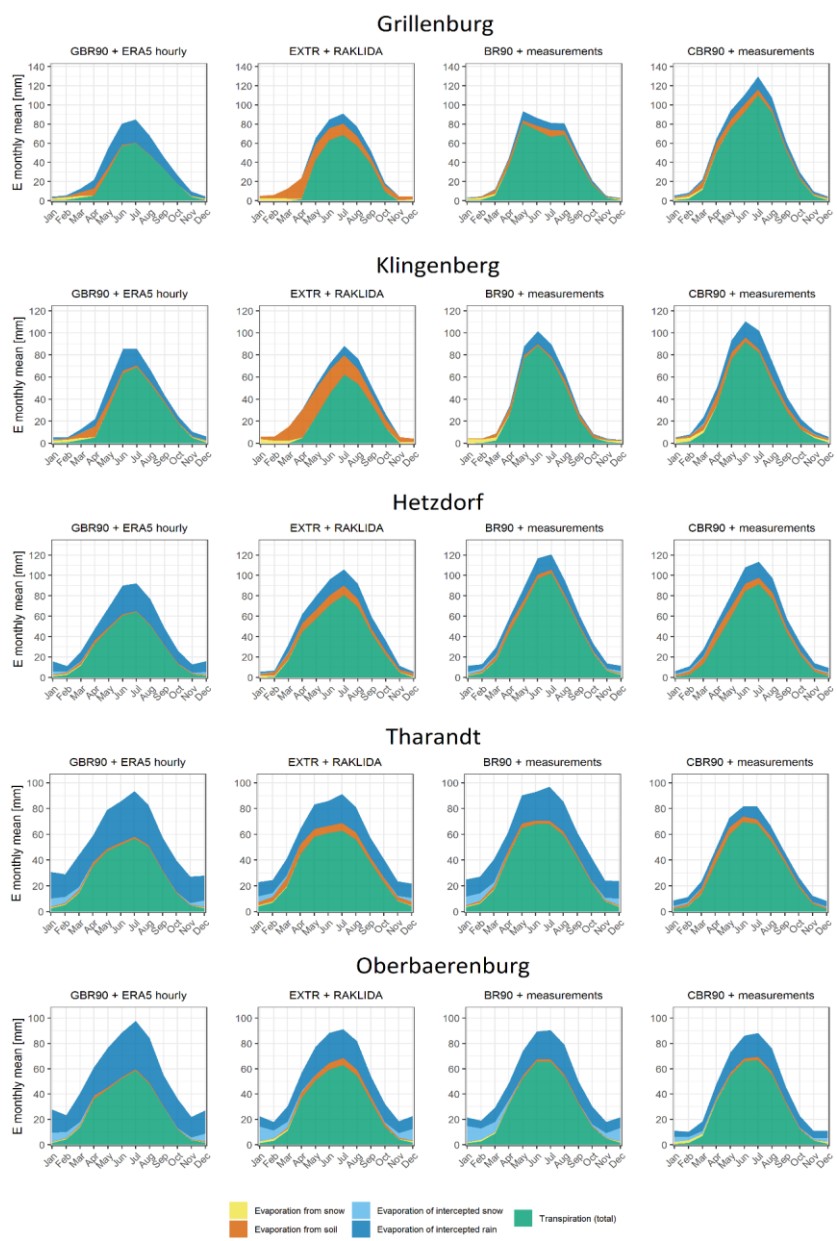

**Figure 8. Modelled mean monthly evaporation components.**

**3.3 Grass-reference evaporation: comparison of BROOK90 and FAO model with measurements**

The results of the FAO "potential" and BROOK90 "actual" grass-reference evaporation output are presented in Fig. 9. To simulate a BROOK90-based grass-reference evaporation, the original site-specific vegetation parameters were replaced by "grassland" parameters assumed at the Grillenburg site in the model. The meteorological input data remained site specific for both approaches.

The FAO estimations of the field sites Grillenburg and Klingenberg showed a good fit with the observed data. Deviations are observed as a time lag of one month in autumn and minor overestimations of evaporation in winter time (5-10 mm per month). BROOK90 simulations possess a noticeable time lag of 2-3 weeks in the spring period. Also an up to 20 % underestimation of evaporation in spring and summer months is visible.

Minor variances of around 10 mm per month between FAO and measured evaporation are observed in the deciduous forest of Hetzdorf. Namely there is a small overestimation in the spring period and an underestimation in summer months. The "actual" grass-reference evaporation from BROOK90, on the other hand, was mainly lower than the eddy-covariance measurements for all months, except for April and May.

In evergreen forests the FAO approach depicted considerably higher potential grass-reference evaporation than it was observed throughout the whole year. These high evaporation estimates of up to 30-40 % (July) are very high in summer months. BROOK90 did not show such high systematic deviations from the observations in Tharadt except for a peak in May. While in Oberbaerenburg the simulated evaporation was systematically lower for all months and especially in summer time.

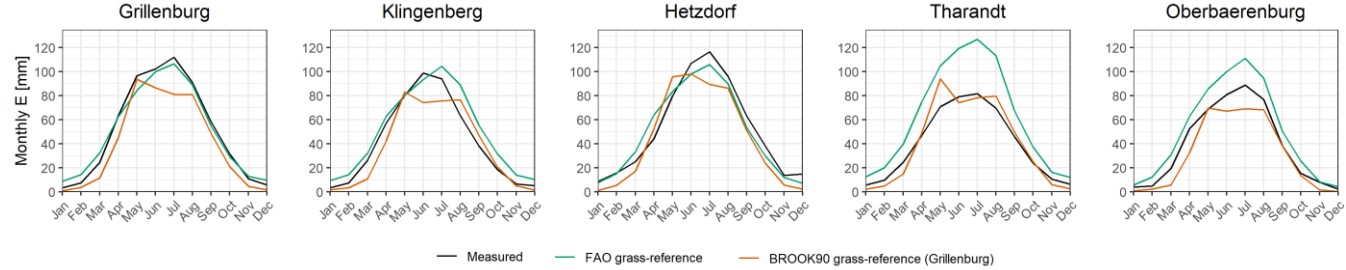

**Figure 9. Observed and modelled monthly mean grass-reference evaporation.**

**4 Discussion**

**4.1 Role of the framework's spatial scale in parameterization and forcing**

The comparison of GBR, EXTR and BR90 frameworks showed how sensible BROOK90 is to the spatial scale of the setup with regard to evaporation. Moreover, the fact that CBR90 showed significantly higher performance skill scores than the other

setups for almost all the sites, confirms indirectly that the BROOK90 is more sensible to the scale of parameterization scheme rather than to the scale of meteorological forcing. However, these conclusions need to be backed up with the assumption that both meteorological data and parameters used for each spatial scale come from state-of-the-art sources. Thus, they are both representative and possess the best quality (currently) for global, regional and local scales respectively.

The analysis of the parameters used in the study and their ranges revealed which groups of them possess the most noticeable influence on the accuracy of evaporation simulations and are at the same time affected by the scale of the model setup (Appendix C, Table C1).

At first, the plant leave's parameters must be highlighted, namely albedo, LAI and height, interception storages. Surface reflectivity with and without snow regulates the net radiation and thus directly affects potential evaporation. The values generally have a wide range 0.1-0.3 for vegetation and 0.2-0.9 for snow and their estimations are subject of high uncertainties (Alessandri et al., 2020; Myhre and Myhre, 2003; Page, 2003; Park and Park, 2016; Wang et al., 2017). For GBR90 and EXTR respectively, albedo was assigned by values taken from global and regional studies, while for BR90 measured values were used. Maximum LAI and its seasonal cycle are probably the most sensible and uncertain parameters in the model regardless of the vegetation type, while plant height and its seasonality plays a greater role and is more uncertain for the short, rather than in tall canopies. These two parameters often control the largest portion of potential evaporation controlling transpiration and interception as well as its partitioning (Hoek van Dijke et al., 2020; Wegehenkel and Gerke, 2013; Yan et al., 2012). On the global scale both parameters are derived by remote sensing estimates, while on the regional and local scale fixed values from regional studies and expert knowledge were taken. Therefore, at these scales the simulations apparently showed better results for the case-study. The interception storage and intercepted precipitation fraction are the key parameters for the correct estimation of interception amount (Wu et al., 2019). They are all plant-, season- and age-dependent, and possess a high variability, which makes its very challenging to generalize their values for the vegetation classes (Federer and Douglas, 1983; Leaf and Brink, 1973; Pypker et al., 2005; Yang et al., 2019). In all frameworks they are set up as default or with expert knowledge. Nevertheless, only due to these parameters, the interception uncertainty could be as high as ± 20 mm per month, especially in forests.

The second group denotes soil parameters. The soil structure, profile depth and coarse fragment's fraction directly determine the maximum water storage capacity for a site. Here, the parameter scale plays a crucial role, since, the quality of available datasets decreases drastically from a local to a global scale due to scarcity of soil profile data and very high heterogeneity of soils (Hengl et al., 2017). Soil hydraulic properties certainly have a big influence on the water retention and holding capacity, controlling water supply for the actual soil evaporation and transpiration (Carminati and Javaux, 2020; Lehmann et al., 2018; Verhoef and Egea, 2014). However, the scale uncertainty due to this parameter group is difficult to assess, since these parameters are assigned indirectly based on sand, silt and clay content for each layer and fixed parameter set. Thus, the problem is narrowed to correct identification of the soil texture, which is still a very challenging task even for a regional scale (Hengl et al., 2017).

Significant difference in the model performance due to different meteorological input datasets was not evident for all setups
and sites (bootstrapped values on Fig. 5). Here, the spatial scale did not follow the main hypothesis, as the global dataset ERA5
was not the worst and in many cases outperformed in-situ meteorological data. It would appear that the RaKliDa dataset with
its 1 km spatial resolution could fit the eddy-covariance footprint at least as good as station data, however, it sometimes
demonstrated the worst performance or close to hourly ERA5. This outcome contradicts with the generally accepted
application of regional meteorological forcings to simulate evaporation in high resolution (Martens et al., 2018; Rudd and Kay,
2015; Wang et al., 2015; Zink et al., 2017). However, probably due to location peculiarities of the study sites, and good
agreement of the global reanalysis with station data, regional dataset did not show a competitive performance. Namely, ERA5
showed slightly better precipitation BIAS values, than RaKliDa (Table 3). Moreover, RaKliDa exhibits a systematic
underestimation of the global radiation, especially in the summer months (Appendix A).

**4.2 Challenges in the model process representation**

Although BROOK90 has a decent physically-based representation of the evaporation process, it shows some limitations as
well. At first, BROOK90 treats the vegetation as a single layer (big-leaf). Thus, the complexity of canopy vertical structure is
omitted, which can be insignificant for simple ecosystems like meadows or cropland, but might play a big role in multi-layered
vegetation (Bonan et al., 2021; Luo et al., 2018; Raupach and Finnigan, 1988). For example, the lack of undergrowth
representation could have an effect on the evaporation underestimation in forests with a dense floor like Hetzdorf. Additionally,
there is no allowance for non-green leaves, which intercept precipitation and radiation, but in the meantime do not transpire.
This process can play a role in deciduous forests like Hetzdorf in autumn and winter, as they generate too much transpiration.
Furthermore, since the phenomenon of ground frost is not considered, soil evaporation is not limited on these days, which
could lead to an overestimation in winter. As canopy parameters are assumed constants, phenology or growth (e.g. crop rotation
in Klingenberg and continuous forest growth in Hetzdorf) as well as drought affecting LAI (reduction due to prolonged water
stress) are not considered in the model. Snowpack energy and evaporation modules suffer from overestimations in tall canopies,
thus an arbitrary reduction factor is applied. Finally, albedo does not depend on solar elevation angle, canopy structure, or
snow age. These limitations alone could have a substantial influence on total evaporation and its timing.

In addition, the PM equation uses vapour pressure deficit and net energy as the main factors to calculate potential evaporation.
The first variable is derived directly from the daily input temperature and available vapour pressure using the Magnus equation
and does not vary much between different methods (Lide, 2005). For net energy, the situation is different. The shortwave
radiation is an input and its net value is controlled by the rather vague albedo, while the longwave radiation is estimated
internally using the effective emissivity of the clear sky. Under these assumptions, the potential discrepancy between different
formulas can be as high as 20-30 $W*m^{-2}$. After obtaining a persistent positive BIAS for evaporation in the forests, we checked
the energy balance of the model with in-situ measurements (Fig. 10). In fact, minor differences were found for all input
datasets. In the summer period, minor overestimation was found for ERA5 and station data in Grillenburg, Klingenberg and

Tharandt, and underestimates for RaKliDa in Hetzdorf and Tharandt. In winter (especially in December and January), large
relative underestimation was discovered in Grillenburg, Hetzdorf and Oberbaerenburg. Therefore, with a negative amount of
energy, BROOK90 still showed higher monthly evaporation than measured. Specifically, according to Fig. 8, 90 % of the
actual evaporation in forests in winter consists of interception, and normally there is no absence of precipitation input during
this period. Because of the peculiarities of the PM approach, positive potential evaporation can be estimated with negative net
energy, positive vapour pressure deficit, and low estimated atmospheric and canopy resistances. Thus, as long as vapour
pressure deficit exists, the evaporation flux tries to fill the gradient.

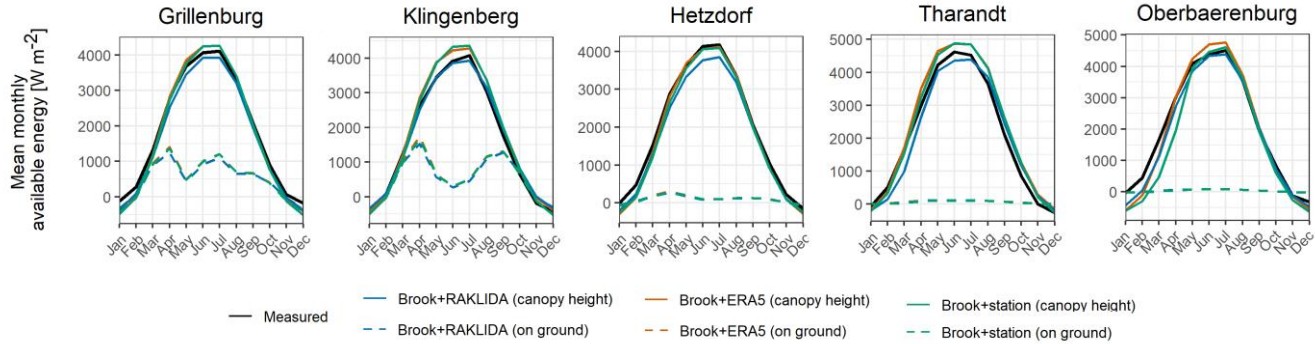

**Figure 10. Observed and modelled monthly mean net energy on canopy and ground level.**

Finally, as it was found, the hourly-resolved input precipitation data did not produce better results, showing the worst
performance (hourly ERA5 data) on the annual scale in most cases. This brings up the question of reliability of the subdaily
calculations in BROOK90 interception module, which omits i.e. diurnal cycle of potential evaporation and consistently
produces too much interception if hourly input is used (Federer, 2002). However, it could also be the quality of subdaily
precipitation distribution in the ERA5 data for the study region, since on daily, monthly and annual scales ERA5 did not show
a significant difference with the station data, which could account for that high differences in daily and hourly performance.

**4.3 Reliability of eddy-covariance measurements**

Reliability of the evaporation measurements with eddy-covariance techniques themselves is a widely discussed question.
Standard methods of the "energy-balance-closure" corrections (Wilson et al., 2002; Richardson et al., 2012) does not always
lead to necessary BIAS adjustment (Foken, 2008; Imukova et al., 2016). Therefore, largest systematic deviations between
observed and modelled evaporation, which could be discussed in the context of inaccuracy of the measurements, were
discovered in the evergreen forests in winter, in grassland in summer and in pasture in growing season. Analysis of the
evaporation components and comparison of the FAO with the BROOK90 grass-reference evaporation helped to reveal some
discrepancies in the eddy-covariance measurements.

The time lag during the growing and harvesting periods for Klingenberg could be explained with permanent crop rotation and

inability of FAO and BROOK90 models to cope with non-stationarity in vegetation parameters. Overestimation in winter for

the FAO method for both sites could be a result of simplifications of FAO-modified PM equation against SW approach in

BROOK90 (i.e. neglecting the soil water holding capacity). According to the continuous long-term measurements of grass

height in Grillenburg, regular grass cutting is performed in June-July. This in general should lead to evaporation decline, which

can be seen clearly on Fig. 4 for monthly evaporation of BR90. However, this effect was not found in the measurements (even

on a daily scale). Moreover, mean evaporation usually shows maximum annual values in July. Besides possible systematic

measurement errors, this could be explained either by an underestimation of the real site footprint. Another explanation is near-

saturation conditions of the soils. Thus, almost unlimited water supply and perturbation of the evaporation components after

grass cutting (drastic increase of soil evaporation). Nevertheless, while calibrating the model, it was realized that it is

impossible to increase soil evaporation by almost 30 mm during the summer months and stay within the physically meaningful

boundaries for soil parameters for the given soil profile. The findings are consistent with other studies, where latent heat fluxes

were systematically over- and underestimated depending on season in in short canopies (Moorhead et al., 2019; Perez-Priego

et al., 2017; Twine et al., 2000).

In Tharandt and Oberbaerenburg FAO approach showed 10-20 mm evaporation in the winter months, while BROOK90

resulted in 3-5 mm (consisting only of soil and snow evaporation). At the same time, all model setups showed 20-30 mm of

evaporation per month in winter (which is more than 80 % consists of intercepted precipitation), while only 5-10 mm is

observed. Thus, it is possible that the interception is generally underestimated by eddy-covariance measurements in the forests.

Moreover, while the calibration in Tharandt helped to adjust the simulated evaporation in winter months as well (primarily by

increasing the winter albedo), in Oberbaerenburg even a relatively wide parameters' range was not sufficient. Here, the large

variations between two approaches emphasize the importance of the soil and in a regulation of the evaporation, since different

soil types appear at the grassland and evergreen forest sites (gleysols and podzols respectively). As few researchers pointed

out, the reliability of eddy-covariance data within the rainy days and when the interception dominates is indeed questionable

(Dijk et al., 2015; Wilson et al., 2001).

In addition, previous analysis of eddy-covariance data for some of the study sites showed, that the possible under and

overestimations in measurements could be as large as ± 8-11 % for Tharandt, ± 29-36 % for Grillenburg and ± 28-44 % for

Klingenberg (Spank et al., 2013).

Therefore, in addition to reliability of the mean net energy and precipitation (Sect. 2.4 and 4.2), it is possible that the quality

of the eddy-covariance data is questionable due to at least systematic underestimation of interception and non-representative

footprint.

**Conclusion and outlook**

This study presents the qualitative analysis and discussion of the BROOK90 model scale uncertainties with regard to evaporation simulations. We tried to answer the question how the model setup scale influences the performance and whether the model is more sensitive to the parameter set or to the meteorological input. For this, three frameworks (Global BROOK90, EXTRUSO and BROOK90 with manual parameterization) and three forcing datasets (ERA5, RaKliDa, in-situ measurements) were used, representing the global, regional and local scale, respectively. We made cross-combinations of them and model evaporation components for five locations in Saxony, Germany, covered by long-term eddy-covariance measurements: grassland (Grillenburg), cropland (Klingenberg), deciduous broadleaf forest (Hetzdorf) and two evergreen needleleaf forests (Tharandt, Oberbaerenburg).

Our results indicated that all setups perform well even on a daily scale, with KGE values ranging from 0.35-0.80. KGE decomposition demonstrated that with high correlation coefficients in grassland, cropland and deciduous forest performance was affected here mainly by BIAS and variance ratios, whereas in evergreen forest all three components varied greatly. The highest and lowest values among all setups were achieved by the same combination of Global BROOK90 and ERA5 in Hetzdorf and Grillenburg respectively. Calibration of the model increased KGE significantly, especially for Grillenburg and Tharandt. In the vegetation period when 90-95 % of the total annual evaporation was observed, the agreement with the observations was much higher than in the winter period.

The main finding of the study is that the spread in model performances is four times higher due to the parameter datasets compared to the meteorological forcings based on the tested setups. Furthermore, while the spread of model performances due to parameter sets mattered throughout the year, the spread due to the meteorological datasets was evident only in summer months. The breakdown of evaporation components revealed that in the vegetation period transpiration yields up to 65-75 % of total evaporation, while in the winter months' interception (in forests) and soil/snow evaporation (in fields) play a major role. Moreover, the studied parameter sets showed substantial differences in the redistribution of evaporation components. Finally, the results raised questions about meteorological data quality, limitations of the model and the reliability of the eddy-covariance measurements as evaporation benchmark data. Finally our results suggested that the ERA5 dataset works as a meteorological forcing of choice even for a local scale.

In the outlook, we would like to suggest possible future directions on this topic:

- expand the number of study sites with other FLUXNET towers
- run similar analysis for other physically-based models
- analyse model uncertainty by incorporating runoff and soil moisture in the analysis
- apply and validate different methods to breakdown eddy-covariance data in components

**Appendix A. Comparison of BROOK90 meteorological input data (ERA5, RaKliDa and station measurements)**

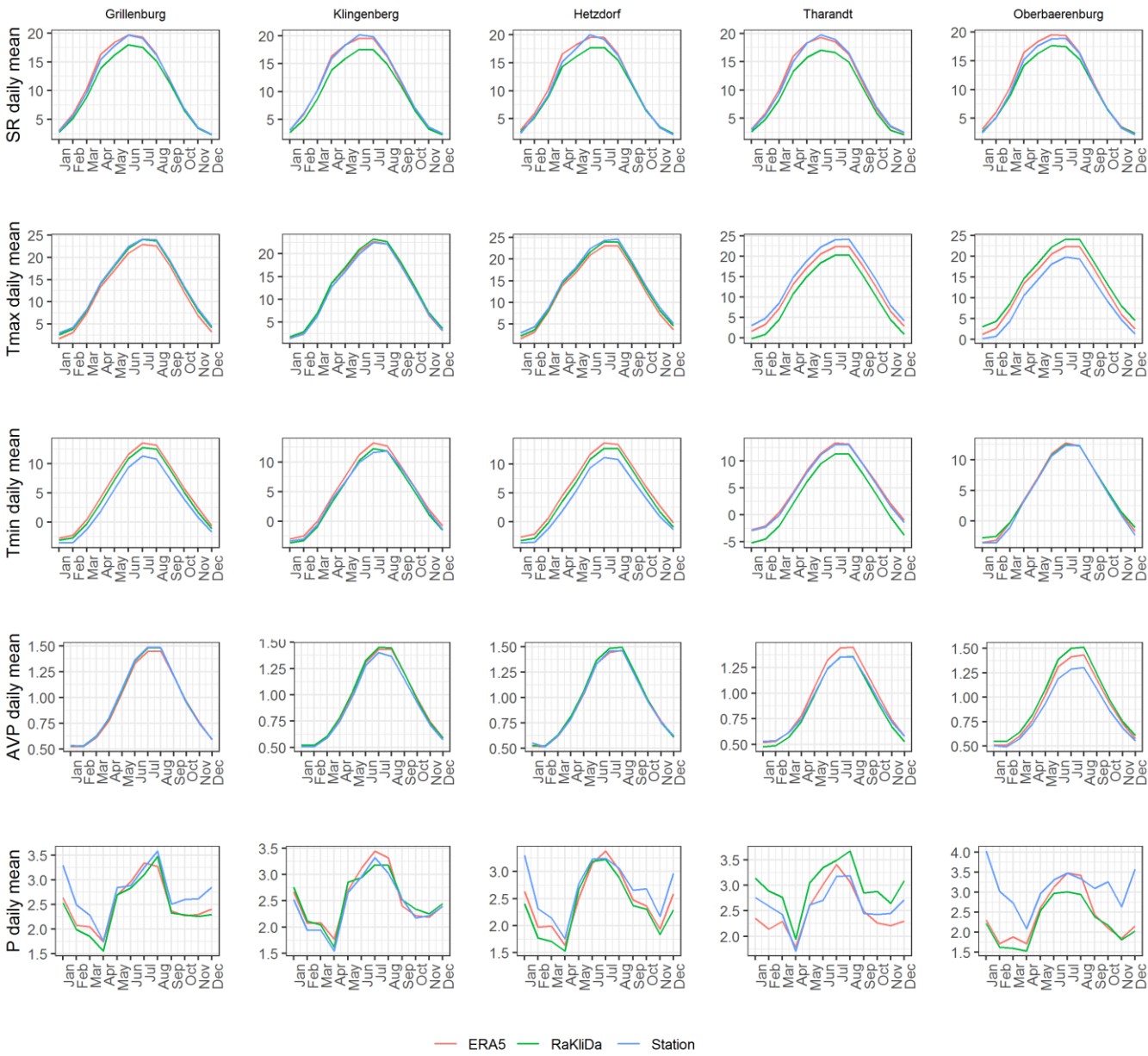

**Figure A1 Monthly daily mean meteorological variables**

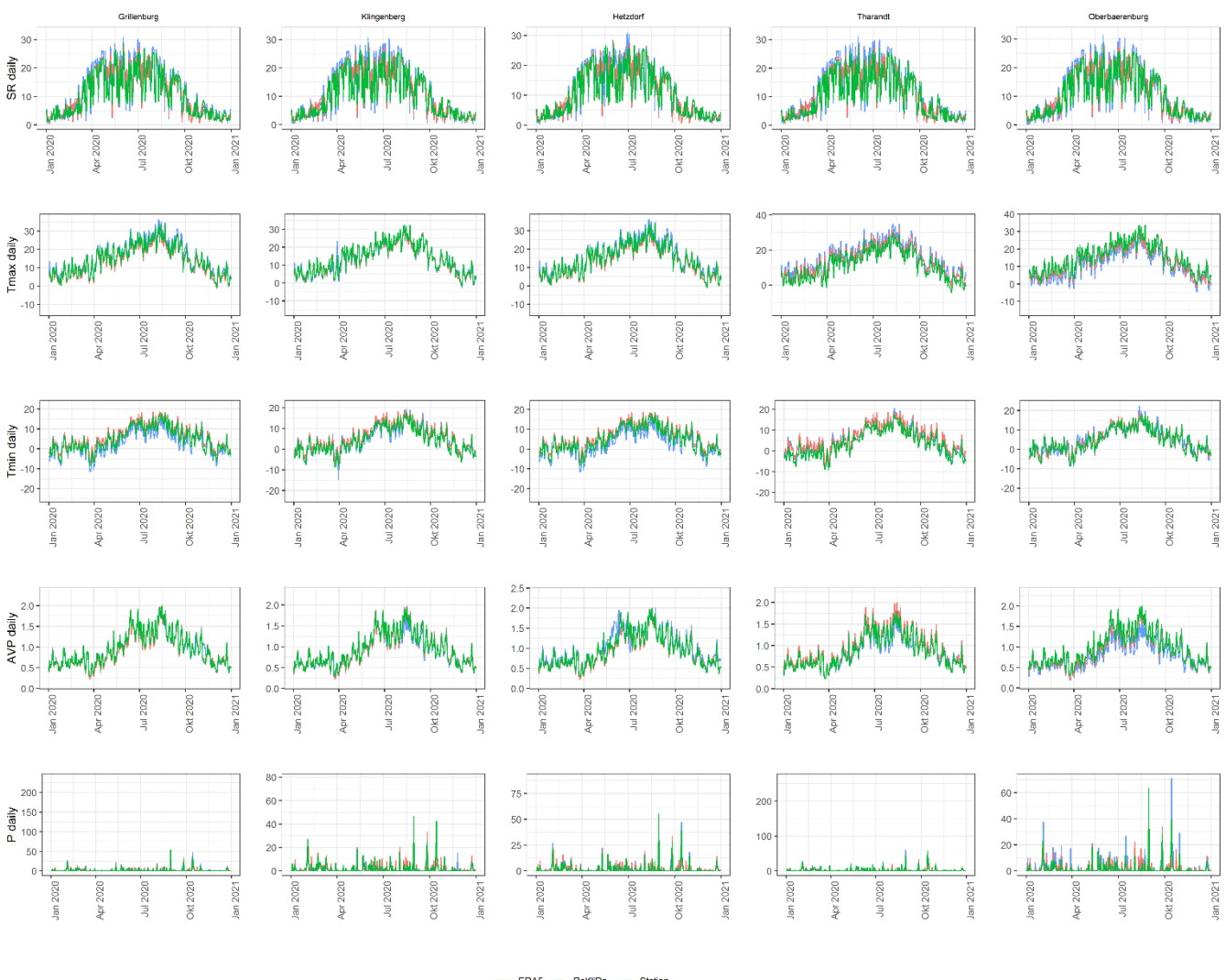

**Figure A2 Daily values of meteorological variables for 2020**

 **Appendix B. Skill-scores**

| Name | Range | Optimum value | Formula |
|---|---|---|---|
| Mean Absolute Error (MAE) | $[0, +\infty]$, | 0 | $$MAE = \frac{\sum_{t=1}^{T} |E_m^t - E_o^t|}{T}$$ where $E_m^t$ and $E_o^t$ are the modelled and observed evaporation values (in mm) at time $t$, and $T$ is the overall length of time-series |
| Nash-Sutcliffe Efficiency (NSE) (Nash and Sutcliffe, 1970) | $[-\infty, 1]$ | 1 | $$NSE = 1 - \frac{\sum_{t=1}^{T}(E_m^t - E_o^t)^2}{\sum_{t=1}^{T}(E_o^t - \overline{E_o})^2}$$ where $E_m^t$ and $E_o^t$ are the modelled and observed evaporation values (in mm) at time $t$, and $T$ is the overall length of time-series |
| Kling-Gupta Efficiency (KGE) (Gupta et al., 2009) | $[-\infty, 1]$ | 1 | $$KGE = 1 - \sqrt{(r-1)^2 + (\alpha-1)^2 + (\beta-1)^2}$$ where $r$ is the Pearson correlation coefficient between the modelled and observed evaporation, $\alpha$ is the ratio between the simulated and observed evaporation variability, $\beta$ is the ratio between the mean simulated and mean observed evaporation: |
| | $[-1, 1]$ | 1 | $$r = \frac{cov(E_m, E_o)}{\sigma_m \sigma_o} = \frac{\sum_{t=1}^{T}(E_m^t - \overline{E_m})(E_o^t - \overline{E_o})}{\sqrt{\sum_{t=1}^{T}(E_m^t - \overline{E_m})^2 \cdot \sum_{t=1}^{T}(E_o^t - \overline{E_o})^2}}$$ |
| | $[-\infty, +\infty,]$ | 1 | $$\alpha = \frac{\sqrt{\sum_{t=1}^{T}(E_m^t - \overline{E_m})^2}}{\sqrt{\sum_{t=1}^{T}(E_o^t - \overline{E_o})^2}}$$ |
| | $[-\infty, +\infty,]$ | 1 | $$\beta = \frac{\overline{E_m}}{\overline{E_o}}$$ |

**Appendix C. BROOK90 main parameters and calibration results**

Table C1 Main site-specific parameters (topography, coil and land cover related) used in tested BROOK90 frameworks*.

Grillenburg

| Parameter abbreviation | Physical meaning | Unit | GBR90 | EXTR | BR90 | CBR90 |
|---|---|---|---|---|---|---|
| ALB | albedo or surface reflectivity without snow | - | 0.2 | | 0.18 | 0.24 |
| ALBSN | albedo or surface reflectivity with snow | - | 0.45 | 0.5 | | 0.44 |
| ASPECTD | aspect, degrees through east from north | degrees | 180 | 0 | 251 | |
| BEXP | exponent for ψ-θ relation | - | 5.39 | 5.3 | | |
| CINTRL | maximum interception storage of rain per unit LAI | mm | 0.15 | 0.06 | 0.2 | 0.10 |
| CINTRS | maximum interception storage of rain per unit SAI | mm | 0.15 | 0.06 | 0.2 | 0.2 |
| CINTSL | maximum interception storage of snow per unit LAI | mm | 0.6 | | | 0.78 |
| CINTSS | maximum interception storage of snow per unit SAI | mm | 0.6 | | | |
| CR | extinction coefficient for photosynthetically-active radiation in the canopy | - | 0.7 | 0.5 | 0.7 | 0.8 |
| CVPD | vapor pressure deficit at which stomatal conductance is halved | kPa | 2 | | | 1.8 |
| CS | ratio of projected SAI to HEIGHT | - | 0.035 | | 0.1 | |
| ESLOPED | slope for evapotranspiration and snowmelt | degrees | 0 | | 1 | |
| FRINTL | intercepted fraction of rain per unit LAI | - | 0.06 | 0.15 | 0.06 | 0.08 |
| FRINTS | intercepted fraction of rain per unit SAI | - | 0.06 | 0.15 | 0.06 | |
| FSINTL | intercepted fraction of snow per unit LAI | - | 0.04 | | | |
| FSINTS | intercepted fraction of snow per unit SAI | - | 0.04 | | | |
| FXYLEM | fraction of plant resistance that is in the xylem | - | 0 | | | |
| GLMAXC | maximum leaf conductance | cm/s | 0.8 | 0.53 | 1.50 | 1.47 |
| GLMINC | minimum leaf conductance | cm/s | 0.03 | 0.01 | 0.03 | |
| IMPERV | fraction of the soil surface that is impermeable and always routes water reaching it directly to streamflow | - | 0.01 | 0 | | |
| KF | hydraulic conductivity at field capacity corresponding to THETAF and PSIF for a soil layer | mm/d | 6.3 | 13.1 | | |
| KSNVP | reduction factor between 0.05 and 1 to reduce snow evaporation | - | 1 | 0.3 | 1 | |
| LATD | latitude | degrees | 50.95 | | | |
| LWIDTH | average leaf width | m | 0.01 | 0.006 | 0.01 | 0.024 |
| MAXHT | maximum canopy height for the year | m | 0.5 | 0.8 | 0.80 | |

| | | | | | | |
|---|---|---|---|---|---|---|
| MAXLAI | maximum projected LAI for the year | m²/m² | 5.8 | 4 | 5 | 5.9 |
| MXKPL | maximum plant conductivity | mm day⁻¹ MPa⁻¹ | 8 | | | 7.3 |
| MXRTLN | maximum length of fine roots per unit ground area | m²/m² | 1000 | | 800 | 601 |
| NLAYER | number of soil layers to be used | - | 7 | 5 | | |
| PSICR | minimum plant leaf water potential | MPa | -2 | -2.5 | -2 | -1.9 |
| PSIF | matric potential at "field capacity" corresponding to KF and THETAF for a soil layer | kPa | -8.5 | -25 | | |
| RELHT | pairs of day of the year and relative height between 0 and 1 | - | 1,0.03,120,0.03, 210,1,330,0.03, 366,0.03 | 1,0.1,115,0.1, 145,1,268,1, 298,1,366,0.1 | 1,0.1,80,0.1, 105,0.3,130,0.4, 160,1,170,0.15, 220,0.46,270,0.25, 320,0.12,366,0.12 | 1,0.16,80,0.2, 105,0.6,130,0.57, 160,0.6,170,1, 220,0.9,270,0.37, 320,0.28,366,0.10 |
| RELLAI | pairs of day of the year and relative LAI between 0 and 1 | - | 1,0.087,41,0.101, 82,0.223,122,0.836, 163,1,203,0.983, 244,0.76,284,0.577, 325,0.279,366,0.087 | 1,0,115,0, 145,1,268,1, 298,0,366,0 | 1,0.05,80,0.05, 105,0.15,130,0.5, 160,1,170,0.2, 220,0.5,270,0.25, 320,0.05,366,0.05 | 1,0.12,80,0.17, 105,0.41,130,0.62, 160,1,170,0.60, 220,1,270,0.15, 320,0.15,366,0.06 |
| ROOTDEN | relative root density (per unit stonefree volume) of fine or absorbing roots for given layer | m³/m² | 100,0.44,100,0.25, 100,0.14,100,0.08, 100,0.04,100,0.02, 100,0.02,100,0.01, 100,0 | 100,0.44,100,0.25, 100,0.14,100,0.08, 100, 0.04,100,0.02,100,0.01 | 100,0.44,100,0.25,100,0.14,100,0.05,100,0 | |
| STONEF | stone volume fraction in each soil layer | - | 0.10, 0.10, 0.11, 0.11, 0.13, 0.17, 0.17 | 0.01 | | |
| THETAF | volumetric water content at "field capacity" corresponding to KF and PSIF for soil layer | m³/m² | 0.324 | 0.365 | | |
| THICK | layer thicknesses | mm | 25,75,125, 225, 350, 700,500 | 100,130,100, 500,500 | | |
| THSAT | THETA at saturation | m³/m² | 0.451 | 0.485 | | |
| WETINF | wetness at dry end of near-saturation range for a soil layer | - | 0.92 | | | |
| Z0G | ground surface roughness | m | 0.01 | 0.02 | | |

## Klingenberg

| Parameter abbreviation | Physical meaning | Unit | GBR90 | EXTR | BR90 | CBR90 |
|---|---|---|---|---|---|---|
| ALB | albedo or surface reflectivity without snow | - | 0.22 | | 0.18 | 0.13 |
| ALBSN | albedo or surface reflectivity with snow | - | 0.50 | | | 0.6 |
| ASPECTD | aspect, degrees through east from north | degrees | 225 | 0 | 213 | |
| BEXP | exponent for ψ-θ relation | - | 5.39 | 11.4,11.4,8.52,5.39 | 11.4,11.4,8.52,5.39 | |
| CINTRL | maximum interception storage of rain per unit LAI | mm | 0.15 | | 0.2 | 0.10 |

| Code | Description | Units | | | | |
|---|---|---|---|---|---|---|
| CINTRS | maximum interception storage of rain per unit SAI | mm | 0.15 | | 0.2 | |
| CINTSL | maximum interception storage of snow per unit LAI | mm | 0.6 | | | 0.8 |
| CINTSS | maximum interception storage of snow per unit SAI | mm | 0.6 | | | |
| CR | extinction coefficient for photosynthetically-active radiation in the canopy | - | 0.7 | 0.5 | 0.7 | 0.73 |
| CVPD | vapor pressure deficit at which stomatal conductance is halved | kPa | 2 | | | 0.5 |
| CS | ratio of projected SAI to HEIGHT | - | 0.035 | | 0.1 | |
| ESLOPED | slope for evapotranspiration and snowmelt | degrees | 5 | 0 | 1 | |
| FRINTL | intercepted fraction of rain per unit LAI | - | 0.06 | | | 0.1 |
| FRINTS | intercepted fraction of rain per unit SAI | - | 0.06 | | | |
| FSINTL | intercepted fraction of snow per unit LAI | - | 0.04 | | | 0.035 |
| FSINTS | intercepted fraction of snow per unit SAI | - | 0.04 | | | |
| FXYLEM | fraction of plant resistance that is in the xylem | - | 0 | | | |
| GLMAXC | maximum leaf conductance | cm/s | 1.1 | | 1.3 | 1.5 |
| GLMINC | minimum leaf conductance | cm/s | 0.03 | | 0.05 | |
| IMPERV | fraction of the soil surface that is impermeable and always routes water reaching it directly to streamflow | - | 0.01 | 0 | | |
| KF | hydraulic conductivity at field capacity corresponding to THETAF and PSIF for a soil layer | mm/d | 6.3 | 4.3,4.3,7.3,6.3 | 4.3,4.3,7.3,6.3 | |
| KSNVP | reduction factor between 0.05 and 1 to reduce snow evaporation | - | 1 | | | |
| LATD | latitude | degrees | 50.89 | | | |
| LWIDTH | average leaf width | m | 0.05 | 0.1 | 0.025 | 0.035 |
| MAXHT | maximum canopy height for the year | m | 1.3 | 2.2 | 1.4 | |
| MAXLAI | maximum projected LAI for the year | $m^2/m^2$ | 5.2 | 4.7 | 4 | 6 |
| MXKPL | maximum plant conductivity | mm day$^{-1}$ MPa$^{-1}$ | 8 | | | 7 |
| MXRTLN | maximum length of fine roots per unit ground area | $m^2/m^2$ | 110 | 110 | 500 | 374 |
| NLAYER | number of soil layers to be used | - | 7 | 4 | | |

| Parameter abbreviation | Physical meaning | Unit | | | | |
|---|---|---|---|---|---|---|
| PSICR | minimum plant leaf water potential | MPa | -2 | | | -2.1 |
| PSIF | matric potential at "field capacity" corresponding to KF and THETAF for a soil layer | kPa | -8.5 | -7.7,-7.7,-14.7,-8.5 | -7.7,-7.7,-14.7,-8.5 | |
| RELHT | pairs of day of the year and relative height between 0 and 1 | - | 1,0.03,120,0.03, 210,1,330,0.03, 366,0.03 | 1,0,100,0, 213,1,278,1 ,308,0,366,0 | 1,0.07,100,0.10, 130,0.57,160,1, 190,1,210,0.5, 240,0.29,270,0.07, 320,0.09,366,0.07 | 1,0.03,100,0.13, 130,0.52,160,1, 190,1,210,0.4, 240,0.32,270,0.1, 320,0.1,366,0.1 |
| RELLAI | pairs of day of the year and relative LAI between 0 and 1 | - | 1,0.286,41,0.054, 82,0.243,122,0571, 163,1,203,0.486, 244,0.318,284,0.3, 325,0.393,366,0.286 | 1,0,100,0, 213,1,278,1, 308,0,366,0 | 1,0.01,100,0.05, 130,0.57,160,0.9, 190,1,210,0.5, 240,0.29,270,0.05, 320,0.05,366,0.01 | 1,0.03,100,0.05, 130,0.6,160,0.6, 190,0.78,210,1, 240,0.9,270,0.68, 320,0.20,366,0.03 |
| ROOTDEN | relative root density (per unit stonefree volume) of fine or absorbing roots for given layer | m$^3$/m$^2$ | 100,0.34,100,0.22, 100,0.15,100,0.10, 100,0.07,100,0.04, 100,0.03,100,0.02, 100,0.01,100,0.01, 100,0.01,100,0.01, 100,0 | 100,0.34,100,0.22, 100,0.15,100,0.1, 100,0.07,100.0.04 | 100,0.4,100,0.3,100,0.15,100,0.1, 100,.1,100,0.05,100,0.05,100,0 | |
| STONEF | stone volume fraction in each soil layer | - | 0.15,0.15,0.15,0.16,0.17 0.21,0.23 | 0.11,0.11,0.11,0.11 | | |
| THETAF | volumetric water content at "field capacity" corresponding to KF and PSIF for soil layer | m$^3$/m$^2$ | 0.324 | 0.425,0.425,0.402,0.324 | | |
| THICK | layer thicknesses | mm | 25,75,125,225,350,700,500 | 200,300,200,100 | | |
| THSAT | THETA at saturation | m$^3$/m$^2$ | 0.451 | 0.482,0.482,0.476,0.451 | | |
| WETINF | wetness at dry end of near-saturation range for a soil layer | - | 0.92 | 0.94,0.94, 0.92,0.92 | | |
| Z0G | ground surface roughness | m | 0.005 | 0.02 | | |

Hetzdorf

| Parameter abbreviation | Physical meaning | Unit | GBR90 | EXTR | BR90 | CBR90 |
|---|---|---|---|---|---|---|
| ALB | albedo or surface reflectivity without snow | - | 0.18 | 0.21 | 0.21 | 0.10 |
| ALBSN | albedo or surface reflectivity with snow | - | 0.22 | 0.47 | 0.50 | 0.49 |
| ASPECTD | aspect, degrees through east from north | degrees | 315 | 0 | 148 | |
| BEXP | exponent for ψ-θ relation | - | 5.39 | 5.3,5.3,5.3,5.3,4.9 | | |
| CINTRL | maximum interception storage of rain per unit LAI | mm | 0.15 | 0.7 | 0.15 | 0.10 |
| CINTRS | maximum interception storage of rain per unit SAI | mm | 0.15 | 1 | 0.15 | |
| CINTSL | maximum interception storage of snow per unit LAI | mm | 0.6 | 2.8 | 0.6 | 0.10 |
| CINTSS | maximum interception storage of snow per unit SAI | mm | 0.6 | 4 | 0.6 | |
| CR | extinction coefficient for photosynthetically-active radiation in the canopy | - | 0.6 | 0.5 | 0.6 | 0.7 |

| Name | Description | Units | | | | |
|------|-------------|-------|---|---|---|---|
| CVPD | vapor pressure deficit at which stomatal conductance is halved | kPa | 2 | | | 0.55 |
| CS | ratio of projected SAI to HEIGHT | - | 0.035 | | | |
| ESLOPED | slope for evapotranspiration and snowmelt | degrees | 5 | 0 | 4 | |
| FRINTL | intercepted fraction of rain per unit LAI | - | 0.06 | 0.1 | 0.06 | 0.10 |
| FRINTS | intercepted fraction of rain per unit SAI | - | 0.06 | 0.1 | 0.06 | |
| FSINTL | intercepted fraction of snow per unit LAI | - | 0.04 | 0.1 | 0.04 | 0.09 |
| FSINTS | intercepted fraction of snow per unit SAI | - | 0.04 | 0.5 | 0.04 | |
| FXYLEM | fraction of plant resistance that is in the xylem | - | 0.5 | | | |
| GLMAXC | maximum leaf conductance | cm/s | 0.45 | 0.7 | 0.7 | 0.80 |
| GLMINC | minimum leaf conductance | cm/s | 0.03 | 0.07 | 0.03 | |
| IMPERV | fraction of the soil surface that is impermeable and always routes water reaching it directly to streamflow | - | 0.01 | 0 | | |
| KF | hydraulic conductivity at field capacity corresponding to THETAF and PSIF for a soil layer | mm/d | 6.3 | 13.1,13.1,13.1,13.1,5.5 | | |
| KSNVP | reduction factor between 0.05 and 1 to reduce snow evaporation | - | 0.3 | | | 0.08 |
| LATD | latitude | degrees | 50.96 | | | |
| LWIDTH | average leaf width | m | 0.07 | 0.05 | 0.03 | 0.05 |
| MAXHT | maximum canopy height for the year | m | 20.5 | 26 | 9 | |
| MAXLAI | maximum projected LAI for the year | $m^2/m^2$ | 6.3 | 4.5 | 6 | 5.65 |
| MXKPL | maximum plant conductivity | $mm\ day^{-1}\ MPa^{-1}$ | 8 | | 7 | 13.4 |
| MXRTLN | maximum length of fine roots per unit ground area | $m^2/m^2$ | 3200 | | 2000 | 3492 |
| NLAYER | number of soil layers to be used | - | 7 | 5 | | |
| PSICR | minimum plant leaf water potential | MPa | -2 | | -2.5 | -1.9 |
| PSIF | matric potential at "field capacity" corresponding to KF and THETAF for a soil layer | kPa | -8.5 | -25,-25,-25,-25,-7.9 | | |
| RELHT | pairs of day of the year and relative height between 0 and 1 | - | 1,1,366,1 | | | |
| RELLAI | pairs of day of the year and relative LAI between 0 and 1 | - | 1,0.482,41,0.219, 82,0.401,122,0.568, 163,1,203,0.826, 244,0.842,284,0.494, 325,0.393,366,0.482 | 1,0,54,0,84,1, 299,1, 329,0,366,0 | 1,0.3,40,0.4, 80,0.5,120,0.6, 160,1,200,1, 240,0.8,280,0.6, 320,0.4,366,0.3 | 1,0.06,40,0.23, 80,0.49,120,0.55, 160,1,200,1, 240,0.7,280,0.7, 320,0.33,366,0.2 |
| ROOTDEN | relative root density (per unit stonefree volume) of fine or absorbing roots for given layer | $m^3/m^2$ | 100,0.305,100,0.215, 100,0.15,100,0.10, 100,0.07,100,0.05, 100,0.045,100,0.025, | 100,0.22,100,0.17, 100,0.13,100,0.10, 100,0.08,100,0.06, 100,0.05 | 100,0.20,100,0.15, 100,0.12,100,0.09, 100,0.08,100,0.07, 100,0.05,100,0.04, | |

| | | | | |
|---|---|---|---|---|
| | | | 100,0.02,100,0.015,<br>100,0.01,100,0.01,<br>100,0.01,100,0.01,<br>100,0.005,100,0.005,<br>100,0 | 100,0.03,100,0.02,<br>100,0.01,100,0 |
| STONEF | stone volume fraction in each soil layer | - | 0.13,0.12,<br>0.12,0.14,<br>0.17,0.17,<br>0.18 | 0.09,0.10,0.12,0.10,0.4 |
| THETAF | volumetric water content at "field capacity" corresponding to KF and PSIF for soil layer | $m^3/m^2$ | 0.324 | 0.365,0.365,0.365,0.365,0.266 |
| THICK | layer thicknesses | mm | 25,75,125,225,<br>350,700,500 | 250,450,200,200,400 |
| THSAT | THETA at saturation | $m^3/m^2$ | 0.451 | 0.485,0.485,0.485,0.485,0.435 |
| WETINF | wetness at dry end of near-saturation range for a soil layer | - | 0.92 | |
| Z0G | ground surface roughness | m | 0.02 | |

## Tharandt

| Parameter abbreviation | Physical meaning | Unit | GBR90 | EXTR | BR90 | CBR90 |
|---|---|---|---|---|---|---|
| ALB | albedo or surface reflectivity without snow | - | 0.1 | 0.22 | 0.08 | 0.13 |
| ALBSN | albedo or surface reflectivity with snow | - | 0.28 | 0.34 | 0.40 | 0.60 |
| ASPECTD | aspect, degrees through east from north | degrees | 45 | 0 | 161 | |
| BEXP | exponent for ψ-θ relation | - | 5.39,5.39,<br>5.39,5.39,<br>5.39,4.9,<br>4.9 | 5.3 | | |
| CINTRL | maximum interception storage of rain per unit LAI | mm | 0.15 | 0.4 | 0.10 | 0.07 |
| CINTRS | maximum interception storage of rain per unit SAI | mm | 0.15 | 0.2 | 0,10 | |
| CINTSL | maximum interception storage of snow per unit LAI | mm | 0.6 | 1.6 | 0.5 | 0.2 |
| CINTSS | maximum interception storage of snow per unit SAI | mm | 0.6 | 0.8 | 0,5 | |
| CR | extinction coefficient for photosynthetically-active radiation in the canopy | - | 0.5 | | | 0.61 |
| CVPD | vapor pressure deficit at which stomatal conductance is halved | kPa | 2 | | | 0.78 |
| CS | ratio of projected SAI to HEIGHT | - | 0.035 | | 0.02 | |
| ESLOPED | slope for evapotranspiration and snowmelt | degrees | 5 | 0 | 4 | |
| FRINTL | intercepted fraction of rain per unit LAI | - | 0.06 | 0.08 | 0.06 | 0.02 |
| FRINTS | intercepted fraction of rain per unit SAI | - | 0.06 | 0.08 | 0.06 | |
| FSINTL | intercepted fraction of snow per unit LAI | - | 0.04 | 0.08 | 0.04 | 0.01 |
| FSINTS | intercepted fraction of snow per unit SAI | - | 0.04 | 0.1 | 0.04 | |

| | | | | | | |
|---|---|---|---|---|---|---|
| FXYLEM | fraction of plant resistance that is in the xylem | - | 0.5 | | 0.3 | |
| GLMAXC | maximum leaf conductance | cm/s | 0.34 | | 0.35 | 0.69 |
| GLMINC | minimum leaf conductance | cm/s | 0.03 | 0.01 | 0.02 | |
| IMPERV | fraction of the soil surface that is impermeable and always routes water reaching it directly to streamflow | - | 0.01 | 0 | | |
| KF | hydraulic conductivity at field capacity corresponding to THETAF and PSIF for a soil layer | mm/d | 6.3,6.3, 6.3,6.3, 6.3,5.5, 5.5 | 13.1 | | |
| KSNVP | reduction factor between 0.05 and 1 to reduce snow evaporation | - | 0.3 | | | 0.08 |
| LATD | latitude | degrees | 50.96 | | | |
| LWIDTH | average leaf width | m | 0.002 | 0.001 | 0.002 | 0.003 |
| MAXHT | maximum canopy height for the year | m | 23.2 | 29 | 30 | |
| MAXLAI | maximum projected LAI for the year | m²/m² | 6.2 | 7.6 | 7 | 5 |
| MXKPL | maximum plant conductivity | mm day$^{-1}$ MPa$^{-1}$ | 8 | | 7 | 7.5 |
| MXRTLN | maximum length of fine roots per unit ground area | m²/m² | 3100 | 3000 | 1700 | 1809 |
| NLAYER | number of soil layers to be used | - | 7 | 6 | | |
| PSICR | minimum plant leaf water potential | MPa | -2 | | -2.5 | -2.0 |
| PSIF | matric potential at "field capacity" corresponding to KF and THETAF for a soil layer | kPa | -8.5,-8.5, -8.5,-8.5, -8.5,-7.9, -7.9 | -25 | | |
| RELHT | pairs of day of the year and relative height between 0 and 1 | - | 1,1,366,1 | | | |
| RELLAI | pairs of day of the year and relative LAI between 0 and 1 | - | 1,1,366,1 | 1,0.8,160,1, 220,1,366,0.8 | 1,0.8,160,1, 220,1,366,0.8 | 1,0.5,140,0.8, 190,1,230,0.73, 320,0.6,366,0.5 |
| ROOTDEN | relative root density (per unit stonefree volume) of fine or absorbing roots for given layer | m³/m² | 100,0.27,100,0.195, 100,0.14,100,0.10, 100,0.075,100,0.065, 100,0.04,100,0.03, 100,0.025,100,0.015, 100,0.015,100,0.01, 100,0.005,100,0.005, 100,0.005,100,0.005, 100,0.005,100,0.005, 100,0.005,100,0 | 100,0.22,100,017, 100,0.13,100,0.1, 100,0.08,100,0.06, 100,0.05,100,0.04, 100,0.03,100,0.02, 100,0.01,100,0.01, 100,0.01 | 100,0.25,100,0.2, 100,0.15,100,0.1, 100,0.08,100,0.06, 100,0.05,100,0.04, 100,0.03,100,0.02, 100,0.01,100,0.01, 100,0.01 | |
| STONEF | stone volume fraction in each soil layer | - | 0.14,0.13,0.14, 0.16,0.18,0.21, 0.23 | 0.19,0.20,0.32, 0.40,0.42,0.42 | | |
| THETAF | volumetric water content at "field capacity" corresponding to KF and PSIF for soil layer | m³/m² | 0.324,0.324, 0.324,0.324, 0.324,0.266, 0.266 | 0.365 | | |
| THICK | layer thicknesses | mm | 25,75,125,225, 350,700,500 | 60,60,240,300,300,300 | | |
| THSAT | THETA at saturation | m³/m² | 0.451,0.451, 0.451,0.451, 0.451,0.435, 0.435 | 0.485 | | |

| WETINF | wetness at dry end of near-saturation range for a soil layer | - | 0.92 |
|---|---|---|---|
| Z0G | ground surface roughness | m | 0.02 |

## Oberbaerenburg

| Parameter abbreviation | Physical meaning | Unit | GBR90 | EXTR | BR90 | CBR90 |
|---|---|---|---|---|---|---|
| ALB | albedo or surface reflectivity without snow | - | 0.1 | 0.13 | 0.1 | 0.07 |
| ALBSN | albedo or surface reflectivity with snow | - | 0.28 | 0.34 | 0.4 | 0.45 |
| ASPECTD | aspect, degrees through east from north | degrees | 45 | 0 | 55 | |
| BEXP | exponent for ψ-θ relation | - | 5.39,5.39, 5.39, 4.9, 4.9,4.9, 4.9 | 4.9,4.9,4.9,4.9, 5.39,5.39,4.9, 4.9,5.3,5.3 | 4.9,5.39,4.9,5.3 | |
| CINTRL | maximum interception storage of rain per unit LAI | mm | 0.15 | 0.4 | 0.10 | |
| CINTRS | maximum interception storage of rain per unit SAI | mm | 0.15 | 0.2 | 0.10 | |
| CINTSL | maximum interception storage of snow per unit LAI | mm | 0.6 | 1.6 | 0.10 | |
| CINTSS | maximum interception storage of snow per unit SAI | mm | 0.6 | 0.8 | 0.5 | |
| CR | extinction coefficient for photosynthetically-active radiation in the canopy | - | 0.5 | | | |
| CVPD | vapor pressure deficit at which stomatal conductance is halved | kPa | 2 | | | |
| CS | ratio of projected SAI to HEIGHT | - | 0.035 | 0.02 | 0.02 | |
| ESLOPED | slope for evapotranspiration and snowmelt | degrees | 5 | 0 | 6 | |
| FRINTL | intercepted fraction of rain per unit LAI | - | 0.06 | 0.08 | 0.06 | |
| FRINTS | intercepted fraction of rain per unit SAI | - | 0.06 | 0.08 | 0.06 | |
| FSINTL | intercepted fraction of snow per unit LAI | - | 0.04 | 0.08 | 0.04 | 0.02 |
| FSINTS | intercepted fraction of snow per unit SAI | - | 0.04 | 0.1 | 0.04 | |
| FXYLEM | fraction of plant resistance that is in the xylem | - | 0.5 | | | |
| GLMAXC | maximum leaf conductance | cm/s | 0.34 | 0.34 | 0.45 | 0.60 |
| GLMINC | minimum leaf conductance | cm/s | 0.03 | 0.01 | 0.03 | |
| IMPERV | fraction of the soil surface that is impermeable and always routes water reaching it directly to streamflow | - | 0.01 | 0 | | |
| KF | hydraulic conductivity at field capacity corresponding to THETAF and PSIF for a soil layer | mm/d | 6.3,6.3, 6.3, 5.5, 5.5,5.5, 5.5 | 5.5,5.5,5.5,5.5, 6.3,6.6,5.5,5.5, 5.5,13.1,13.1 | 5.5,6.3,5.5,13.1 | |
| KSNVP | reduction factor between 0.05 and 1 to reduce snow evaporation | - | 0.3 | | | 0.5 |
| LATD | latitude | degrees | 50.797 | | | |
| LWIDTH | average leaf width | m | 0.002 | 0.001 | 0.002 | 0.003 |

| | | | | | | |
|---|---|---|---|---|---|---|
| MAXHT | maximum canopy height for the year | m | 20 | 29 | 25 | |
| MAXLAI | maximum projected LAI for the year | m²/m² | 7 | 7.6 | 7.5 | 6 |
| MXKPL | maximum plant conductivity | mm day⁻¹ MPa⁻¹ | 8 | 8 | 7 | |
| MXRTLN | maximum length of fine roots per unit ground area | m²/m² | 3100 | 3000 | 1500 | 2000 |
| NLAYER | number of soil layers to be used | - | 7 | 11 | 4 | |
| PSICR | minimum plant leaf water potential | MPa | -2 | | -2.5 | -1.5 |
| PSIF | matric potential at "field capacity" corresponding to KF and THETAF for a soil layer | kPa | -8.5,-8.5, -8.5,-7.9, -7.9,-7.9, -7.9 | -25 | -7.9,-8.5,-7.9,-25 | |
| RELHT | pairs of day of the year and relative height between 0 and 1 | - | 1,1,366,1 | | | |
| RELLAI | pairs of day of the year and relative LAI between 0 and 1 | - | 1,1,366,1 | 1,0.8,160,1, 220,1,366,0.8 | 1,0.8,160,1, 220,1,366,0.8 | 1,0.6,75,0.6, 100,0.98,140,1, 200,1,230,0.9, 300,0.6,366,0.6 |
| ROOTDEN | relative root density (per unit stonefree volume) of fine or absorbing roots for given layer | m³/m² | 100,0.27,100,0.195, 100,0.14,100,0.10, 100,0.075,100,0.065, 100,0.04,100,0.03, 100,0.025,100,0.015, 100,0.015,100,0.01, 100,0.005,100,0.005, 100,0.005,100,0.005, 100,0.005,100,0.005, 100,0.005,100,0 | 100,0.3,100, 0.2,100,0.13, 100,0.1,100,0.08, 100,0.06,100,0.05, 100,0.04,100,0.03, 100,0.02,100,0.01, 100,0.01,100,0 | 100,0.3,100, 0.2,100,0.13, 100,0.1,100,0.08, 100,0.06,100,0.05, 100,0.04,100,0.03, 100,0.02,100,0.01, 100,0.01,100,0 | |
| STONEF | stone volume fraction in each soil layer | - | 0.16,0.16,0.17, 0.20,0.24,0.26, 0.27 | 0.737,0.737,0.771, 0.771,0.518,0.518, 0.574,0.574,0.581, 0.711,0.722 | 0.115,0.23,0.29,0.42 | |
| THETAF | volumetric water content at "field capacity" corresponding to KF and PSIF for soil layer | m³/m² | 0.324,0.324, 0.324, 0.266, 0.266,0.266, 0.266 | 0.266,0.266,0.266, 0.266,0.324,0.324, 0.266,0.2660.266, 0.365,0.365 | 0.266,0.324,0.266,0.365 | |
| THICK | layer thicknesses | mm | 25,75,125,225, 350,700,500 | 30,40,50,60, 60,50,50,60, 60,70,490 | 180,110,170,560 | |
| THSAT | THETA at saturation | m³/m² | 0.451,0.451, 0.451, 0.435, 0.435,0.435, 0.435 | 0.435,0.435,0.435, 0.435,0.451,0.451, 0.435,0.435,0.435, 0.485,0.485 | 0.435,0.451,0.435,0.485 | |
| WETINF | wetness at dry end of near-saturation range for a soil layer | - | 0.92 | | | |
| Z0G | ground surface roughness | m | 0.02 | | | |

*for GBR90 and EXTRUSO listed parameters denote to the dominant HRU

Table C2 BROOK90 parameters and their ranges chosen for the calibration

| Parameter abbreviation | Physical meaning | Unit | Range | | | | |
|---|---|---|---|---|---|---|---|
| | | | G | K | H | T | O |
| ALB | albedo or surface reflectivity without snow | - | 0.1-0.3 | 0.1-0.3 | 0.1-0.3 | 0.05-0.15 | 0.07-0.13 |
| ALBSN | albedo or surface reflectivity with snow | - | 0.4-0.6 | 0.4-0.6 | 0.3-0.5 | 0.4-0.6 | 0.35-0.45 |
| CINTRL | maximum interception storage of rain per unit LAI | mm | 0.1-0.3 | 0.1-0.3 | 0.1-0.3 | 0.07-0.15 | 0.10-0.15 |
| CINTSL | maximum interception storage of snow per unit LAI | mm | 0.4-0.8 | 0.4-0.8 | 0.1-0.6 | 0.2-0.4 | 0.1-0.3 |
| CR | extinction coefficient for photosynthetically-active radiation in the canopy | - | 0.6-0.8 | 0.6-0.8 | 0.5-0.7 | 0.5-0.7 | 0.5-0.7 |
| CVPD | vapor pressure deficit at which stomatal conductance is halved | kPa | 0.5-2 | 0.5-2 | 0.5-2 | 0.5-2 | 0.5-2 |
| FRINTL | intercepted fraction of rain per unit LAI | - | 0.04-0.1 | 0.04-0.1 | 0.01-0.1 | 0.02-0.06 | 0.06-0.08 |
| FSINTL | intercepted fraction of snow per unit LAI | - | 0.04-0.07 | 0.01-0.05 | 0.01-0.1 | 0.01-0.04 | 0.02-0.04 |
| GLMAXC | maximum leaf conductance | cm/s | 1-1.5 | 1-1.5 | 0.3-2 | 0.3-0.7 | 0.3-0.6 |
| KSNVP | reduction factor for snow evaporation | - | - | - | 0.05-0.5 | 0.05-0.5 | 0.05-0.5 |
| LWIDTH | average leaf width | m | 0.010-0.025 | 0.015-0.045 | 0.02-0.05 | 0.001-0.003 | 0.001-0.003 |
| MAXLAI | maximum projected LAI for the year | $m^2/m^2$ | 4-6 | 3-6 | 5-7 | 5-8 | 6-8 |
| MXKPL | maximum plant conductivity | mm day$^{-1}$ MPa$^{-1}$ | 7-30 | 7-30 | 7-30 | 7-30 | 7-30 |
| MXRTLN | maximum length of fine roots per unit ground area | $m^2/m^2$ | 600-1000 | 300-700 | 1500-4000 | 1500-2500 | 2000-3500 |
| PSICR | minimum plant leaf water potential | MPa | -2.5 – -1.5 | -2.5 – -1.5 | -2.5 – -1.5 | -2.5 – -1.5 | -2.5 – -1.5 |
| RELHT | pairs of day of the year and relative height between 0 and 1 | - | Adjusting relative values for spring and autumn (G,K,H) and for winter (T,O) periods for fixed time-steps | | | | |
| RELLAI | pairs of day of the year and relative LAI between 0 and 1 | - | | | | | |
| IDEPTH | depth over which infiltration is distributed | mm | 0-1330 | 0-800 | 0-1500 | 0-1260 | 0-1020 |
| QFFC | quick flow fraction bypass flow at field capacity | - | 0-0.5 | 0-0.5 | 0-0.5 | 0-0.5 | 0-0.5 |
| QFPAR | fraction of the water content between field capacity and saturation at which the quick flow fraction is 1 | - | 0-0.5 | 0-0.5 | 0-0.5 | 0-0.5 | 0-0.5 |
| DRAIN | multiplier between 0 and 1 of drainage from the lowest soil layer | - | 0-1 | 0-1 | 0-1 | 0-1 | 0-1 |

Abbreviations for ranges: G – Grillenburg, K – Klingenberg, H – Hetzdorf, T – Tharandt, O – Oberbaerenburg

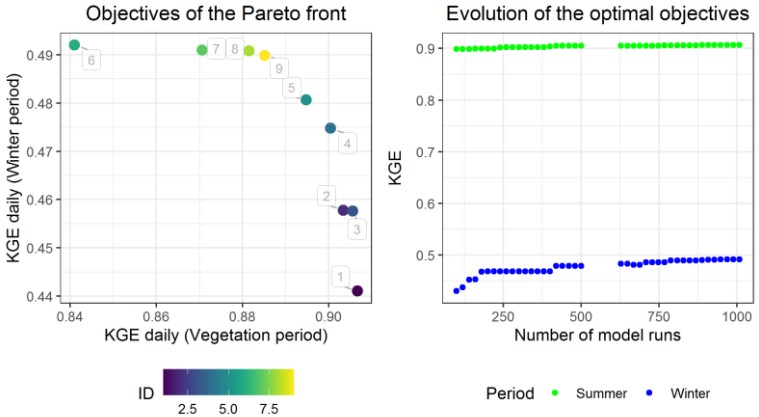

**Figure C1 Resulted calibration Pareto fronts for Grillenburg (chosen ID – 9)**

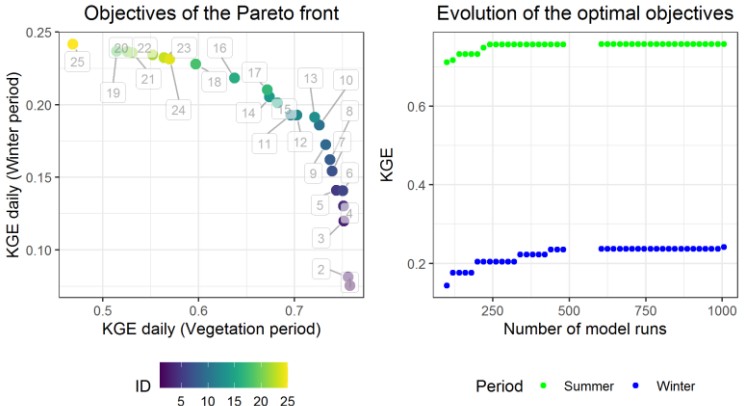

**Figure C2 Resulted calibration Pareto fronts for Klingenberg (chosen ID – 13)**

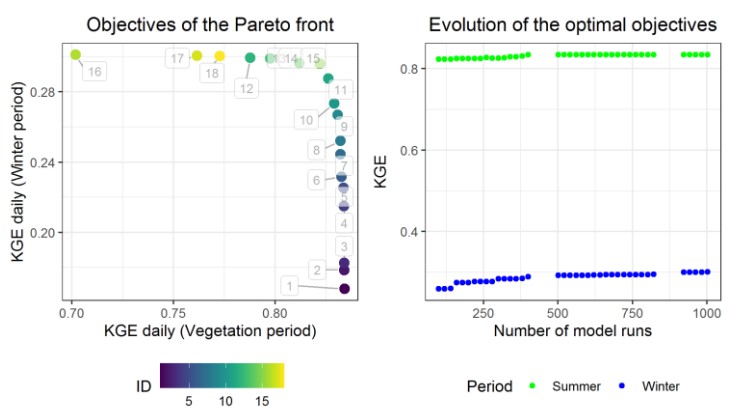

**Figure C3 Resulted calibration Pareto fronts for Hetzdorf (chosen ID – 15)**

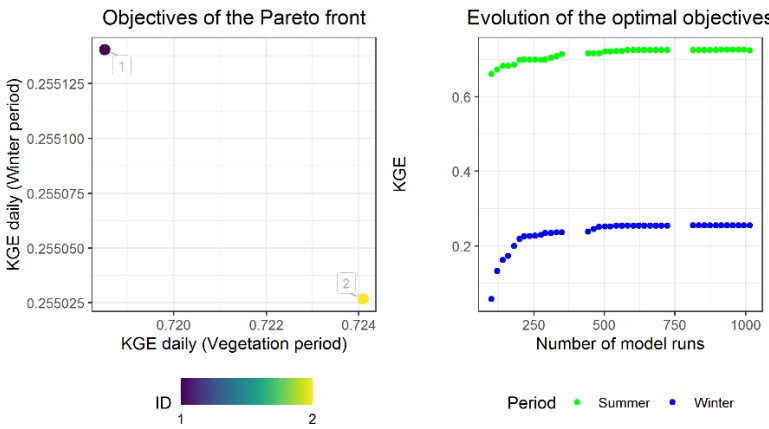

**Figure C4 Resulted calibration Pareto fronts for Tharandt (chosen ID – 2)**

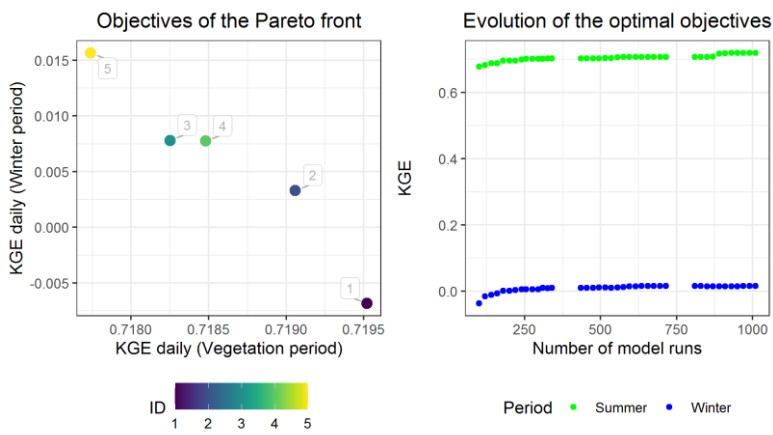

**Figure C5 Resulted calibration Pareto fronts for Oberbaerenburg (chosen ID – 5)**

 **Appendix D. Daily (2020) and monthly (whole time-series) simulations**

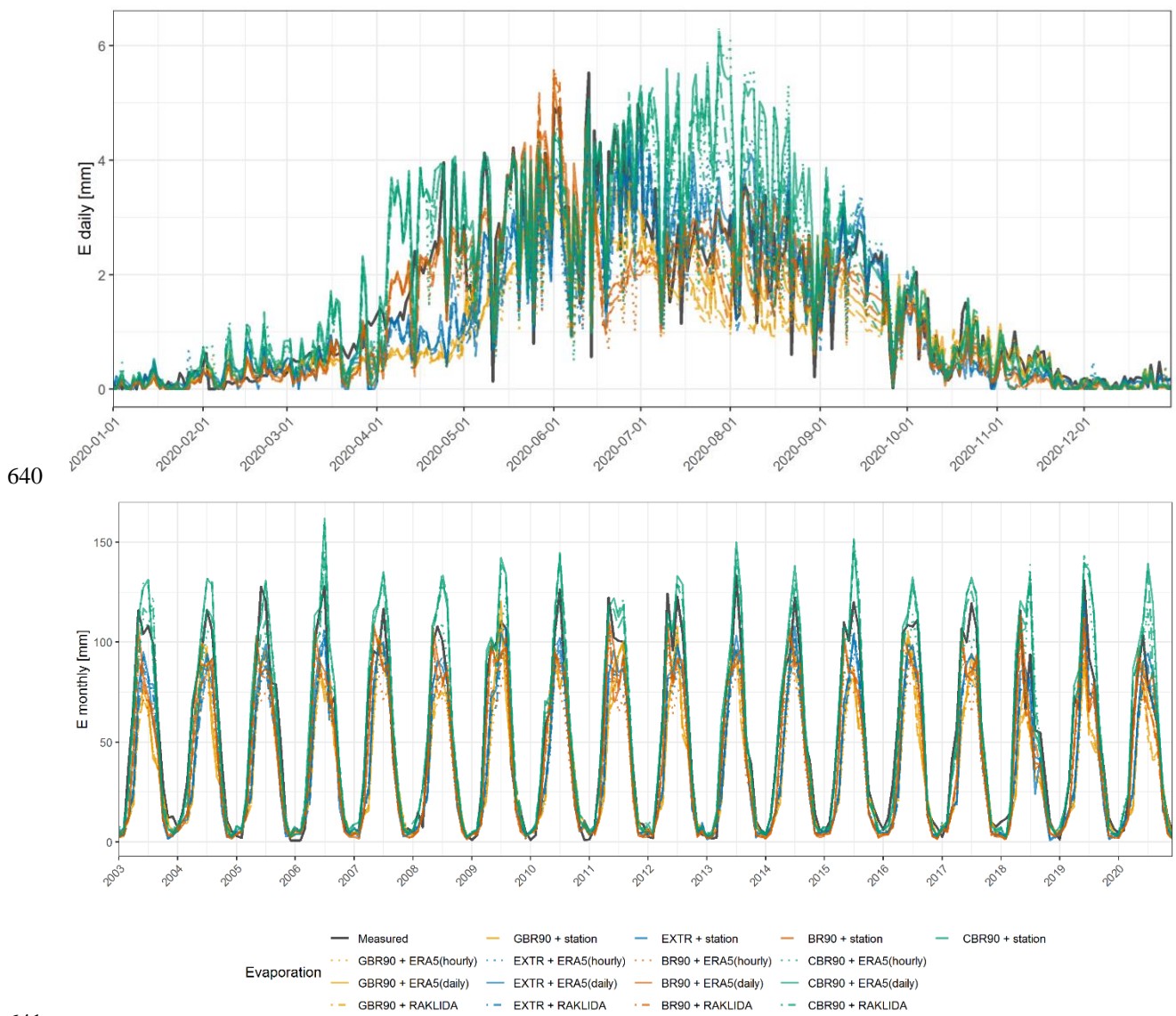

**Figure D1 Grillenburg**

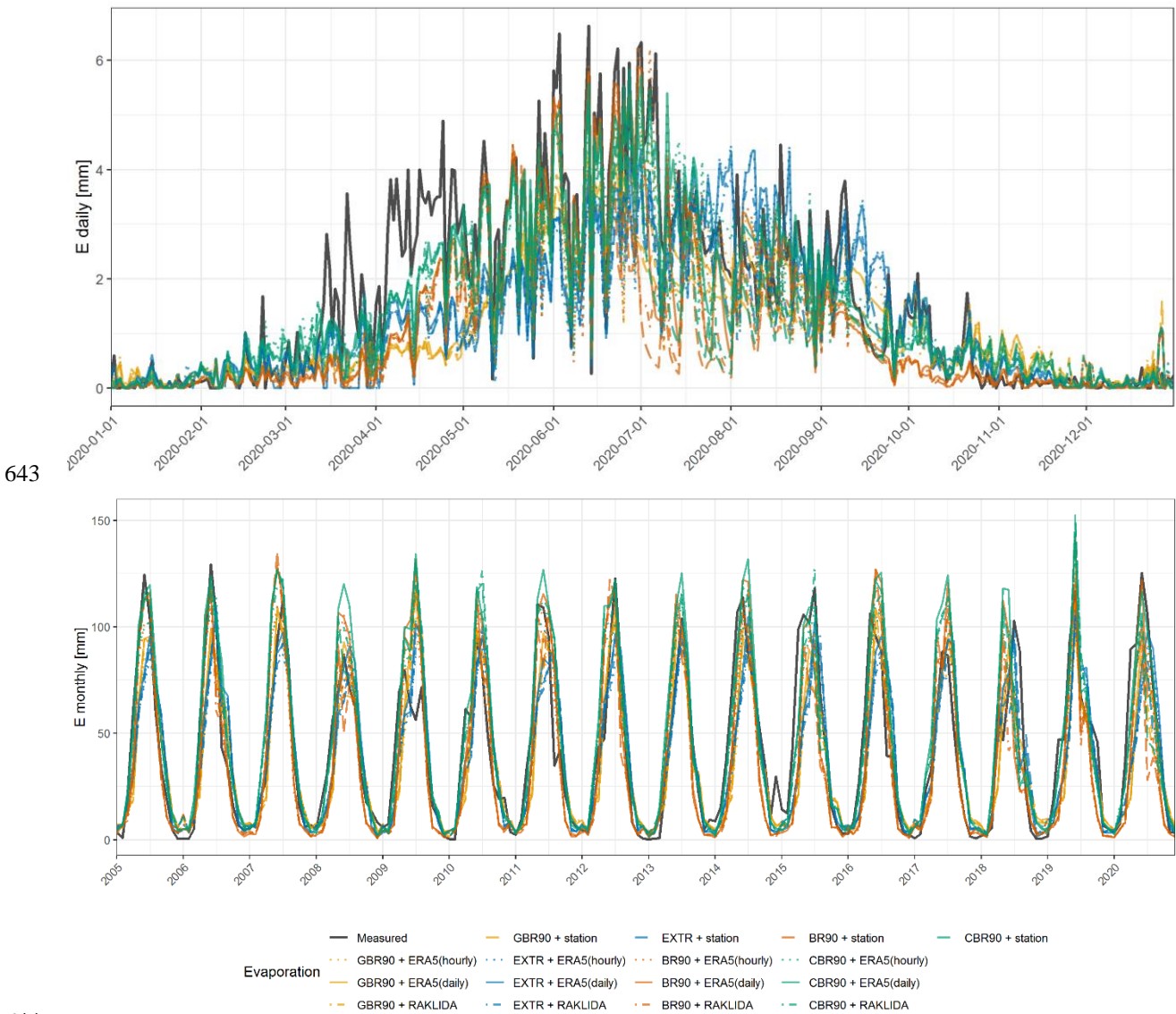

**Figure D2 Klingenberg**

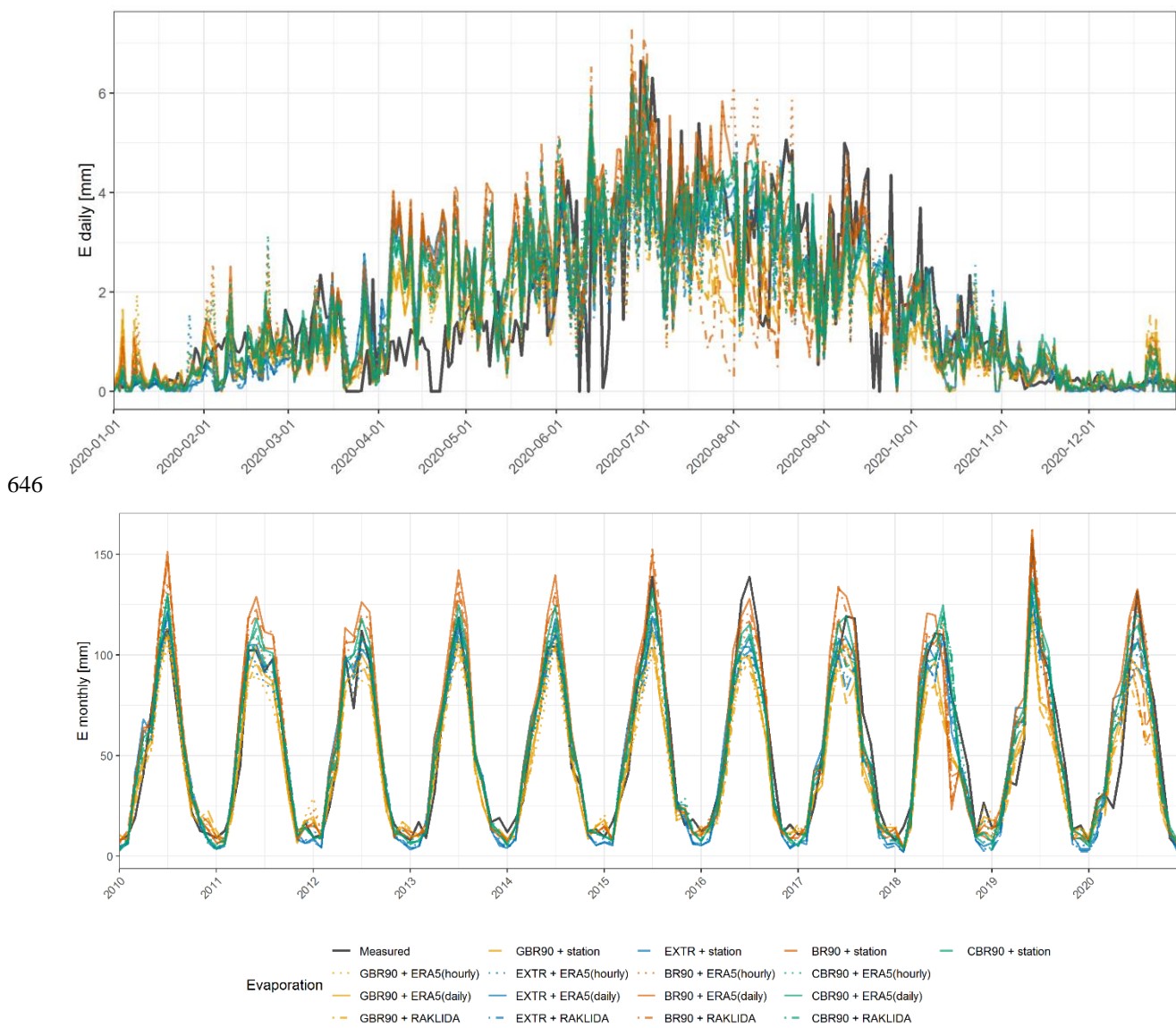

**Figure D3 Hetzdorf**

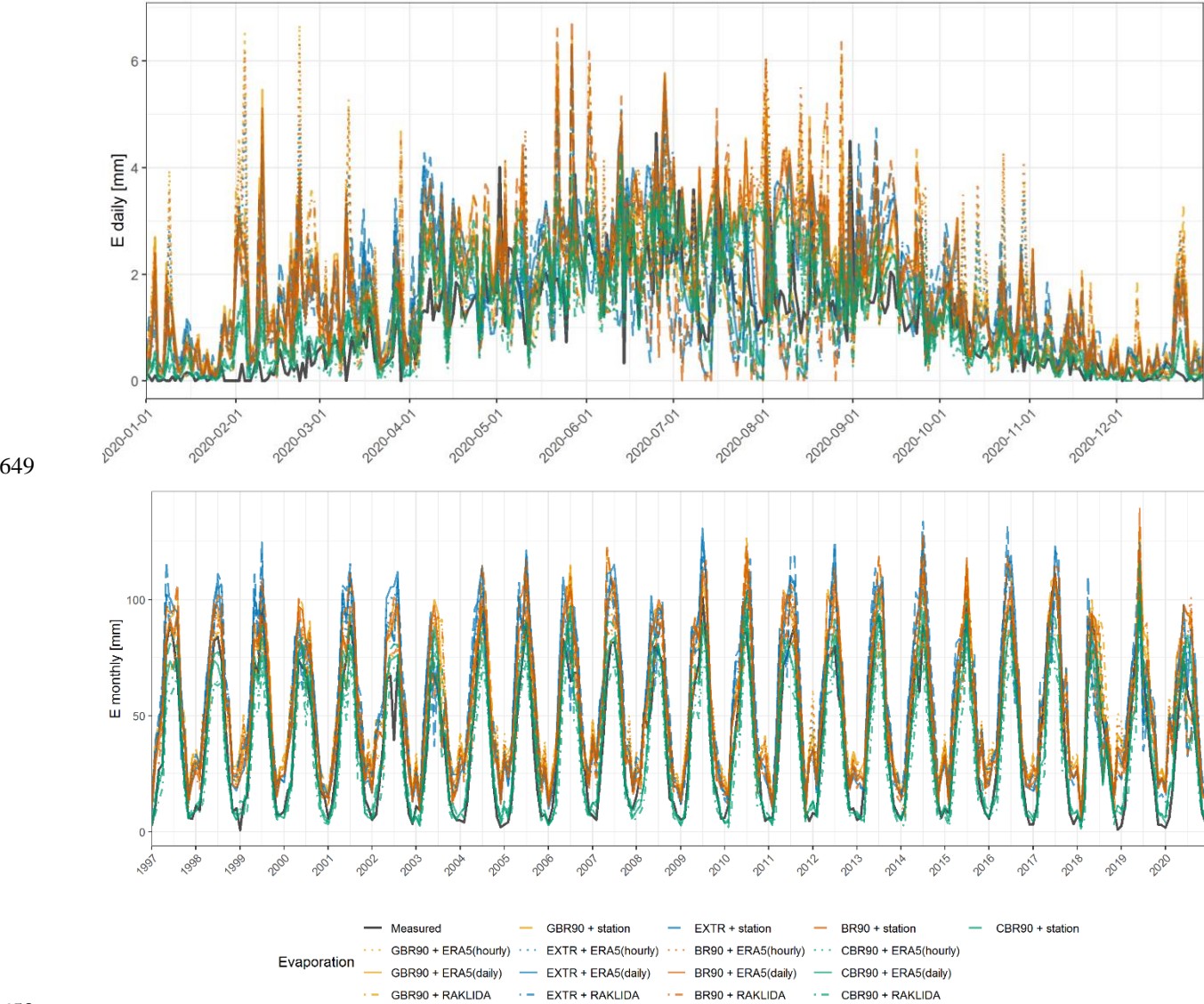

**Figure D4 Tharandt**

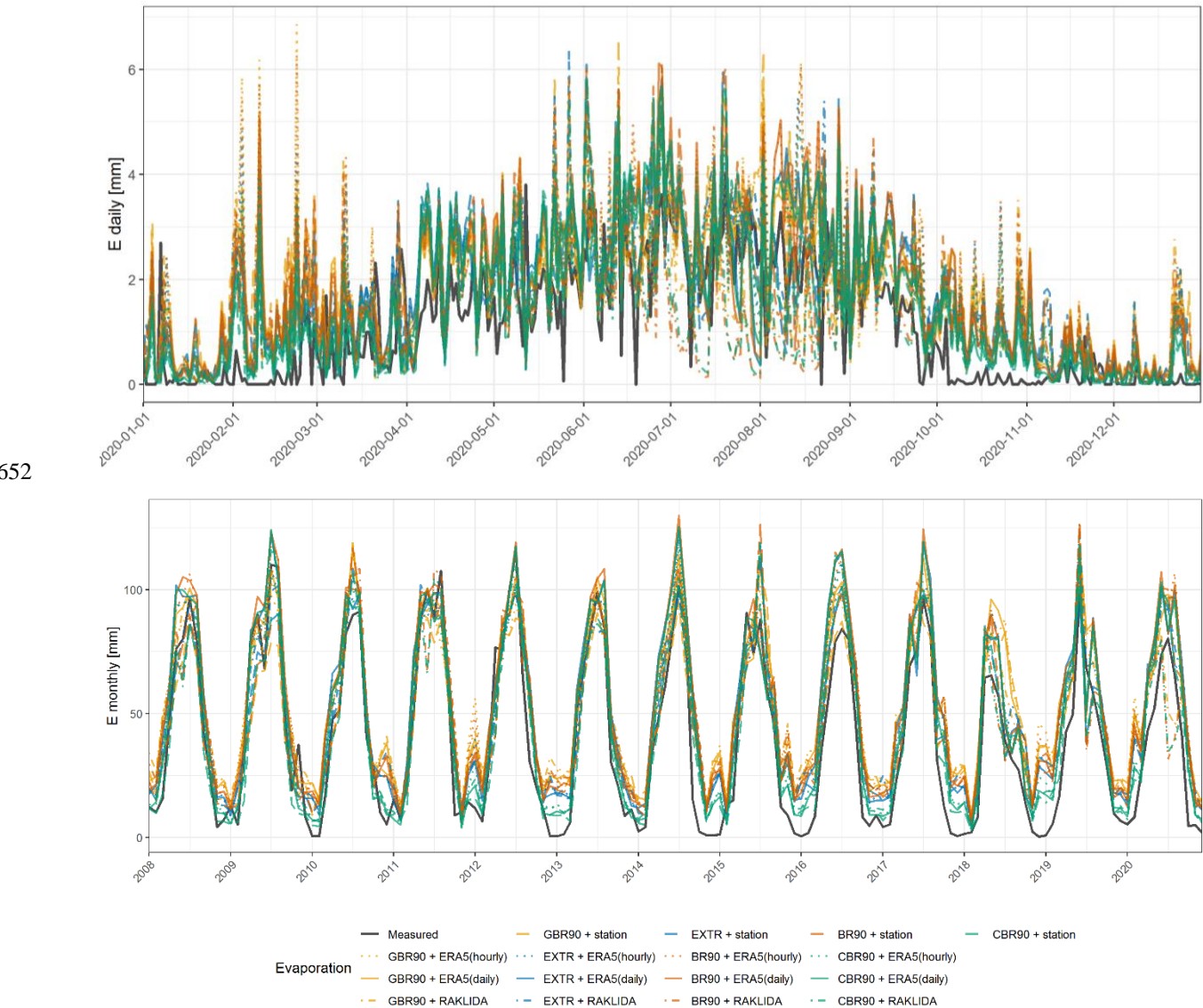

**Figure D5 Oberbaerenburg**

 **Appendix E. Evaluation of the simulated evaporation**

Table E1. Daily evaporation skill-scores for the whole year

| Model/Station | | Grillenburg | Klingenberg | Hetzdorf | Tharandt | Oberbaerenburg |
|---|---|---|---|---|---|---|
| NSE | | | | | | |
| GBR90 | ERA5 h | 0.03 | 0.2 | 0.37 | 0.05 | -0.09 |
| | ERA5 d | 0.06 | 0.29 | 0.56 | 0.25 | 0.13 |
| | RaKliDa | -0.05 | 0.23 | 0.49 | 0.09 | 0.06 |
| | Station | 0.08 | 0.25 | 0.53 | 0.23 | 0.14 |
| EXTR | ERA5 h | 0.45 | 0.32 | 0.55 | 0.26 | 0.19 |
| | ERA5 d | 0.57 | 0.43 | 0.68 | 0.38 | 0.33 |
| | RaKliDa | 0.5 | 0.3 | 0.65 | 0.32 | 0.26 |
| | Station | 0.61 | 0.4 | 0.69 | 0.29 | 0.36 |
| BR90 | ERA5 h | 0.46 | 0.53 | 0.61 | 0.13 | 0.09 |
| | ERA5 d | 0.61 | 0.56 | 0.69 | 0.36 | 0.31 |
| | RaKliDa | 0.59 | 0.51 | 0.67 | 0.17 | 0.18 |
| | Station | 0.63 | 0.5 | 0.71 | 0.32 | 0.33 |
| CBR90 | ERA5 h | 0.76 | 0.51 | 0.57 | 0.48 | 0.35 |
| | ERA5 d | 0.83 | 0.61 | 0.72 | 0.59 | 0.52 |
| | RaKliDa | 0.85 | 0.59 | 0.69 | 0.28 | 0.41 |
| | Station | 0.86 | 0.6 | 0.74 | 0.63 | 0.53 |
| KGE | | | | | | |
| GBR90 | ERA5 h | 0.36 | 0.57 | 0.65 | 0.45 | 0.46 |
| | ERA5 d | 0.4 | 0.63 | 0.74 | 0.58 | 0.56 |
| | RaKliDa | 0.33 | 0.58 | 0.69 | 0.47 | 0.52 |
| | Station | 0.36 | 0.6 | 0.7 | 0.5 | 0.57 |
| EXTR | ERA5 h | 0.51 | 0.62 | 0.77 | 0.54 | 0.58 |
| | ERA5 d | 0.59 | 0.7 | 0.84 | 0.59 | 0.63 |
| | RaKliDa | 0.53 | 0.6 | 0.82 | 0.57 | 0.61 |
| | Station | 0.59 | 0.67 | 0.84 | 0.52 | 0.66 |
| BR90 | ERA5 h | 0.53 | 0.72 | 0.78 | 0.47 | 0.5 |
| | ERA5 d | 0.7 | 0.76 | 0.78 | 0.6 | 0.6 |
| | RaKliDa | 0.65 | 0.72 | 0.78 | 0.51 | 0.55 |
| | Station | 0.66 | 0.72 | 0.82 | 0.52 | 0.63 |
| CBR90 | ERA5 h | 0.88 | 0.76 | 0.79 | 0.73 | 0.66 |
| | ERA5 d | 0.84 | 0.76 | 0.85 | 0.79 | 0.71 |
| | RaKliDa | 0.86 | 0.79 | 0.85 | 0.59 | 0.69 |
| | Station | 0.9 | 0.79 | 0.86 | 0.81 | 0.77 |
| Correlation | | | | | | |
| GBR90 | ERA5 h | 0.78 | 0.73 | 0.77 | 0.54 | 0.53 |
| | ERA5 d | 0.79 | 0.75 | 0.83 | 0.69 | 0.67 |
| | RaKliDa | 0.79 | 0.75 | 0.81 | 0.52 | 0.59 |
| | Station | 0.81 | 0.75 | 0.83 | 0.67 | 0.62 |
| EXTR | ERA5 h | 0.88 | 0.77 | 0.79 | 0.67 | 0.66 |
| | ERA5 d | 0.89 | 0.78 | 0.84 | 0.75 | 0.73 |
| | RaKliDa | 0.89 | 0.77 | 0.83 | 0.68 | 0.66 |
| | Station | 0.9 | 0.78 | 0.85 | 0.69 | 0.71 |
| BR90 | ERA5 h | 0.86 | 0.78 | 0.79 | 0.57 | 0.59 |

| | | | | | | |
|---|---|---|---|---|---|---|
| | ERA5 d | 0.86 | 0.77 | 0.84 | 0.72 | 0.71 |
| | RaKliDa | 0.87 | 0.77 | 0.82 | 0.55 | 0.62 |
| | Station | 0.89 | 0.76 | 0.85 | 0.68 | 0.68 |
| CBR90 | ERA5 h | 0.88 | 0.77 | 0.8 | 0.78 | 0.71 |
| | ERA5 d | 0.92 | 0.81 | 0.86 | 0.82 | 0.78 |
| | RaKliDa | 0.93 | 0.8 | 0.85 | 0.73 | 0.72 |
| | Station | 0.93 | 0.8 | 0.87 | 0.81 | 0.77 |
| BIAS | | | | | | |
| GBR90 | ERA5 h | 0.69 | 0.84 | 0.85 | 1.38 | 1.37 |
| | ERA5 d | 0.72 | 0.91 | 0.89 | 1.39 | 1.4 |
| | RaKliDa | 0.7 | 0.87 | 0.84 | 1.22 | 1.35 |
| | Station | 0.7 | 0.87 | 0.85 | 1.49 | 1.23 |
| EXTR | ERA5 h | 0.73 | 0.88 | 0.94 | 1.4 | 1.31 |
| | ERA5 d | 0.77 | 0.94 | 0.99 | 1.42 | 1.35 |
| | RaKliDa | 0.73 | 0.87 | 0.95 | 1.34 | 1.26 |
| | Station | 0.75 | 0.9 | 0.95 | 1.44 | 1.21 |
| BR90 | ERA5 h | 0.73 | 0.86 | 1.03 | 1.36 | 1.37 |
| | ERA5 d | 0.83 | 0.94 | 1.1 | 1.34 | 1.38 |
| | RaKliDa | 0.8 | 0.87 | 1.05 | 1.17 | 1.31 |
| | Station | 0.8 | 0.87 | 1.04 | 1.41 | 1.21 |
| CBR90 | ERA5 h | 0.99 | 1.06 | 0.99 | 0.9 | 1.19 |
| | ERA5 d | 1.13 | 1.16 | 1.03 | 0.94 | 1.23 |
| | RaKliDa | 1.11 | 1.09 | 0.98 | 0.78 | 1.16 |
| | Station | 1.07 | 1.09 | 0.98 | 1.02 | 1.06 |
| Variance ratio | | | | | | |
| GBR90 | ERA5 h | 0.51 | 0.62 | 0.7 | 1.31 | 0.95 |
| | ERA5 d | 0.5 | 0.64 | 0.74 | 1.15 | 0.87 |
| | RaKliDa | 0.47 | 0.59 | 0.76 | 1.29 | 0.97 |
| | Station | 0.49 | 0.61 | 0.74 | 1.47 | 0.9 |
| EXTR | ERA5 h | 0.59 | 0.62 | 0.88 | 1.32 | 0.92 |
| | ERA5 d | 0.64 | 0.7 | 0.98 | 1.31 | 0.95 |
| | RaKliDa | 0.61 | 0.61 | 0.97 | 1.35 | 0.97 |
| | Station | 0.66 | 0.66 | 0.97 | 1.51 | 0.94 |
| BR90 | ERA5 h | 0.63 | 0.96 | 1.17 | 1.42 | 1.08 |
| | ERA5 d | 0.75 | 1.09 | 1.31 | 1.25 | 1.04 |
| | RaKliDa | 0.7 | 0.97 | 1.31 | 1.35 | 1.08 |
| | Station | 0.71 | 1 | 1.21 | 1.61 | 1.03 |
| CBR90 | ERA5 h | 0.98 | 0.91 | 0.89 | 0.79 | 0.94 |
| | ERA5 d | 1.18 | 1.08 | 1.03 | 0.86 | 1.01 |
| | RaKliDa | 1.15 | 0.96 | 0.99 | 0.76 | 0.96 |
| | Station | 1.11 | 1.01 | 1.02 | 1.02 | 0.97 |
| MAE | | | | | | |
| GBR90 | ERA5 h | 0.76 | 0.69 | 0.71 | 0.86 | 0.97 |
| | ERA5 d | 0.72 | 0.66 | 0.61 | 0.77 | 0.88 |
| | RaKliDa | 0.75 | 0.66 | 0.67 | 0.88 | 0.91 |
| | Station | 0.69 | 0.66 | 0.62 | 0.87 | 0.86 |
| EXTR | ERA5 h | 0.64 | 0.64 | 0.66 | 0.81 | 0.84 |
| | ERA5 d | 0.59 | 0.62 | 0.59 | 0.78 | 0.82 |
| | RaKliDa | 0.62 | 0.64 | 0.62 | 0.78 | 0.81 |

| | | | | | |
|---|---|---|---|---|---|
| | Station | 0.56 | 0.62 | 0.58 | 0.89 | 0.76 |
| BR90 | ERA5 h | 0.64 | 0.65 | 0.7 | 0.85 | 0.94 |
| | ERA5 d | 0.59 | 0.67 | 0.65 | 0.74 | 0.86 |
| | RaKliDa | 0.59 | 0.66 | 0.67 | 0.85 | 0.92 |
| | Station | 0.55 | 0.67 | 0.61 | 0.85 | 0.82 |
| CBR90 | ERA5 h | 0.52 | 0.66 | 0.64 | 0.5 | 0.73 |
| | ERA5 d | 0.48 | 0.64 | 0.57 | 0.47 | 0.69 |
| | RaKliDa | 0.46 | 0.62 | 0.58 | 0.6 | 0.73 |
| | Station | 0.42 | 0.61 | 0.54 | 0.5 | 0.63 |

Table E2. Daily evaporation skill-scores for the vegetation period

| Model/Station | | Grillenburg | Klingenberg | Hetzdorf | Tharandt | Oberbaerenburg |
|---|---|---|---|---|---|---|
| NSE | | | | | | |
| GBR90 | ERA5 h | -0.46 | -0.13 | 0.09 | -0.12 | -0.33 |
| | ERA5 d | -0.52 | -0.07 | 0.33 | 0.06 | -0.09 |
| | RaKliDa | -0.64 | -0.13 | 0.28 | 0 | -0.06 |
| | Station | -0.45 | -0.08 | 0.33 | 0.08 | 0.04 |
| EXTR | ERA5 h | 0.17 | -0.08 | 0.21 | 0.03 | -0.07 |
| | ERA5 d | 0.33 | 0.08 | 0.4 | 0.14 | 0.1 |
| | RaKliDa | 0.26 | -0.09 | 0.4 | 0.15 | 0.14 |
| | Station | 0.41 | 0.09 | 0.47 | 0.12 | 0.27 |
| BR90 | ERA5 h | 0.19 | 0.38 | 0.43 | -0.03 | -0.11 |
| | ERA5 d | 0.39 | 0.41 | 0.53 | 0.2 | 0.13 |
| | RaKliDa | 0.35 | 0.37 | 0.52 | 0.08 | 0.08 |
| | Station | 0.43 | 0.37 | 0.58 | 0.2 | 0.26 |
| CBR90 | ERA5 h | 0.62 | 0.24 | 0.3 | 0.22 | 0.11 |
| | ERA5 d | 0.72 | 0.37 | 0.52 | 0.37 | 0.32 |
| | RaKliDa | 0.75 | 0.38 | 0.51 | 0 | 0.23 |
| | Station | 0.78 | 0.42 | 0.59 | 0.45 | 0.42 |
| KGE | | | | | | |
| GBR90 | ERA5 h | 0.33 | 0.49 | 0.57 | 0.38 | 0.39 |
| | ERA5 d | 0.34 | 0.52 | 0.67 | 0.54 | 0.53 |
| | RaKliDa | 0.28 | 0.48 | 0.64 | 0.39 | 0.49 |
| | Station | 0.33 | 0.51 | 0.66 | 0.47 | 0.54 |
| EXTR | ERA5 h | 0.51 | 0.5 | 0.63 | 0.48 | 0.52 |
| | ERA5 d | 0.59 | 0.57 | 0.71 | 0.55 | 0.59 |
| | RaKliDa | 0.53 | 0.49 | 0.71 | 0.49 | 0.58 |
| | Station | 0.6 | 0.56 | 0.74 | 0.46 | 0.64 |
| BR90 | ERA5 h | 0.53 | 0.66 | 0.68 | 0.39 | 0.45 |
| | ERA5 d | 0.67 | 0.68 | 0.71 | 0.56 | 0.58 |
| | RaKliDa | 0.64 | 0.66 | 0.7 | 0.41 | 0.51 |
| | Station | 0.66 | 0.65 | 0.75 | 0.46 | 0.61 |
| CBR90 | ERA5 h | 0.81 | 0.65 | 0.67 | 0.63 | 0.58 |
| | ERA5 d | 0.82 | 0.68 | 0.77 | 0.7 | 0.67 |
| | RaKliDa | 0.84 | 0.7 | 0.76 | 0.51 | 0.62 |
| | Station | 0.87 | 0.71 | 0.8 | 0.71 | 0.71 |
| Correlation | | | | | | |
| GBR90 | ERA5 h | 0.67 | 0.61 | 0.68 | 0.43 | 0.43 |

| | | | | | | |
|---|---|---|---|---|---|---|
| | ERA5 d | 0.66 | 0.63 | 0.75 | 0.59 | 0.6 |
| | RaKliDa | 0.67 | 0.64 | 0.73 | 0.42 | 0.52 |
| | Station | 0.71 | 0.64 | 0.76 | 0.58 | 0.55 |
| EXTR | ERA5 h | 0.81 | 0.66 | 0.67 | 0.55 | 0.56 |
| | ERA5 d | 0.83 | 0.68 | 0.74 | 0.64 | 0.65 |
| | RaKliDa | 0.83 | 0.66 | 0.73 | 0.57 | 0.6 |
| | Station | 0.85 | 0.69 | 0.76 | 0.57 | 0.65 |
| BR90 | ERA5 h | 0.79 | 0.7 | 0.7 | 0.45 | 0.49 |
| | ERA5 d | 0.78 | 0.69 | 0.76 | 0.62 | 0.64 |
| | RaKliDa | 0.8 | 0.69 | 0.74 | 0.45 | 0.54 |
| | Station | 0.82 | 0.68 | 0.78 | 0.59 | 0.62 |
| CBR90 | ERA5 h | 0.81 | 0.66 | 0.69 | 0.67 | 0.61 |
| | ERA5 d | 0.87 | 0.72 | 0.78 | 0.72 | 0.71 |
| | RaKliDa | 0.88 | 0.72 | 0.77 | 0.61 | 0.64 |
| | Station | 0.89 | 0.72 | 0.8 | 0.72 | 0.71 |
| BIAS | | | | | | |
| GBR90 | ERA5 h | 0.68 | 0.83 | 0.83 | 1.22 | 1.22 |
| | ERA5 d | 0.72 | 0.9 | 0.88 | 1.26 | 1.27 |
| | RaKliDa | 0.68 | 0.85 | 0.84 | 1.07 | 1.2 |
| | Station | 0.69 | 0.85 | 0.84 | 1.34 | 1.1 |
| EXTR | ERA5 h | 0.73 | 0.88 | 0.97 | 1.29 | 1.22 |
| | ERA5 d | 0.77 | 0.94 | 1.03 | 1.32 | 1.26 |
| | RaKliDa | 0.73 | 0.87 | 0.99 | 1.23 | 1.15 |
| | Station | 0.76 | 0.9 | 1 | 1.32 | 1.11 |
| BR90 | ERA5 h | 0.74 | 0.87 | 1.04 | 1.23 | 1.25 |
| | ERA5 d | 0.84 | 0.96 | 1.12 | 1.24 | 1.27 |
| | RaKliDa | 0.81 | 0.88 | 1.07 | 1.05 | 1.18 |
| | Station | 0.81 | 0.88 | 1.05 | 1.29 | 1.1 |
| CBR90 | ERA5 h | 0.99 | 1.06 | 0.99 | 0.89 | 1.15 |
| | ERA5 d | 1.13 | 1.17 | 1.05 | 0.94 | 1.2 |
| | RaKliDa | 1.11 | 1.08 | 1 | 0.78 | 1.11 |
| | Station | 1.07 | 1.08 | 1 | 1.01 | 1.03 |
| Variance ratio | | | | | | |
| GBR90 | ERA5 h | 0.55 | 0.62 | 0.71 | 1.32 | 0.87 |
| | ERA5 d | 0.5 | 0.6 | 0.72 | 1.13 | 0.77 |
| | RaKliDa | 0.49 | 0.57 | 0.8 | 1.45 | 0.97 |
| | Station | 0.51 | 0.6 | 0.75 | 1.59 | 0.91 |
| EXTR | ERA5 h | 0.63 | 0.56 | 0.75 | 1.33 | 0.83 |
| | ERA5 d | 0.67 | 0.61 | 0.78 | 1.31 | 0.85 |
| | RaKliDa | 0.65 | 0.55 | 0.85 | 1.48 | 0.97 |
| | Station | 0.7 | 0.61 | 0.83 | 1.68 | 0.97 |
| BR90 | ERA5 h | 0.67 | 1.05 | 1.2 | 1.49 | 1.03 |
| | ERA5 d | 0.75 | 1.15 | 1.29 | 1.3 | 0.99 |
| | RaKliDa | 0.72 | 1.07 | 1.36 | 1.59 | 1.14 |
| | Station | 0.72 | 1.11 | 1.22 | 1.84 | 1.1 |
| CBR90 | ERA5 h | 0.99 | 0.83 | 0.81 | 0.81 | 0.86 |
| | ERA5 d | 1.1 | 0.96 | 0.91 | 0.85 | 0.92 |
| | RaKliDa | 1.1 | 0.89 | 0.93 | 0.86 | 0.95 |
| | Station | 1.07 | 0.96 | 0.96 | 1.06 | 1.02 |

| | | | | | | |
|---|---|---|---|---|---|---|
| MAE | | | | | | |
| GBR90 | ERA5 h | 1.04 | 0.91 | 0.87 | 0.92 | 1.05 |
| | ERA5 d | 0.98 | 0.87 | 0.74 | 0.83 | 0.95 |
| | RaKliDa | 1.02 | 0.86 | 0.83 | 0.95 | 0.98 |
| | Station | 0.95 | 0.86 | 0.76 | 0.93 | 0.95 |
| EXTR | ERA5 h | 0.86 | 0.86 | 0.83 | 0.91 | 0.95 |
| | ERA5 d | 0.79 | 0.83 | 0.73 | 0.88 | 0.94 |
| | RaKliDa | 0.83 | 0.85 | 0.77 | 0.89 | 0.9 |
| | Station | 0.74 | 0.82 | 0.7 | 1.02 | 0.85 |
| BR90 | ERA5 h | 0.85 | 0.87 | 0.88 | 0.93 | 1.05 |
| | ERA5 d | 0.77 | 0.89 | 0.82 | 0.81 | 0.97 |
| | RaKliDa | 0.77 | 0.88 | 0.84 | 0.94 | 1.03 |
| | Station | 0.72 | 0.89 | 0.75 | 0.94 | 0.91 |
| CBR90 | ERA5 h | 0.68 | 0.88 | 0.8 | 0.63 | 0.87 |
| | ERA5 d | 0.63 | 0.85 | 0.7 | 0.58 | 0.83 |
| | RaKliDa | 0.59 | 0.81 | 0.72 | 0.76 | 0.87 |
| | Station | 0.53 | 0.8 | 0.65 | 0.61 | 0.77 |

659          Table E3. Daily evaporation skill-scores for the winter period

| Model/Station | | Grillenburg | Klingenberg | Hetzdorf | Tharandt | Oberbaerenburg |
|---|---|---|---|---|---|---|
| NSE | | | | | | |
| GBR90 | ERA5 h | -0.86 | -2.08 | -0.3 | -0.42 | -0.79 |
| | ERA5 d | -0.7 | -1.8 | -0.47 | -0.56 | -1.13 |
| | RaKliDa | -0.56 | -1.54 | -0.51 | -0.36 | -0.91 |
| | Station | -0.54 | -1.22 | -0.5 | -0.57 | -0.6 |
| EXTR | ERA5 h | -1.05 | -2.42 | -0.85 | -0.44 | -0.96 |
| | ERA5 d | -1.13 | -2.14 | -1.33 | -0.52 | -1.3 |
| | RaKliDa | -0.98 | -1.69 | -1.58 | -0.42 | -0.9 |
| | Station | -1.19 | -1.29 | -1.6 | -0.56 | -0.82 |
| BR90 | ERA5 h | -2.07 | -4.25 | -0.29 | -0.37 | -0.8 |
| | ERA5 d | -1.81 | -3.67 | -0.37 | -0.46 | -1.2 |
| | RaKliDa | -1.48 | -2.94 | -0.41 | -0.32 | -0.94 |
| | Station | -1.83 | -2.13 | -0.43 | -0.46 | -0.67 |
| CBR90 | ERA5 h | -0.26 | -1.5 | -0.16 | -0.61 | -1.16 |
| | ERA5 d | -0.21 | -1.4 | -0.41 | -0.66 | -1.93 |
| | RaKliDa | -0.08 | -1.23 | -0.4 | -0.83 | -1.34 |
| | Station | -0.05 | -0.96 | -0.64 | -0.34 | -1.6 |
| KGE | | | | | | |
| GBR90 | ERA5 h | 0.24 | -0.04 | 0.15 | -0.32 | -0.38 |
| | ERA5 d | 0.3 | 0.02 | 0.25 | -0.21 | -0.32 |
| | RaKliDa | 0.32 | 0.06 | 0.17 | -0.29 | -0.33 |
| | Station | 0.34 | 0.12 | 0.13 | -0.22 | -0.2 |
| EXTR | ERA5 h | 0.17 | -0.13 | 0.07 | -0.22 | -0.27 |
| | ERA5 d | 0.11 | -0.06 | -0.1 | -0.1 | -0.22 |
| | RaKliDa | 0.14 | 0.06 | -0.18 | -0.14 | -0.26 |
| | Station | 0.05 | 0.14 | -0.22 | -0.14 | -0.15 |
| BR90 | ERA5 h | -0.22 | -0.63 | 0.22 | -0.3 | -0.35 |
| | ERA5 d | -0.17 | -0.52 | 0.3 | -0.16 | -0.28 |

| | | | | | | |
|---|---|---|---|---|---|---|
| | RaKliDa | -0.06 | -0.32 | 0.24 | -0.26 | -0.28 |
| | Station | -0.2 | -0.16 | 0.19 | -0.19 | -0.17 |
| CBR90 | ERA5 h | 0.41 | 0.1 | 0.32 | 0.22 | -0.16 |
| | ERA5 d | 0.43 | 0.12 | 0.33 | 0.26 | -0.15 |
| | RaKliDa | 0.45 | 0.15 | 0.3 | 0.12 | -0.11 |
| | Station | 0.49 | 0.2 | 0.22 | 0.26 | -0.02 |
| Correlation | | | | | | |
| GBR90 | ERA5 h | 0.33 | 0.21 | 0.19 | 0.14 | -0.06 |
| | ERA5 d | 0.36 | 0.24 | 0.25 | 0.21 | -0.05 |
| | RaKliDa | 0.35 | 0.2 | 0.19 | 0.15 | -0.02 |
| | Station | 0.42 | 0.22 | 0.15 | 0.24 | 0.13 |
| EXTR | ERA5 h | 0.32 | 0.29 | 0.28 | 0.18 | -0.04 |
| | ERA5 d | 0.27 | 0.24 | 0.34 | 0.27 | -0.03 |
| | RaKliDa | 0.25 | 0.25 | 0.28 | 0.26 | -0.02 |
| | Station | 0.32 | 0.24 | 0.28 | 0.29 | 0.08 |
| BR90 | ERA5 h | 0.2 | 0.05 | 0.24 | 0.13 | -0.07 |
| | ERA5 d | 0.19 | 0.05 | 0.31 | 0.21 | -0.05 |
| | RaKliDa | 0.22 | 0.03 | 0.25 | 0.14 | -0.01 |
| | Station | 0.29 | 0.06 | 0.21 | 0.23 | 0.1 |
| CBR90 | ERA5 h | 0.42 | 0.26 | 0.34 | 0.22 | -0.05 |
| | ERA5 d | 0.44 | 0.29 | 0.37 | 0.28 | -0.03 |
| | RaKliDa | 0.46 | 0.26 | 0.34 | 0.15 | 0.02 |
| | Station | 0.5 | 0.27 | 0.3 | 0.28 | 0.11 |
| BIAS | | | | | | |
| GBR90 | ERA5 h | 0.85 | 1.15 | 1.01 | 3.45 | 3.92 |
| | ERA5 d | 0.9 | 1.23 | 0.92 | 3.15 | 3.69 |
| | RaKliDa | 0.94 | 1.29 | 0.88 | 3.13 | 3.97 |
| | Station | 0.83 | 1.3 | 0.9 | 3.46 | 3.59 |
| EXTR | ERA5 h | 0.76 | 0.85 | 0.63 | 2.91 | 2.97 |
| | ERA5 d | 0.71 | 0.83 | 0.55 | 2.72 | 2.83 |
| | RaKliDa | 0.74 | 0.95 | 0.53 | 2.79 | 3.11 |
| | Station | 0.65 | 0.98 | 0.51 | 3.1 | 2.91 |
| BR90 | ERA5 h | 0.57 | 0.56 | 0.97 | 3.15 | 3.49 |
| | ERA5 d | 0.59 | 0.57 | 0.9 | 2.75 | 3.16 |
| | RaKliDa | 0.62 | 0.64 | 0.88 | 2.76 | 3.46 |
| | Station | 0.56 | 0.69 | 0.9 | 3.01 | 3.11 |
| CBR90 | ERA5 h | 1.05 | 1.12 | 0.96 | 1.01 | 2 |
| | ERA5 d | 1 | 1.11 | 0.81 | 0.98 | 1.78 |
| | RaKliDa | 1.1 | 1.2 | 0.8 | 0.82 | 2.01 |
| | Station | 0.96 | 1.24 | 0.75 | 1.21 | 1.62 |
| Variance ratio | | | | | | |
| GBR90 | ERA5 h | 0.59 | 0.36 | 1.7 | 11.57 | 3.47 |
| | ERA5 d | 0.63 | 0.4 | 1.05 | 6.56 | 2.15 |
| | RaKliDa | 0.73 | 0.49 | 1.19 | 10.35 | 2.86 |
| | Station | 0.65 | 0.57 | 1.35 | 7.88 | 2.87 |
| EXTR | ERA5 h | 0.54 | 0.29 | 0.85 | 6.8 | 1.88 |
| | ERA5 d | 0.57 | 0.34 | 0.61 | 4.38 | 1.26 |
| | RaKliDa | 0.65 | 0.41 | 0.6 | 5.53 | 2.02 |
| | Station | 0.52 | 0.51 | 0.61 | 5.61 | 1.74 |

| | | | | | | |
|---|---|---|---|---|---|---|
| BR90 | ERA5 h | 0.42 | 0.24 | 1.43 | 10.51 | 2.91 |
| | ERA5 d | 0.47 | 0.27 | 1.03 | 5.42 | 1.64 |
| | RaKliDa | 0.53 | 0.34 | 1.17 | 8.52 | 2.23 |
| | Station | 0.42 | 0.45 | 1.27 | 6.6 | 2.21 |
| CBR90 | ERA5 h | 0.86 | 0.44 | 1.37 | 0.93 | 1.1 |
| | ERA5 d | 0.86 | 0.44 | 0.88 | 0.78 | 0.6 |
| | RaKliDa | 1.02 | 0.52 | 0.98 | 0.89 | 0.86 |
| | Station | 0.92 | 0.62 | 0.86 | 1.22 | 0.56 |
| MAE | | | | | | |
| GBR90 | ERA5 h | 0.19 | 0.23 | 0.36 | 0.75 | 0.8 |
| | ERA5 d | 0.19 | 0.23 | 0.32 | 0.67 | 0.74 |
| | RaKliDa | 0.19 | 0.24 | 0.34 | 0.73 | 0.78 |
| | Station | 0.18 | 0.24 | 0.36 | 0.74 | 0.69 |
| EXTR | ERA5 h | 0.19 | 0.19 | 0.31 | 0.61 | 0.61 |
| | ERA5 d | 0.2 | 0.2 | 0.31 | 0.55 | 0.58 |
| | RaKliDa | 0.2 | 0.2 | 0.32 | 0.57 | 0.64 |
| | Station | 0.19 | 0.21 | 0.32 | 0.65 | 0.57 |
| BR90 | ERA5 h | 0.21 | 0.21 | 0.33 | 0.69 | 0.72 |
| | ERA5 d | 0.21 | 0.22 | 0.3 | 0.58 | 0.65 |
| | RaKliDa | 0.21 | 0.22 | 0.33 | 0.66 | 0.69 |
| | Station | 0.2 | 0.23 | 0.34 | 0.64 | 0.62 |
| CBR90 | ERA5 h | 0.2 | 0.22 | 0.31 | 0.25 | 0.45 |
| | ERA5 d | 0.19 | 0.21 | 0.28 | 0.24 | 0.41 |
| | RaKliDa | 0.19 | 0.22 | 0.29 | 0.26 | 0.44 |
| | Station | 0.18 | 0.23 | 0.3 | 0.27 | 0.36 |

Table E4. Monthly evaporation skill-scores for the whole year

| Model/Station | | Grillenburg | Klingenberg | Hetzdorf | Tharandt | Oberbaerenburg |
|---|---|---|---|---|---|---|
| NSE | | | | | | |
| GBR90 | ERA5 h | 0.37 | 0.56 | 0.74 | 0.44 | 0.49 |
| | ERA5 d | 0.49 | 0.65 | 0.84 | 0.57 | 0.59 |
| | RaKliDa | 0.37 | 0.59 | 0.78 | 0.54 | 0.54 |
| | Station | 0.4 | 0.56 | 0.77 | 0.47 | 0.55 |
| EXTR | ERA5 h | 0.63 | 0.61 | 0.84 | 0.59 | 0.7 |
| | ERA5 d | 0.74 | 0.68 | 0.88 | 0.61 | 0.71 |
| | RaKliDa | 0.66 | 0.55 | 0.88 | 0.63 | 0.72 |
| | Station | 0.72 | 0.6 | 0.89 | 0.48 | 0.75 |
| BR90 | ERA5 h | 0.65 | 0.77 | 0.89 | 0.57 | 0.63 |
| | ERA5 d | 0.84 | 0.77 | 0.88 | 0.69 | 0.69 |
| | RaKliDa | 0.8 | 0.74 | 0.88 | 0.67 | 0.63 |
| | Station | 0.81 | 0.72 | 0.9 | 0.6 | 0.72 |
| CBR90 | ERA5 h | 0.93 | 0.83 | 0.9 | 0.84 | 0.84 |
| | ERA5 d | 0.92 | 0.79 | 0.92 | 0.9 | 0.85 |
| | RaKliDa | 0.93 | 0.81 | 0.92 | 0.67 | 0.83 |
| | Station | 0.93 | 0.79 | 0.93 | 0.91 | 0.87 |
| KGE | | | | | | |
| GBR90 | ERA5 h | 0.41 | 0.68 | 0.66 | 0.67 | 0.65 |
| | ERA5 d | 0.51 | 0.79 | 0.79 | 0.71 | 0.69 |

| | | | | | | |
|---|---|---|---|---|---|---|
| | RaKliDa | 0.43 | 0.72 | 0.71 | 0.69 | 0.66 |
| | Station | 0.44 | 0.72 | 0.72 | 0.67 | 0.67 |
| EXTR | ERA5 h | 0.54 | 0.71 | 0.86 | 0.71 | 0.74 |
| | ERA5 d | 0.65 | 0.82 | 0.94 | 0.69 | 0.74 |
| | RaKliDa | 0.57 | 0.7 | 0.91 | 0.73 | 0.75 |
| | Station | 0.62 | 0.75 | 0.92 | 0.67 | 0.77 |
| BR90 | ERA5 h | 0.54 | 0.8 | 0.94 | 0.72 | 0.71 |
| | ERA5 d | 0.76 | 0.82 | 0.84 | 0.74 | 0.72 |
| | RaKliDa | 0.7 | 0.8 | 0.89 | 0.76 | 0.72 |
| | Station | 0.7 | 0.8 | 0.91 | 0.7 | 0.77 |
| CBR90 | ERA5 h | 0.96 | 0.9 | 0.89 | 0.82 | 0.83 |
| | ERA5 d | 0.82 | 0.79 | 0.94 | 0.91 | 0.8 |
| | RaKliDa | 0.85 | 0.86 | 0.95 | 0.65 | 0.84 |
| | Station | 0.88 | 0.85 | 0.96 | 0.95 | 0.91 |
| Correlation | | | | | | |
| GBR90 | ERA5 h | 0.92 | 0.86 | 0.96 | 0.91 | 0.91 |
| | ERA5 d | 0.91 | 0.86 | 0.95 | 0.94 | 0.94 |
| | RaKliDa | 0.91 | 0.85 | 0.96 | 0.89 | 0.93 |
| | Station | 0.91 | 0.84 | 0.95 | 0.94 | 0.89 |
| EXTR | ERA5 h | 0.95 | 0.87 | 0.94 | 0.93 | 0.94 |
| | ERA5 d | 0.95 | 0.86 | 0.94 | 0.93 | 0.94 |
| | RaKliDa | 0.95 | 0.85 | 0.95 | 0.92 | 0.94 |
| | Station | 0.95 | 0.85 | 0.95 | 0.89 | 0.93 |
| BR90 | ERA5 h | 0.96 | 0.9 | 0.95 | 0.92 | 0.93 |
| | ERA5 d | 0.96 | 0.88 | 0.95 | 0.94 | 0.94 |
| | RaKliDa | 0.96 | 0.88 | 0.94 | 0.9 | 0.91 |
| | Station | 0.96 | 0.87 | 0.95 | 0.93 | 0.92 |
| CBR90 | ERA5 h | 0.97 | 0.92 | 0.96 | 0.95 | 0.95 |
| | ERA5 d | 0.98 | 0.91 | 0.96 | 0.95 | 0.95 |
| | RaKliDa | 0.98 | 0.91 | 0.96 | 0.93 | 0.94 |
| | Station | 0.97 | 0.9 | 0.96 | 0.96 | 0.94 |
| BIAS | | | | | | |
| GBR90 | ERA5 h | 0.69 | 0.84 | 0.85 | 1.38 | 1.37 |
| | ERA5 d | 0.72 | 0.91 | 0.89 | 1.39 | 1.4 |
| | RaKliDa | 0.7 | 0.87 | 0.84 | 1.22 | 1.35 |
| | Station | 0.7 | 0.87 | 0.85 | 1.49 | 1.23 |
| EXTR | ERA5 h | 0.73 | 0.88 | 0.94 | 1.4 | 1.31 |
| | ERA5 d | 0.77 | 0.94 | 0.99 | 1.42 | 1.35 |
| | RaKliDa | 0.73 | 0.87 | 0.95 | 1.34 | 1.26 |
| | Station | 0.75 | 0.9 | 0.95 | 1.44 | 1.21 |
| BR90 | ERA5 h | 0.73 | 0.86 | 1.03 | 1.36 | 1.37 |
| | ERA5 d | 0.83 | 0.94 | 1.1 | 1.34 | 1.38 |
| | RaKliDa | 0.8 | 0.87 | 1.05 | 1.17 | 1.31 |
| | Station | 0.8 | 0.87 | 1.04 | 1.41 | 1.21 |
| CBR90 | ERA5 h | 0.99 | 1.06 | 0.99 | 0.9 | 1.19 |
| | ERA5 d | 1.13 | 1.16 | 1.03 | 0.94 | 1.23 |
| | RaKliDa | 1.11 | 1.09 | 0.98 | 0.78 | 1.16 |
| | Station | 1.07 | 1.09 | 0.98 | 1.02 | 1.06 |
| Variance ratio | | | | | | |

| | | | | | | |
|---|---|---|---|---|---|---|
| GBR90 | ERA5 h | 0.53 | 0.67 | 0.61 | 0.75 | 0.69 |
| | ERA5 d | 0.6 | 0.8 | 0.74 | 0.91 | 0.81 |
| | RaKliDa | 0.54 | 0.71 | 0.68 | 0.66 | 0.68 |
| | Station | 0.56 | 0.73 | 0.68 | 1 | 0.64 |
| EXTR | ERA5 h | 0.62 | 0.68 | 0.81 | 1.01 | 0.84 |
| | ERA5 d | 0.73 | 0.82 | 1 | 1.16 | 0.97 |
| | RaKliDa | 0.66 | 0.68 | 0.92 | 0.98 | 0.78 |
| | Station | 0.71 | 0.72 | 0.94 | 1.09 | 0.76 |
| BR90 | ERA5 h | 0.61 | 1.03 | 1.03 | 0.88 | 0.87 |
| | ERA5 d | 0.82 | 1.28 | 1.3 | 1.03 | 0.99 |
| | RaKliDa | 0.75 | 1.1 | 1.19 | 0.75 | 0.8 |
| | Station | 0.76 | 1.1 | 1.13 | 1.1 | 0.8 |
| CBR90 | ERA5 h | 0.97 | 1.01 | 0.83 | 0.79 | 0.95 |
| | ERA5 d | 1.36 | 1.32 | 1.06 | 0.9 | 1.12 |
| | RaKliDa | 1.28 | 1.12 | 0.95 | 0.7 | 0.91 |
| | Station | 1.23 | 1.15 | 1.01 | 0.98 | 0.93 |
| MAE | | | | | | |
| GBR90 | ERA5 h | 17.04 | 13.93 | 11.7 | 16.25 | 16.99 |
| | ERA5 d | 15.94 | 13.78 | 9.95 | 16.05 | 16.91 |
| | RaKliDa | 17.17 | 14.09 | 11.05 | 13.05 | 16.15 |
| | Station | 16.9 | 14.71 | 11.22 | 19.56 | 15.01 |
| EXTR | ERA5 h | 15.12 | 13.21 | 10.08 | 16.85 | 14.43 |
| | ERA5 d | 13.59 | 13.37 | 9.82 | 17.6 | 15.15 |
| | RaKliDa | 14.75 | 14.32 | 9.69 | 15.5 | 13.14 |
| | Station | 13.77 | 13.93 | 9.32 | 19.99 | 12.26 |
| BR90 | ERA5 h | 14.6 | 12.81 | 9.48 | 15.45 | 16.49 |
| | ERA5 d | 11.31 | 13.91 | 11.25 | 14.38 | 15.96 |
| | RaKliDa | 12.11 | 14.09 | 10.67 | 11.8 | 15.29 |
| | Station | 11.86 | 14.47 | 9.8 | 17.32 | 13.02 |
| CBR90 | ERA5 h | 7.08 | 10.51 | 8.36 | 7.7 | 10.74 |
| | ERA5 d | 9.12 | 12.59 | 8.39 | 6.69 | 11.16 |
| | RaKliDa | 8.24 | 11.56 | 8.01 | 10.93 | 10.51 |
| | Station | 7.9 | 12.11 | 7.9 | 6.35 | 8.85 |

661        Table E5. Monthly evaporation skill-scores for the vegetation period

| Model/Station | | Grillenburg | Klingenberg | Hetzdorf | Tharandt | Oberbaerenburg |
|---|---|---|---|---|---|---|
| NSE | | | | | | |
| GBR90 | ERA5 h | -0.18 | 0.23 | 0.5 | 0.32 | 0.3 |
| | ERA5 d | 0.07 | 0.4 | 0.69 | 0.4 | 0.41 |
| | RaKliDa | -0.14 | 0.3 | 0.58 | 0.57 | 0.43 |
| | Station | -0.1 | 0.27 | 0.56 | 0.22 | 0.48 |
| EXTR | ERA5 h | 0.3 | 0.17 | 0.59 | 0.3 | 0.5 |
| | ERA5 d | 0.54 | 0.35 | 0.71 | 0.29 | 0.49 |
| | RaKliDa | 0.39 | 0.11 | 0.72 | 0.42 | 0.65 |
| | Station | 0.49 | 0.21 | 0.74 | 0.13 | 0.68 |
| BR90 | ERA5 h | 0.29 | 0.64 | 0.78 | 0.45 | 0.48 |
| | ERA5 d | 0.69 | 0.65 | 0.75 | 0.55 | 0.53 |
| | RaKliDa | 0.62 | 0.62 | 0.77 | 0.68 | 0.51 |

| | | | | | | |
|---|---|---|---|---|---|---|
| | Station | 0.63 | 0.59 | 0.81 | 0.41 | 0.68 |
| CBR90 | ERA5 h | 0.86 | 0.66 | 0.75 | 0.68 | 0.72 |
| | ERA5 d | 0.83 | 0.61 | 0.83 | 0.79 | 0.71 |
| | RaKliDa | 0.86 | 0.65 | 0.84 | 0.39 | 0.7 |
| | Station | 0.86 | 0.62 | 0.86 | 0.83 | 0.8 |
| KGE | | | | | | |
| GBR90 | ERA5 h | 0.45 | 0.63 | 0.62 | 0.72 | 0.62 |
| | ERA5 d | 0.54 | 0.71 | 0.76 | 0.77 | 0.7 |
| | RaKliDa | 0.47 | 0.66 | 0.69 | 0.78 | 0.66 |
| | Station | 0.48 | 0.65 | 0.7 | 0.73 | 0.69 |
| EXTR | ERA5 h | 0.59 | 0.58 | 0.63 | 0.74 | 0.72 |
| | ERA5 d | 0.68 | 0.69 | 0.78 | 0.72 | 0.75 |
| | RaKliDa | 0.61 | 0.58 | 0.74 | 0.76 | 0.76 |
| | Station | 0.66 | 0.62 | 0.77 | 0.67 | 0.79 |
| BR90 | ERA5 h | 0.57 | 0.76 | 0.89 | 0.78 | 0.74 |
| | ERA5 d | 0.78 | 0.72 | 0.82 | 0.78 | 0.76 |
| | RaKliDa | 0.73 | 0.72 | 0.85 | 0.84 | 0.75 |
| | Station | 0.74 | 0.71 | 0.89 | 0.73 | 0.82 |
| CBR90 | ERA5 h | 0.93 | 0.83 | 0.75 | 0.81 | 0.82 |
| | ERA5 d | 0.79 | 0.75 | 0.9 | 0.89 | 0.82 |
| | RaKliDa | 0.82 | 0.8 | 0.87 | 0.67 | 0.83 |
| | Station | 0.85 | 0.78 | 0.91 | 0.91 | 0.9 |
| Correlation | | | | | | |
| GBR90 | ERA5 h | 0.85 | 0.75 | 0.92 | 0.87 | 0.88 |
| | ERA5 d | 0.83 | 0.74 | 0.92 | 0.91 | 0.91 |
| | RaKliDa | 0.83 | 0.74 | 0.92 | 0.84 | 0.89 |
| | Station | 0.83 | 0.72 | 0.91 | 0.9 | 0.84 |
| EXTR | ERA5 h | 0.91 | 0.75 | 0.9 | 0.87 | 0.91 |
| | ERA5 d | 0.91 | 0.74 | 0.9 | 0.87 | 0.91 |
| | RaKliDa | 0.91 | 0.7 | 0.91 | 0.85 | 0.91 |
| | Station | 0.91 | 0.71 | 0.91 | 0.79 | 0.89 |
| BR90 | ERA5 h | 0.93 | 0.83 | 0.9 | 0.88 | 0.9 |
| | ERA5 d | 0.92 | 0.81 | 0.9 | 0.91 | 0.91 |
| | RaKliDa | 0.92 | 0.81 | 0.89 | 0.85 | 0.86 |
| | Station | 0.93 | 0.79 | 0.91 | 0.88 | 0.88 |
| CBR90 | ERA5 h | 0.93 | 0.84 | 0.92 | 0.89 | 0.92 |
| | ERA5 d | 0.95 | 0.83 | 0.93 | 0.91 | 0.92 |
| | RaKliDa | 0.95 | 0.83 | 0.93 | 0.87 | 0.89 |
| | Station | 0.94 | 0.81 | 0.93 | 0.91 | 0.91 |
| BIAS | | | | | | |
| GBR90 | ERA5 h | 0.68 | 0.83 | 0.83 | 1.22 | 1.22 |
| | ERA5 d | 0.72 | 0.9 | 0.88 | 1.26 | 1.27 |
| | RaKliDa | 0.68 | 0.85 | 0.84 | 1.07 | 1.2 |
| | Station | 0.69 | 0.85 | 0.84 | 1.34 | 1.1 |
| EXTR | ERA5 h | 0.73 | 0.88 | 0.97 | 1.29 | 1.22 |
| | ERA5 d | 0.77 | 0.94 | 1.03 | 1.32 | 1.26 |
| | RaKliDa | 0.73 | 0.87 | 0.99 | 1.23 | 1.15 |
| | Station | 0.76 | 0.9 | 1 | 1.32 | 1.11 |
| BR90 | ERA5 h | 0.74 | 0.87 | 1.04 | 1.23 | 1.25 |

| | | | | | |
|---|---|---|---|---|---|
| | ERA5 d | 0.84 | 0.96 | 1.12 | 1.24 | 1.27 |
| | RaKliDa | 0.81 | 0.88 | 1.07 | 1.05 | 1.18 |
| | Station | 0.81 | 0.88 | 1.06 | 1.29 | 1.1 |
| CBR90 | ERA5 h | 0.99 | 1.06 | 0.99 | 0.89 | 1.15 |
| | ERA5 d | 1.13 | 1.17 | 1.05 | 0.94 | 1.2 |
| | RaKliDa | 1.11 | 1.08 | 1 | 0.78 | 1.11 |
| | Station | 1.07 | 1.08 | 1 | 1.01 | 1.03 |
| Variance ratio | | | | | | |
| GBR90 | ERA5 h | 0.64 | 0.71 | 0.58 | 0.73 | 0.58 |
| | ERA5 d | 0.74 | 0.86 | 0.72 | 0.91 | 0.7 |
| | RaKliDa | 0.69 | 0.79 | 0.67 | 0.77 | 0.61 |
| | Station | 0.69 | 0.82 | 0.67 | 1.03 | 0.64 |
| EXTR | ERA5 h | 0.73 | 0.59 | 0.55 | 0.97 | 0.7 |
| | ERA5 d | 0.9 | 0.75 | 0.7 | 1.14 | 0.82 |
| | RaKliDa | 0.82 | 0.64 | 0.65 | 1.01 | 0.73 |
| | Station | 0.87 | 0.68 | 0.68 | 1.15 | 0.75 |
| BR90 | ERA5 h | 0.67 | 1.23 | 0.95 | 0.91 | 0.78 |
| | ERA5 d | 0.91 | 1.55 | 1.23 | 1.1 | 0.91 |
| | RaKliDa | 0.85 | 1.37 | 1.15 | 0.92 | 0.78 |
| | Station | 0.84 | 1.37 | 1.07 | 1.22 | 0.86 |
| CBR90 | ERA5 h | 0.97 | 0.93 | 0.66 | 0.83 | 0.84 |
| | ERA5 d | 1.42 | 1.29 | 0.9 | 0.95 | 1.01 |
| | RaKliDa | 1.35 | 1.11 | 0.81 | 0.84 | 0.84 |
| | Station | 1.31 | 1.17 | 0.89 | 1.03 | 0.96 |
| MAE | | | | | | |
| GBR90 | ERA5 h | 24.02 | 18.64 | 15.24 | 14.33 | 15.87 |
| | ERA5 d | 22.44 | 18.33 | 12.84 | 15.23 | 16.54 |
| | RaKliDa | 24.23 | 18.65 | 14.4 | 10.68 | 14.7 |
| | Station | 23.78 | 19.65 | 14.38 | 19.24 | 13.97 |
| EXTR | ERA5 h | 20.8 | 18.05 | 12.28 | 17.44 | 14.9 |
| | ERA5 d | 18.27 | 18.12 | 11.52 | 19.34 | 16.42 |
| | RaKliDa | 20.12 | 19.52 | 11.16 | 15.93 | 12.55 |
| | Station | 18.67 | 18.98 | 10.45 | 21.39 | 11.88 |
| BR90 | ERA5 h | 19.72 | 17.03 | 12.24 | 14.3 | 16.36 |
| | ERA5 d | 14.77 | 18.62 | 15 | 14.32 | 16.64 |
| | RaKliDa | 15.99 | 18.86 | 13.95 | 10.23 | 14.91 |
| | Station | 15.58 | 19.57 | 12.45 | 17.71 | 12.43 |
| CBR90 | ERA5 h | 9.07 | 13.66 | 10.68 | 9.82 | 11.91 |
| | ERA5 d | 12.11 | 16.86 | 10.55 | 8.35 | 13.09 |
| | RaKliDa | 10.8 | 15.2 | 9.89 | 14.54 | 11.77 |
| | Station | 10.35 | 16.02 | 9.58 | 7.76 | 10.19 |

Table E6. Monthly evaporation skill-scores for the winter period

| Model/Station | | Grillenburg | Klingenberg | Hetzdorf | Tharandt | Oberbaerenburg |
|---|---|---|---|---|---|---|
| NSE | | | | | | |
| GBR90 | ERA5 h | -0.84 | -3.36 | -0.21 | -3.65 | -3.23 |
| | ERA5 d | -0.62 | -2.97 | -0.56 | -4.55 | -4.59 |
| | RaKliDa | -0.48 | -2.77 | -0.88 | -3.28 | -4.82 |

|  | Station | -0.46 | -2.6 | -1.21 | -6.21 | -4.03 |
|---|---|---|---|---|---|---|
| EXTR | ERA5 h | -4.44 | -5.59 | -2.96 | -3.47 | -3.15 |
|  | ERA5 d | -4.71 | -6.57 | -4.39 | -3.68 | -3.9 |
|  | RaKliDa | -3.93 | -5.71 | -4.81 | -3.62 | -3.5 |
|  | Station | -4.19 | -4.8 | -4.49 | -5.1 | -3.8 |
| BR90 | ERA5 h | -8.08 | -16.29 | -0.02 | -3.13 | -3 |
|  | ERA5 d | -7.88 | -14.62 | -0.18 | -3.66 | -4.2 |
|  | RaKliDa | -6.26 | -9.67 | -0.45 | -2.75 | -4.27 |
|  | Station | -6.69 | -7.49 | -0.91 | -4.85 | -3.74 |
| CBR90 | ERA5 h | -0.4 | -1.97 | 0.27 | -0.86 | -1.95 |
|  | ERA5 d | -0.49 | -2.02 | -0.21 | -0.83 | -2.61 |
|  | RaKliDa | -0.35 | -2.27 | -0.23 | -2.12 | -2.36 |
|  | Station | -0.22 | -2.08 | -0.96 | -0.45 | -2.65 |
| KGE |  |  |  |  |  |  |
| GBR90 | ERA5 h | 0.27 | -0.3 | 0.32 | -0.32 | -0.32 |
|  | ERA5 d | 0.33 | -0.21 | 0.35 | -0.22 | -0.28 |
|  | RaKliDa | 0.39 | -0.15 | 0.27 | -0.34 | -0.2 |
|  | Station | 0.4 | -0.11 | 0.09 | -0.16 | -0.27 |
| EXTR | ERA5 h | -0.45 | -0.86 | 0.02 | -0.17 | -0.16 |
|  | ERA5 d | -0.44 | -0.97 | -0.17 | -0.08 | -0.14 |
|  | RaKliDa | -0.33 | -0.8 | -0.26 | -0.02 | -0.23 |
|  | Station | -0.35 | -0.66 | -0.3 | -0.02 | -0.18 |
| BR90 | ERA5 h | -0.84 | -1.98 | 0.47 | -0.29 | -0.27 |
|  | ERA5 d | -0.82 | -1.8 | 0.48 | -0.16 | -0.23 |
|  | RaKliDa | -0.63 | -1.2 | 0.4 | -0.3 | -0.15 |
|  | Station | -0.68 | -0.95 | 0.22 | -0.09 | -0.23 |
| CBR90 | ERA5 h | 0.42 | -0.01 | 0.58 | 0.27 | -0.05 |
|  | ERA5 d | 0.38 | -0.04 | 0.49 | 0.28 | -0.07 |
|  | RaKliDa | 0.44 | -0.07 | 0.47 | 0 | 0.05 |
|  | Station | 0.47 | -0.02 | 0.29 | 0.42 | -0.08 |
| Correlation |  |  |  |  |  |  |
| GBR90 | ERA5 h | 0.54 | 0.2 | 0.33 | 0.05 | 0 |
|  | ERA5 d | 0.56 | 0.23 | 0.37 | 0.1 | -0.01 |
|  | RaKliDa | 0.51 | 0.15 | 0.31 | -0.01 | 0.11 |
|  | Station | 0.55 | 0.21 | 0.11 | 0.21 | 0 |
| EXTR | ERA5 h | 0.27 | 0.39 | 0.3 | 0.16 | 0.07 |
|  | ERA5 d | 0.06 | 0.29 | 0.29 | 0.22 | 0.06 |
|  | RaKliDa | 0.16 | 0.23 | 0.2 | 0.33 | -0.01 |
|  | Station | 0.29 | 0.29 | 0.17 | 0.35 | 0.03 |
| BR90 | ERA5 h | 0.21 | 0.06 | 0.47 | 0.07 | 0.01 |
|  | ERA5 d | 0.17 | 0.03 | 0.5 | 0.13 | -0.01 |
|  | RaKliDa | 0.15 | -0.11 | 0.42 | -0.01 | 0.12 |
|  | Station | 0.25 | 0.01 | 0.25 | 0.24 | -0.01 |
| CBR90 | ERA5 h | 0.52 | 0.32 | 0.6 | 0.35 | 0.07 |
|  | ERA5 d | 0.52 | 0.36 | 0.55 | 0.37 | 0.07 |
|  | RaKliDa | 0.55 | 0.29 | 0.53 | 0.24 | 0.21 |
|  | Station | 0.56 | 0.33 | 0.39 | 0.46 | 0.09 |
| BIAS |  |  |  |  |  |  |
| GBR90 | ERA5 h | 0.85 | 1.15 | 1.01 | 3.45 | 3.93 |

| | | | | | |
|---|---|---|---|---|---|
| | ERA5 d | 0.9 | 1.23 | 0.92 | 3.16 | 3.69 |
| | RaKliDa | 0.94 | 1.29 | 0.88 | 3.14 | 3.97 |
| | Station | 0.83 | 1.3 | 0.9 | 3.46 | 3.59 |
| EXTR | ERA5 h | 0.76 | 0.85 | 0.63 | 2.91 | 2.97 |
| | ERA5 d | 0.71 | 0.83 | 0.54 | 2.72 | 2.83 |
| | RaKliDa | 0.74 | 0.95 | 0.53 | 2.79 | 3.11 |
| | Station | 0.65 | 0.98 | 0.51 | 3.1 | 2.91 |
| BR90 | ERA5 h | 0.57 | 0.55 | 0.97 | 3.15 | 3.49 |
| | ERA5 d | 0.59 | 0.57 | 0.9 | 2.76 | 3.16 |
| | RaKliDa | 0.63 | 0.64 | 0.88 | 2.76 | 3.47 |
| | Station | 0.55 | 0.69 | 0.9 | 3.01 | 3.11 |
| CBR90 | ERA5 h | 1.05 | 1.12 | 0.96 | 1.01 | 2 |
| | ERA5 d | 1 | 1.11 | 0.81 | 0.98 | 1.78 |
| | RaKliDa | 1.1 | 1.2 | 0.8 | 0.82 | 2.01 |
| | Station | 0.96 | 1.24 | 0.75 | 1.21 | 1.62 |
| Variance ratio | | | | | | |
| GBR90 | ERA5 h | 0.42 | 0.24 | 1.27 | 5.85 | 3.09 |
| | ERA5 d | 0.45 | 0.28 | 0.73 | 3.64 | 1.88 |
| | RaKliDa | 0.54 | 0.33 | 0.68 | 5.09 | 2.07 |
| | Station | 0.56 | 0.33 | 0.74 | 3.39 | 2 |
| EXTR | ERA5 h | 0.2 | 0.13 | 0.55 | 3.71 | 1.55 |
| | ERA5 d | 0.24 | 0.13 | 0.5 | 2.83 | 1.12 |
| | RaKliDa | 0.26 | 0.15 | 0.51 | 2.99 | 1.63 |
| | Station | 0.25 | 0.16 | 0.58 | 2.92 | 1.25 |
| BR90 | ERA5 h | 0.16 | 0.07 | 1.08 | 5.28 | 2.5 |
| | ERA5 d | 0.17 | 0.08 | 0.84 | 3.05 | 1.42 |
| | RaKliDa | 0.2 | 0.13 | 0.8 | 4.22 | 1.66 |
| | Station | 0.19 | 0.15 | 0.72 | 2.91 | 1.53 |
| CBR90 | ERA5 h | 0.57 | 0.33 | 1.3 | 0.56 | 0.97 |
| | ERA5 d | 0.52 | 0.3 | 0.96 | 0.55 | 0.59 |
| | RaKliDa | 0.57 | 0.32 | 1.01 | 0.38 | 0.73 |
| | Station | 0.61 | 0.33 | 0.83 | 0.76 | 0.48 |
| MAE | | | | | | |
| GBR90 | ERA5 h | 3.08 | 4.51 | 4.6 | 20.09 | 19.24 |
| | ERA5 d | 2.95 | 4.69 | 4.17 | 17.68 | 17.65 |
| | RaKliDa | 3.04 | 4.97 | 4.34 | 17.78 | 19.04 |
| | Station | 3.13 | 4.83 | 4.92 | 20.19 | 17.09 |
| EXTR | ERA5 h | 3.77 | 3.54 | 5.67 | 15.67 | 13.5 |
| | ERA5 d | 4.23 | 3.86 | 6.42 | 14.12 | 12.6 |
| | RaKliDa | 4.03 | 3.92 | 6.77 | 14.66 | 14.33 |
| | Station | 3.96 | 3.82 | 7.07 | 17.21 | 13.04 |
| BR90 | ERA5 h | 4.36 | 4.36 | 3.96 | 17.76 | 16.74 |
| | ERA5 d | 4.39 | 4.48 | 3.76 | 14.49 | 14.61 |
| | RaKliDa | 4.33 | 4.55 | 4.11 | 14.92 | 16.07 |
| | Station | 4.42 | 4.27 | 4.49 | 16.53 | 14.22 |
| CBR90 | ERA5 h | 3.1 | 4.21 | 3.72 | 3.46 | 8.4 |
| | ERA5 d | 3.14 | 4.05 | 4.07 | 3.38 | 7.31 |
| | RaKliDa | 3.13 | 4.28 | 4.23 | 3.71 | 8 |
| | Station | 2.99 | 4.27 | 4.54 | 3.53 | 6.18 |

## Data and Code availability

Authors fully support open-source and reproducible research. Therefore, all the data and codes are available as Supplementary material under the following HydroShare composite resource https://doi.org/10.4211/hs.567d7bdc7b84465ca333b6e0c011853a , which include:

- Raw eddy-covariance and meteorological measurement daily data with location files
- Raw results of model runs for each framework, including model calibration and FAO simulations
- R-scripts to reproduce figures and tables for the manuscript

In addition, Global BROOK90 framework is available under https://github.com/hydrovorobey/Global_BROOK90, EXTRUSO framework is available under https://github.com/GeoinformationSystems/xtruso_R, and BROOK90 R-version is available under https://github.com/rkronen/Brook90_R.

## Author contribution

Conceptualization VI, LTT and KR; data curation GT, LTT and VI, formal analysis VI, funding acquisition BC, methodology VI, LTT and KR; supervision KR; visualization VI; writing: original draft preparation VI and LTT, writing: review KR, GT, BC.

## Competing interests

The authors declare that they have no conflict of interest.

## Acknowledgements

Authors would like to express great thanks to Uwe Spank for his valuable advises and comments to the paper draft. Additionally, authors thank BMBF for providing the funding opportunities for the study under the scope of the 'KlimaKonform' project.

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
