# Peer review of "Modelling evaporation with local, regional and global BROOK90 frameworks: importance of parameterization and forcing."

_Hydrology and Earth System Sciences, 2021_

## Author Comment (AC2)

Answers to comments of Reviewer 2

| № | Comment | Answer |
|---|---------|--------|
| 1 | Parameter selection and parametrization is a central issue in the paper, but information about the parameters is mainly lacking. The cited literature for GBR90 (Vorobewskii et al. 2020) and for EXTR (Luong et al. 2020) list various sources for parameter groups without stating parameter values, too. Please include a table with the relevant parameters and their values which differ due to different soil and landcover input. | Agreed, the list of the parameters as well as their ranges (different from the default model ones) for each framework will be listed in extra Appendix. |
| 2 | The final values of the calibrated parameters and for comparison, the parameter values for the other model set-ups are lacking. | Agreed, Appendix C will be extended. |
| 3 | The concept of uncertainties in the paper is not clear. The reader would expect as a result confidence limits for the parameters and model outcome, which is not given. The authors should make clearer what they intent. | Agreed, will be elaborated. The main intention of the study was not to make a detailed assessment of the model's parameters and forcing uncertainty. Rather we want to address the topic mentioned in the main title. Namely, how available parameterization schemes, meteo input, and their scales influence BROOK90 performance regarding evaporation simulations. Thus, we suggest to rephrase/omit 'uncertainty' term confusement or use it with caution, pointing out that we did not present a quantitative 'meteo and parameter uncertainty' evaluation and elaborate the last two paragraphs in the intro. |
| 4 | In the discussion section main parts of the results, e.g., parametrization, are not discussed and new results are presented instead. The discussion nearly comes out without referring results from other researchers; therefore, the authors do not give proper credit to related work. | Agreed, the discussion section will be reorganized and discussion of the initial results will be elaborated. |
| 5 | The abstract does not contain results and a final outcome of the paper | Agreed, will be elaborated. |
| 6 | Elaborating the introduction, work out a hypothesis and state it at the end of the introduction (and not within the method section) | Agreed, will be elaborated. |
| 7 | Reorganize Data & Methods. Why not using traditional Material & methods – section? I suggest lifting "2.1 Eddy-covariance measurements" in the hierarchy and to do not subsume it under "Data", it is a central issue of the paper. When you have a data section, all datasets should be mentioned there. | Agreed, will be elaborated. LC, soil and DEM datasets used for GBR90 and EXTRUSO parameterization will be described more in detail. |
| 8 | The content of the results section and discussion section is not clearly separated. In the results section results are discussed and, in the discussion, new results are presented. Put the results from the discussion in the results section, and if necessary, give a description in the methods section | Agreed partly. See comment 4. |
| 9 | Line 17: I suggest deleting "…and various goodness of fit criteria", because the reader can assume that you do this, when you validate something. | Agreed, will be removed. |
| 10 | Line 25: "…yields approximately 2/3 of the total precipitation" Please add a source for this statement. | Agreed, will be added. |
| 11 | Line 25 -26: "However, with the need of higher spatial and temporal resolution, evaporation exposes larger variability" The | Agreed, will be elaborated. |

| | | |
|---|---|---|
| | context to the preceding lines is not clear to me. Please reword. I suggest adding some sentences to improve the readability. | |
| 12 | Line 34: "eddy-covariance lysimeter" to "eddy-covariance and lysimeter"? | Agreed, will be corrected. |
| 13 | Line 34: "Bowen ratio, gradient, experimental water balance watershed", please be more specific. | Agreed, will be elaborated. |
| 14 | Line 36 - 37: "…a space of scale and time. This footprint…", please check your wording. For the eddy flux community, the context is maybe clear, for other readers maybe not. I think some part of the explanation from line 118-119 should be stated here. | Agreed, will be rephrased. |
| 15 | Line 54: "and evaporation measurements themselves" Do you mean the uncertainty of evaporation measurements used for validation? Please change the wording. | Agreed, will be rephrased. |
| 16 | Line 68: "Data" - The section data does not contain information about many input datasets, which are quoted in "3.1. BROOK90 setups". | Agreed. See comment 7. |
| 17 | Line 72 – 73: "The average temperature varies between -15 °C and +15 °C in summer month", are you sure with -15 in summer month? Why could it be colder in summer than in winter? | Agreed, will be corrected. |
| 18 | Line 81: yarrow to common yarrow? I suggest using Latin names. | Agreed, will be added. |
| 19 | Line 87: Are some of the sites affected by groundwater? How did you solve that problem with Brook90? | According to measurements, the GW table for all sites is at least 3 m deep, thus we assume, that there is no significant GW influence of E. Unfortunately, Brook has simplified the GW module (1 order bucket) which does not allow the influence of GW on E (only as a reservoir for discharge delay). |
| 20 | Line 100 – 101: Do you have a citation for the carbon budget? | We could cite the original datasets published within ICOS, which contain the data behind the statement (https//doi.org/10.18160/YVR0-4898 https//doi.org/10.18160/2G60-ZHAK) |
| 21 | Line 172: "can be set easily (as location or slope)" I can´t imagine that it is easy to set values for 100 parameters. Or did you use in most cases the default parameters provided by Brook90? In that case, please note it. | In most of the cases, these are either default (or adjusted for Saxony, since Federer's study sites, who developed BROOK90 were located in the US). Besides default parameters, and parameters which are valid for whole model site (like i.e. average duration of rain precipitation per month) we specified the following number of parameters for each HRU (related to topography, land cover and soils): 28 in GBR90, 28 in EXTRUSO and 38 in manual BR |
| 22 | Line 183-184: How did you represent forest floor vegetation in the model? Or does it not play a significant role in the three forests, in contradiction e.g., to Scots pine forests? | Unfortunately, Brook does not provide representation of undergrowth of forest floor vegetation. Will be added to model shortcomings. Yes, it does play a role. It could be somewhat significant only in Hetzdorf, since in Tharandt and Oberbaerenburg the undergrowth is very weak and forest floor is almost clear. |
| 23 | Line 192-193: Please specify sources for the datasets | Agreed, will be added. |

| 24 | Line 192: If I correctly understood the Amazon Web Service Terrain Tiles is a web service which chooses the best available DEM for a specific location. So please indicate which DEM was used for saxony. | Agreed, its SRTM30, will be added. |
|----|----|----|
| 25 | Line 199-200: Please specify a source for CORINE, BodenKarte50, Open Sensor Web. It is confusing: From 2.2 I expected that you use RaKliDa – Metdata, but here you state that you use Open Sensor Web. Please clarify. | Agreed, will be added. Originally Extruso used OpenSensor (point meteostation data), which data was used to produce gridded Raklida data. Will be clarified. |
| 26 | Line 205: Please specify a source for the DEM | Agreed, will be added. |
| 27 | Line 215 - 216: "Our main hypothesis is that the goodness of fit of the setups decreases from global to local scale (for both parameterization and forcing)." I would expect the opposite: that the goodness of fit would increase from global to local scale, because local measurements of evapotranspiration should fit better to local measured input data. Please give an explanation how you come to that hypothesis. Furthermore, I suggest stating your hypothesis at the end of the introduction. | Agreed, will be corrected. Both sentences will be moved to the end of the intro. |
| 28 | Line 226: I suggest deleting: "Since all the proposed metrics are well known, we omit formulas in main text and list them in" | Agreed, will be rephrased. |
| 29 | Line 236 – 241: Please give a table of the 20 Parameters with their final values. Please include in that table also the parameter values from the other model setups. I suggest including that table in the main body of the manuscript. | Agreed partly. See comment 1. Including the table in text will require too much space from our point of view and is not worth it. |
| 30 | Line 251: "Before discussing…", delete, because it is the results section. | Agreed, will be deleted. |
| 31 | Line 259: "which got worse …" I suggest to reword. | Agreed, will be rephrased. |
| 32 | Line 263-264: "It was relatively difficult to achieve good timing for the vegetation period even on a monthly scale" I don´t understand what you mean with "achieve"? | Agreed, will be rephrased. |
| 33 | Line 267 "good BIAS", change it to low bias? | Agreed, will be changed. |
| 34 | Line 281 "variance errors" Please use a consistent nomenclature for the statistics throughout the manuscript. | Agreed, will be checked and corrected. |
| 35 | Line 308 – 309 "not so well" "distinctly worse" I suggest describing the results without judgmental adjectives. | Agreed, will be rephrased. |
| 36 | Line 311-321: This paragraph contains many aspects of a discussion. I suggest to restrict the results section to a description of the results and discussing the results in the discussion section. | Agreed partly, this paragraph concludes this section of results without a discussion on their aspects. |
| 37 | Line 322-327: I´m not sure if estimating the uncertainties of KGE by "resampled time-series" contributes significantly to the manuscript. I think this aspect could be omitted, or make clear, why these results are important, at least discuss it in the discussion. | Agreed partly, resampling was conducted to show the possible performance spread due to general time-series shortage and occurrence of some extreme years (e.g. like wet 2003, 2012 or very dry 2018-2019). Will be elaborated in text. |
| 38 | Line 340: "bias and variability are, on the other side, overestimated" What does it mean? | We mean positive deviations from the optimal values (1). Will be rephrased. |
| 39 | Line 355 – 356: Please shift this information to the introduction or discussion. | Agreed, will be shifted in intro. |
| 40 | Line 389 – 402: this paragraph contains a lot of information which should be shifted to the results section. | Agreed partly. See comment 4. |
| 41 | Line 412: "solar elevation" to solar elevation angle? | Agreed, will be corrected. |
| 42 | Line 414 – 430: this paragraph contains a lot of information which should be shifted to the results section. | Agreed partly. See comment 4. |
| 43 | Line 419: "After obtaining a persistent positive BIAS in the forests" BIAS for which variable? | Agreed, will be rephrased. We are referring to evaporation. |

| 44 | Line 431: I´m sure that this is not the first paper which deals with uncertainties of eddy-flux measurements. Maybe some references will help to enhance this section. | Agreed, additional references will be added. |
| 45 | Shouldn't be citations within the text ordered by date? | Agreed, will be corrected according to HESS regulations. |
| 46 | Line 52: "Allen et al., 1998, p.56; Miralles et al., 2016, p.2" Check if this form of citation is correct. | Agreed, will be corrected. |
| 47 | Line 114: correct: "6.90C" | Agreed, will be corrected. |
| 48 | Line 166 & 189: check the citations. | Agreed, will be corrected. |

---

## Author Response (AR1)

Answers to comments of Reviewer 1

| № | Comment | Answer | Line numbers in new clear version |
|---|---------|--------|-----------------------------------|
| 1 | The authors use just 15 combinations of model parameterization and forcing data to arrive at different conclusions regarding the importance of the two for modeling evaporation. In my opinion, this is severely inadequate for a robust assessment of uncertainty, let alone making any absolute conclusions about the importance of either model parameterization or forcing data, especially for a model which has greater than 20 parameters for modeling evaporation. A systematic uncertainty quantification would involve Monte Carlo simulations with a robust sampling scheme such as the Latin hypercube (by varying model parameters and meteorological inputs). As it stands, the results do not offer any conclusive quantitative evidence and as such is very superficial, and frankly not very useful. | Agreed partly. Currently to our knowledge, there are only three existing BROOK90 setups and all of them are used in the presented study: two automatic frameworks and the manual model itself. 'Automatic' means that the framework is collecting all data necessary to run the model, without expert knowledge. Yes, Monte-Carlo simulations will be advantageous to study the BROOK90 model uncertainty more deeply. This, however, was not the main intention of the study and in our opinion goes far beyond the scope of a single article. For example Spank et all 2013 (https://doi.org/10.1016/j.jhydrol.2013.03.047) denoted the whole article to application of the abovementioned technique to study uncertainties of only the meteorological measurements on the evaporation simulations with BROOK90. We purposely used only real data for forcing and parameterization (best-per-scale) instead of statistical bootstrapping, as no one before did such an analysis for the BROOK90 frameworks. We wanted to emphasize the scale problematic, with the practical outcome. Mainly, in a presence of limited resources and data, do the global regional automatic frameworks deliver plausible E results and where the user should put more attention - accurate parameterization or meteorological input. Thus, the introduction was corrected. | 68-81 |
| 2 | I have doubts about what the authors term as uncertainty in "model parameterization". From what I can gather, the only difference among the two models (BROOK90 and EXTRUSO) is land cover type and some input datasets. I do not think this is enough to quantify the uncertainty in model parameterization. The difference in the different models would then mainly arise from the difference parameter values of the calibrated and non-calibrated models. I do not understand how this difference can be construed as parameterization uncertainty. Either the authors should choose models which have completely | Agreed partly. We understand the model parameter uncertainty as follows: 'inability to specify exact values of model parameters' (Renard et al 2010 https://doi.org/10.1029/2009WR008328). The model possesses around 100 physically meaningful parameters, however, "only" around 30 of them are recommended to be changed according to the developer (other parameters refer to as fixed), | 60-63, 68-81, 300-305, 433-451, 612 |

| | | | |
|---|---|---|---|
| | different evaporation models (Penman vs Priestley-Taylor vs Hargreaves etc) or present a more robust quantification of the model parameter uncertainty (Monte Carlo simulations described above). | namely the ones which describe vegetation, soil and topography. We extended this list to around 40 parameters The difference in parameters among the testes frameworks for each site is presented in updated Appendix C. Here we wanted to show the impact (uncertainty) of different BROOK90 parameterization schemes on accuracy of E simulations (automatically or manually derived for different scales from different datasets) – in general, not going into deep analysis of single parameters uncertainty. Additionally see comment №6.
Incorporation of other models or methods to simulate evaporation will go far beyond the scope of the main topic. Nevertheless, as such an example, in results we show a comparison of complex vs simple (BROOK90 vs FAO) model setups. Introduction was corrected. Comparison of BROOK90 with FAO model was highlighted with a separate section. List of model parameters for each framework was added in Appendix. | |
| 3 | In the same vein, the lack of uncertainty seen due to model forcings is just a function of the 3 datasets (in-situ, RaKliDa, and ERA5). The present analysis does not provide sufficient evidence that forcing uncertainty is not as important parameterization uncertainty (Vrugt et al. 2008). doi:10.1029/2007WR006720 | Agreed partly. See comment №1&6. | |
| 4 | The attempt to study the differences in the spatial scale of evaporation modeling is commendable. But the authors do not discuss the differences among the different models from the perspective of spatial scales sufficiently. It is quite obvious that a model calibrated with local data would perform better. However, the interesting thing is to understand the differences in the regional and global model. There is no discussion pertaining to this. I would think this is because of the inadequate sample space in which the study operates. I recommend that the authors perform a systematic quantitative assessment of uncertainty. | Agreed, discussion on the difference in model performance between scales (especially for the GBR90 and EXTRUSO frameworks) was elaborated with a new subsection (4.1). | 453-498 |
| 5 | Many of the design choices are not explained and seem adhoc,
The authors do not explain why a multi-objective optimizer was used here. Why attempt to create a Pareto-optimal solution for calibrating evaporation (growing period vs winter)?

Why compare ERA5 hourly and ERA5 daily? Why only 3 input datasets? I can imagine that for Europe there are many observed forcing datasets (such as E-CAD). | Agreed, argumentation will be elaborated. Pareto-front calibration was used to address two issues. First, as most of E occurs in the vegetation period, it was decided to separate this period from the whole year as winter months should have lesser 'weight' during model fitting. Second we tried to account for possible systematic errors of E-C | 217-233,
238-255,
283-284 |

| | Why was the BROOK90 and EXTRUSO model chosen for this study? | measurements themselves, which could be different in these two periods (i.e. Hollinger et al 2005 https://doi.org/10.1093/treephys/25.7.873, Widmoser et al 2021 https://doi.org/10.5194/hess-25-1151-2021, Twine et al 2000 https://doi.org/10.1016/S0168-1923(00)00123-4). Therefore the pareto front could help to choose an optimal parameter set (i.e. enhance winter month performance with insignificant loss of performance in vegetation period). | |
| | Why were only 20 parameters chosen? Was a sensitivity analysis conducted? Which are the most important parameters which contribute to the uncertainty? | The three used datasets represent 'state-of-the-art' meteorological datasets for global, regional and local scales for the study sites. RAKLIDA data is far better than E-CAD for regional scale as it was specifically designed and produced for Saxony and it has 1 km resolution (while E-CAD has 0.1-0.25 degree). Hourly ERA-5 data was applied only as an additional dataset, since it is implemented as primary forcing in the original GBR90 framework. However, for the comparability of three dataset's performance, ERA5 was upscaled to daily. Additionally we wanted to test and show the sensibility of the model to hourly vs daily data (see comment 11). | |
| | | Here we presume there is a small misunderstanding with the naming. BROOK90 is the model, which is the core of all setups. "EXTRUSO" (like 'Global BROOK90') is a framework (regional) which uses this model. | |
| | | For the calibration we initially took the physical 'location' parameters of the vegetation and flow parameters which are recommended by the developer and other researchers as the most sensible (Vilhar 2016 10.3832/ifor1630-008, Schwärzel et al 2009 https://doi.org/10.1016/j.foreco.2009.03.033, Habel et al 2021 https://forecomon2021.thuenen.de/fileadmin/forecomon/Presentations/132_Puhlmann_2.pdf, Groh et al 2013 http://dx.doi.org/10.5675/HyWa-2013,4-1). Then we conducted | |

| | | | |
|---|---|---|---|
| | | manual sensitivity analysis ('try-tests' with the given data) to come up with the chosen 21 parameters. | |
| 6 | In summary, the study as it stands is very superficial and the authors have to make a strong case for why a qualitative assessment is sufficient to understand the uncertainty in model parameterization and forcings. In my opinion, the evidence provided in the manuscript points to the contrary: uncertainty assessments need far more robust experiment design to weed out spurious conclusions. | The main intention of the study was not to make a detailed assessment of the model's parameters and forcing uncertainty. Rather we want to address the topic mentioned in the main title. Namely, how available parameterization schemes and meteo input and their scales influence BROOK90 performance regarding evaporation simulations and existing model setups. Thus, we suggest to rephrase/omit 'uncertainty' term confusion or use it with caution, pointing out that we did not present a quantitative 'meteo and parameter uncertainty' evaluation and elaborate the last two paragraphs in the intro. | 68-81 + in text (regarding term uncertainty) |
| 7 | The abstract is very vague. What is the main conclusion of the study? What is the main implication of the conclusion? | Agreed, elaborated regarding main outcomes. | 9-24 |
| 8 | The manuscript needs to be edited to remove some idiosyncratic language use. For example Line 9: "Evaporation occurs on each surface…", Line 26: "...evaporation exposes larger variability…". Line 28: "...deepening knowledge…". Line 41: "The project allocates standardized …". Line 65: "the parameter set or meteorological input" should be "the parameter set and meteorological input". | Agreed, checked and corrected. | 26-27, 30-32, 49 |
| 9 | Line 40: I am not sure FLUXNET is an operational measurement network. I would term it as a database which collates measurements from different flux tower sites. | Agreed, corrected. | 50 |
| 10 | Line 215. Do you mean that the goodnes of fit should increase (rather than decrease) from global to local scales? | Agreed, corrected. | 73-74 |
| 11 | Why did the ERA5 daily outperform ERA5 hourly? | We suppose two main reasons. At first, due to the shortcomings in the interception modules of BROOK90. It runs on subdaily basis and if no subdaily P is passed in, it uses 'daily average rain duration in hours' parameter (which varies for each month) to disaggregate daily P into hourly. Furthermore, there are other simplifications (i.e. omitting diurnal cycle of potential evaporation). Federer (model developer) says that the module that uses subdaily P data consistently produces too much interception. Second one could be the poor quality of subdaily precipitation distribution in the ERA5 data for the study region. It was found that on daily, | 530-535 |

| | | monthly and annual scales, ERA5 did not show a significant difference with the station data, which could account for that amount of differences in daily vs hourly KGE values.
Additionally, it could be a case that simulations with hourly P are actually closer to reality, and eddy-covariance measurements themselves systematically underestimate interception.
As we do not have enough evidence to check the plausibility of the abovementioned reasoning (five sites in one region, 10-30 years of data), thus we omitted discussion on this topic. However, we add our suggestions as a discussion statement. | |
|---|---|---|---|
| 12 | Line 355: This is a very absolutist claim. The partitioning of evaporation is a topic of major debate and the 60% estimate from Wei et al. 2017 is just one estimate. There is some uncertainty here varying from 55-85% depending on which study one considers. | Agreed, rephrased and elaborated. | 39-40 |
| 13 | Figure 7: It does not show which model result is shown in which pie chart. | Here we made an average from all model setups to derive general conclusions on the E partitioning for yearly and seasonal scale. Results for specific selected model setups are presented in Fig. 8. | 416, 432 |
| 14 | The results section uses very subjective terms to describe model performance (example, 'fairly good' in Line 404). | Agreed, deleted. | |
| 15 | Line 449: I do not understand "...underestimation of the real site footprint or by permanent". | Agreed, corrected, the last part was deleted. | |
| 16 | Line 487: "...parameterization gave us higher spread". Where is this higher spread quantified? I recommend the authors attach some numbers to such claims, just a visual inspection is not enough. | The spread was described quantitatively in section 3.1. Sentences will be rephrased to add some numbers (%). | 368-371, 590-591 |

Answers to comments of Reviewer 2

| № | Comment | Answer | Line numbers in clear version |
|---|---------|--------|-------------------------------|
| 1 | Parameter selection and parametrization is a central issue in the paper, but information about the parameters is mainly lacking. The cited literature for GBR90 (Vorobewskii et al. 2020) and for EXTR (Luong et al. 2020) list various sources for parameter groups without stating parameter values, too. Please include a table with the relevant parameters and their values which differ due to different soil and landcover input. | Agreed, the list of the parameters as well as their ranges (different from the default model ones) for each framework was presented in extra Appendix. | 612 |
| 2 | The final values of the calibrated parameters and for comparison, the parameter values for the other model set-ups are lacking. | Agreed, Appendix C was extended. | 612 |
| 3 | The concept of uncertainties in the paper is not clear. The reader would expect as a result confidence limits for the parameters and model outcome, which is not given. The authors should make clearer what they intent. | Agreed, elaborated. The main intention of the study was not to make a detailed assessment of the model's parameters and forcing uncertainty. Rather we want to address the topic mentioned in the main title. Namely, how available parameterization schemes, meteo input, and their scales influence BROOK90 performance regarding evaporation simulations. Thus, we suggest to rephrase/omit 'uncertainty' term confusion or use it with caution, pointing out that we did not present a quantitative 'meteo and parameter uncertainty' evaluation and elaborate the last two paragraphs in the intro. | 59-81 |
| 4 | In the discussion section main parts of the results, e.g., parametrization, are not discussed and new results are presented instead. The discussion nearly comes out without referring results from other researchers; therefore, the authors do not give proper credit to related work. | Agreed, the discussion section was reorganized and discussion of the initial results was elaborated. Namely, we added subsection on to discuss the performance with regard to scale (4.1) and partly moved some results from 4.3 subsection to result section (now subsection 2.6). | 300, 453 |
| 5 | The abstract does not contain results and a final outcome of the paper | Agreed, was elaborated. | 9-25 |
| 6 | Elaborating the introduction, work out a hypothesis and state it at the end of the introduction (and not within the method section) | Agreed, was elaborated. | 68-81 |
| 7 | Reorganize Data & Methods. Why not using traditional Material & methods – section? I suggest lifting "2.1 Eddy-covariance measurements" in the hierarchy and to do not subsume it under "Data", it is a central issue of the paper. When you have a data section, all datasets should be mentioned there. | Agreed, was elaborated. LC, soil and DEM datasets used for GBR90 and EXTRUSO parameterization were described more in detail. Section 2 was elaborated. | 82 |

| | | | |
|---|---|---|---|
| 8 | The content of the results section and discussion section is not clearly separated. In the results section results are discussed and, in the discussion, new results are presented. Put the results from the discussion in the results section, and if necessary, give a description in the methods section | Agreed. See comment 4. Sections 3 and 4 were elaborated. | |
| 9 | Line 17: I suggest deleting "…and various goodness of fit criteria", because the reader can assume that you do this, when you validate something. | Agreed, was removed. | |
| 10 | Line 25: "…yields approximately 2/3 of the total precipitation" Please add a source for this statement. | Agreed, was added. | 29 |
| 11 | Line 25 -26: "However, with the need of higher spatial and temporal resolution, evaporation exposes larger variability" The context to the preceding lines is not clear to me. Please reword. I suggest adding some sentences to improve the readability. | Agreed, was elaborated. | 30-32 |
| 12 | Line 34: "eddy-covariance lysimeter" to "eddy-covariance and lysimeter"? | Agreed, was corrected. | 41 |
| 13 | Line 34: "Bowen ratio, gradient, experimental water balance watershed", please be more specific. | Agreed, was elaborated. | 41-42 |
| 14 | Line 36 - 37: "…a space of scale and time. This footprint…", please check your wording. For the eddy flux community, the context is maybe clear, for other readers maybe not. I think some part of the explanation from line 118-119 should be stated here. | Agreed, was rephrased. | 44 |
| 15 | Line 54: "and evaporation measurements themselves" Do you mean the uncertainty of evaporation measurements used for validation? Please change the wording. | Agreed, was rephrased. | 62 |
| 16 | Line 68: "Data" - The section data does not contain information about many input datasets, which are quoted in "3.1. BROOK90 setups". | Agreed. See comment 7. Elaborated in Subsection 2.3. | 173 |
| 17 | Line 72 – 73: "The average temperature varies between -15 °C and +15 °C in summer month", are you sure with -15 in summer month? Why could it be colder in summer than in winter? | Agreed, was corrected. | 87 |
| 18 | Line 81: yarrow to common yarrow? I suggest using Latin names. | Agreed, was added. | 94-96 |
| 19 | Line 87: Are some of the sites affected by groundwater? How did you solve that problem with Brook90? | According to measurements, the GW table for all sites is at least 3 m deep, thus we assume, that there is no significant GW influence of E. Unfortunately, Brook has simplified the GW module (1 order bucket) which does not allow the influence | 133-134 |

| | | | |
|---|---|---|---|
| | | of GW on E (only as a reservoir for discharge delay). | |
| 20 | Line 100 – 101: Do you have a citation for the carbon budget? | We could cite the original datasets published within ICOS, which contain the data behind the statement (https//doi.org/10.18160/YVR0-4898  https//doi.org/10.18160/2G60-ZHAK) | 115-116 |
| 21 | Line 172: "can be set easily (as location or slope)" I can´t imagine that it is easy to set values for 100 parameters. Or did you use in most cases the default parameters provided by Brook90? In that case, please note it. | In most of the cases, these are either default (or adjusted for Saxony, since Federer's study sites, who developed BROOK90 were located in the US). Besides default parameters, and parameters which are valid for whole model site (like i.e. average duration of rain precipitation per month)  we specified the following number of parameters for each HRU (related to topography, land cover and soils): 28 in GBR90, 28 in EXTRUSO and 38 in manual BR. | 153,  612 |
| 22 | Line 183-184: How did you represent forest floor vegetation in the model? Or does it not play a significant role in the three forests, in contradiction e.g., to Scots pine forests? | Unfortunately, Brook does not provide representation of undergrowth of forest floor vegetation. Will be added to model shortcomings. Yes, it does play a role. It could be somewhat significant only in Hetzdorf, since in Tharandt and Oberbaerenburg the undergrowth is very weak and forest floor is almost clear. | 503 |
| 23 | Line 192-193: Please specify sources for the datasets | Agreed, was added. | 184-188 |
| 24 | Line 192: If I correctly understood the Amazon Web Service Terrain Tiles is a web service which chooses the best available DEM for a specific location. So please indicate which DEM was used for saxony. | Agreed, its SRTM30, was added. | 191-192 |
| 25 | Line 199-200: Please specify a source for CORINE, BodenKarte50, Open Sensor Web. It is confusing: From 2.2 I expected that you use RaKliDa – Metdata, but here you state that you use Open Sensor Web. Please clarify. | Agreed, was added. Originally, in automatic mode EXTRUSO used OpenSensor (point meteostation data), which data was used to produce gridded Raklida data, which is quality-checked state-of-the-art climate dataset for Saxony and used in the study. | 201-203 |
| 26 | Line 205: Please specify a source for the DEM | Agreed, was added. | 204 |
| 27 | Line 215 - 216: "Our main hypothesis is that the goodness of fit of the setups decreases from global to local scale (for both parameterization and forcing)." I | Agreed, was corrected. Sentences were moved to the end of the intro. | 73-74 |

| | | | |
|---|---|---|---|
| | would expect the opposite: that the goodness of fit would increase from global to local scale, because local measurements of evapotranspiration should fit better to local measured input data. Please give an explanation how you come to that hypothesis. Furthermore, I suggest stating your hypothesis at the end of the introduction. | | |
| 28 | Line 226: I suggest deleting: "Since all the proposed metrics are well known, we omit formulas in main text and list them in" | Agreed, was rephrased. | 293 |
| 29 | Line 236 – 241: Please give a table of the 20 Parameters with their final values. Please include in that table also the parameter values from the other model setups. I suggest including that table in the main body of the manuscript. | Agreed partly. See comment 1. Including the table in text will require too much space (7 pages) from our point of view and is not worth it. | 612 |
| 30 | Line 251: "Before discussing…", delete, because it is the results section. | Agreed, was deleted. | |
| 31 | Line 259: "which got worse …" I suggest to reword. | Agreed, was rephrased. | 315-316 |
| 32 | Line 263-264: "It was relatively difficult to achieve good timing for the vegetation period even on a monthly scale" I don´t understand what you mean with "achieve"? | Agreed, was rephrased. | 322-323 |
| 33 | Line 267 "good BIAS", change it to low bias? | Agreed, was changed. | 325 |
| 34 | Line 281 "variance errors" Please use a consistent nomenclature for the statistics throughout the manuscript. | Agreed, was checked and corrected. | 325 + whole text |
| 35 | Line 308 – 309 "not so well" "distinctly worse" I suggest describing the results without judgmental adjectives. | Agreed, was rephrased. | 365 |
| 36 | Line 311-321: This paragraph contains many aspects of a discussion. I suggest to restrict the results section to a description of the results and discussing the results in the discussion section. | Agreed. We moved the discussion part to new discussion subsection 4.1. | 453 |
| 37 | Line 322-327: I´m not sure if estimating the uncertainties of KGE by "resampled time-series" contributes significantly to the manuscript. I think this aspect could be omitted, or make clear, why these results are important, at least discuss it in the discussion. | Agreed partly, resampling was conducted to show the possible performance spread due to general time-series shortage and occurrence of some extreme years (e.g. like wet 2003, 2012 or very dry 2018-2019). Was elaborated in text. | 294-299 |
| 38 | Line 340: "bias and variability are, on the other side, overestimated" What does it mean? | We mean positive deviations from the optimal values (1). Was rephrased. | 392 |

| | | | |
|---|---|---|---|
| 39 | Line 355 – 356: Please shift this information to the introduction or discussion. | Agreed, according to recommendations from Rev1 sentence was rephrased. | 40 |
| 40 | Line 389 – 402: this paragraph contains a lot of information which should be shifted to the results section. | Agreed partly. We moved the part to the description of the meteo data in Material and Methods section | 267-279 |
| 41 | Line 412: "solar elevation" to solar elevation angle? | Agreed, was corrected. | 511 |
| 42 | Line 414 – 430: this paragraph contains a lot of information which should be shifted to the results section. | Agreed partly. After careful consideration whether we can put this in Results section, we decided to leave it here, since the info solely denotes to discussion of the possible BROOK90 net energy problem (which was found not sufficient to be responsible for the amount of systematic overestimation of winter evaporation in forests) | |
| 43 | Line 419: "After obtaining a persistent positive BIAS in the forests" BIAS for which variable? | Agreed, was rephrased. We are referring to evaporation. | 518 |
| 44 | Line 431: I´m sure that this is not the first paper which deals with uncertainties of eddy-flux measurements. Maybe some references will help to enhance this section. | Agreed, additional references were added. | 537-570 |
| 45 | Shouldn't be citations within the text ordered by date? | Agreed, was checked and corrected according to HESS regulations. "In terms of in-text citations, the order can be based on relevance, as well as chronological or alphabetical listing, depending on the author's preference". We have chosen the last one. | All text |
| 46 | Line 52: "Allen et al., 1998, p.56; Miralles et al., 2016, p.2" Check if this form of citation is correct. | Agreed, was removed. | |
| 47 | Line 114: correct: "6.90C" | Agreed, was corrected. | 129 |
| 48 | Line 166 & 189: check the citations. | Agreed, was corrected. | 148, 170 |